# Adversarial Attacks on Fairness of Graph Neural Networks

**Binchi Zhang**[1]**, Yushun Dong**[1]**, Chen Chen**[1]**, Yada Zhu**[2]**, Minnan Luo**[3]**, Jundong Li**[1]

[1]University of Virginia        [2]IBM Research        [3]Xi'an Jiaotong University

`epb6gw@virginia.edu`

## Abstract

Fairness-aware graph neural networks (GNNs) have gained a surge of attention as they can reduce the bias of predictions on any demographic group (e.g., female) in graph-based applications. Although these methods greatly improve the algorithmic fairness of GNNs, the fairness can be easily corrupted by carefully designed adversarial attacks. In this paper, we investigate the problem of adversarial attacks on fairness of GNNs and propose *G-FairAttack*, a general framework for attacking *various types of* fairness-aware GNNs in terms of fairness with an *unnoticeable effect* on prediction utility. In addition, we propose a fast computation technique to reduce the time complexity of G-FairAttack. The experimental study demonstrates that G-FairAttack successfully corrupts the fairness of different types of GNNs while keeping the attack unnoticeable. Our study on fairness attacks sheds light on potential vulnerabilities in fairness-aware GNNs and guides further research on the robustness of GNNs in terms of fairness.

## 1 Introduction

Graph Neural Networks (GNNs) have achieved remarkable success across various human-centered applications, e.g., social network analysis (Qiu et al., 2018; Lu & Li, 2020; Feng et al., 2022), recommender systems (Ying et al., 2018; Fan et al., 2019; Yu et al., 2021), and healthcare (Choi et al., 2017; Li et al., 2022b; Fu et al., 2023). Despite that, many recent studies (Dai & Wang, 2021; Wang et al., 2022b; Dong et al., 2022b; 2023b) have shown that GNNs could yield biased predictions for nodes from certain demographic groups (e.g., females). With the increasing significance of GNNs in high-stakes human-centered scenarios, addressing such prediction bias becomes imperative. Consequently, substantial efforts have been devoted to developing fairness-aware GNNs, with the goal of learning fair node representations that can be used to make accurate and fair predictions. Typical strategies to improve the fairness of GNNs include adversarial training (Bose & Hamilton, 2019; Dai & Wang, 2021), regularization (Navarin et al., 2020; Fan et al., 2021; Agarwal et al., 2021; Dong et al., 2023c), and edge rewiring (Spinelli et al., 2021; Li et al., 2021; Dong et al., 2022a).

Although these fairness-aware GNN frameworks can make fair predictions, there remains a high risk of the exhibited fairness being corrupted by malicious attackers (Hussain et al., 2022). Adversarial attacks threaten the fairness of the victim model by perturbing the input graph data, resulting in unfair predictions. Such fairness attacks can severely compromise the reliability of GNNs, even when they have a built-in fairness-enhancing mechanism. For instance, we consider a GNN-based recommender system in social networks. In scenarios regarding commercial competition or personal benefits, a malicious adversary might conduct fairness attacks by hijacking inactive user accounts. The adversary can collect user data from these accounts to train a fairness attack model. According to the fairness attack algorithm, the adversary can then modify the attributes and connections of these compromised accounts. As a result, the GNN-based system is affected by the poisoned input data and makes biased predictions against a specific user group.

To protect the fairness of GNN models from adversarial attacks, we should first fully understand the potential ways to attack fairness. To this end, we investigate the problem of adversarial attacks on fairness of GNNs in this paper. Despite the importance of investigating the fairness attack of GNNs, most existing studies (Zügner et al., 2018; Zügner & Günnemann, 2019; Bojchevski & Günnemann, 2019; Dai et al., 2018; Xu et al., 2019) only focus on attacking the utility of GNNs (e.g., prediction

accuracy) and neglect the vulnerability of GNNs in the fairness aspect. To study fairness attacks, we follow existing works of attacks on GNNs to formulate the attack as an optimization problem and consider the prevalent gray-box attack setting (Zügner et al., 2018; Zügner & Günnemann, 2019; Wang & Gong, 2019; Sun et al., 2019) where the attacker cannot access the architecture or parameters of the victim model (Jin et al., 2021; Sun et al., 2022). A common strategy in gray-box attacks is to train a surrogate model (Zügner et al., 2018; Zügner & Günnemann, 2019) to gain more knowledge on the unseen victim model for the attack. Compared with conventional adversarial attacks, our attack on fairness is more difficult for the following two challenges: **(1) The design of the surrogate loss function**. Previous attacks on prediction utility directly choose the loss function adopted by most victim models, i.e. the cross-entropy (CE) loss as the surrogate loss. However, fairness-aware GNN models are trained based on different loss functions for fairness, e.g., demographic parity loss (Navarin et al., 2020; Zeng et al., 2021; Franco et al., 2022; Jiang et al., 2022), mutual information loss (Kang et al., 2022), and Wasserstein distance loss (Fan et al., 2021; Dong et al., 2022a). Because the victim model is unknown and can be any type of fairness-aware GNN, the attacker should find a proper surrogate loss to represent different types of loss functions of fairness-aware GNNs. **(2) The necessity to make such attacks unnoticeable**. If the poisoned graph data exhibits any clues of being manipulated, the model owner could recognize them and then take defensive actions (Zügner et al., 2018), e.g., abandoning the poisoned input data. Conventionally, attackers restrict the perturbation size to make attacks unnoticeable. In fairness attacks, we argue that a distinct change of prediction utility can also be a strong clue of being manipulated. However, none of the existing works on fairness attacks has considered the unnoticeable utility change.

In light of these challenges, we propose a novel fairness attack method on GNNs named **G-FairAttack**, which consists of two parts: a carefully designed surrogate loss function and an optimization method. To tackle the first challenge, we categorize existing fairness-aware GNNs into three types based on their fairness loss terms. Then we propose a novel surrogate loss function to help the surrogate model learn from all types of fairness-aware GNNs with theoretical analysis. To address the second challenge, we propose a novel unnoticeable constraint in utility change to make the fairness attack unnoticeable. Then we propose a non-gradient attack algorithm to solve the constrained optimization problem, which is verified to have a better performance than previous gradient-based methods. Moreover, we propose a fast computation strategy to improve the scalability of G-FairAttack. Our contributions can be summarized as follows.

- **Attack Setting.** We consider a novel unnoticeable constraint on prediction utility change for unnoticeable fairness attacks of GNNs, which can be extended to general fairness attacks.
- **Objective Design.** We propose a novel surrogate loss to help the surrogate model learn from various types of fairness-aware GNNs with theoretical analysis.
- **Algorithmic Design.** To solve the optimization problem with the unnoticeable constraint, we discuss the deficiency of previous gradient-based optimization methods and design a non-gradient attack method. In addition, we propose a fast computation approach to reduce its time complexity.
- **Experimental Evaluation.** We conduct extensive experiments on three real-world datasets with four types of victim models and verify that our proposed G-FairAttack successfully jeopardizes the fairness of various fairness-aware GNNs with an unnoticeable effect on prediction utility.

## 2 PROBLEM DEFINITION

**Notation and Preliminary.** We use bold uppercase letters, e.g., $\mathbf{X}$, to denote matrices, and use $\mathbf{X}_{[i,:]}$, $\mathbf{X}_{[:,j]}$, and $\mathbf{X}_{[i,j]}$ to denote the $i$-th row, the $j$-th column, and the element at the $i$-th row and the $j$-th column, respectively. We use bold lowercase letters, e.g., $\boldsymbol{x}$, to denote vectors, and use $\boldsymbol{x}_{[i]}$ to denote the $i$-th element. We use $P_X(\cdot)$ and $F_X(\cdot)$ to denote the probability density function and the cumulative distribution function for a random variable $X$, respectively. We use $\mathcal{G} = \{\mathcal{V}, \mathcal{E}, \mathbf{X}\}$ to denote an undirected attributed graph. $\mathcal{V} = \{v_1, \ldots, v_n\}$ and $\mathcal{E} \subseteq \mathcal{V} \times \mathcal{V}$ denote the node set and the edge set, respectively, where $n = |\mathcal{V}|$ is the number of nodes. $\mathbf{X} \in \mathbb{R}^{n \times d_x}$ denotes the attribute matrix, where $d_x$ is the number of node attributes. We use $\mathbf{A} \in \{0,1\}^{n \times n}$ to denote the adjacency matrix, where $\mathbf{A}_{[i,j]} = 1$ when $(i,j) \in \mathcal{E}$ and $\mathbf{A}_{[i,j]} = 0$ otherwise. In the node classification task, some nodes are associated with ground truth labels. We use $\mathcal{V}_{\text{train}}$ to denote the labeled (training) set, $\mathcal{V}_{\text{test}} = \mathcal{V} \backslash \mathcal{V}_{\text{train}}$ to denote the unlabeled (test) set, and $\mathcal{Y}$ to denote the set of labels. Most GNNs take the adjacency matrix $\mathbf{A}$ and the attribute matrix $\mathbf{X}$ as the input, and obtain

the node embeddings used for node classification tasks. Specifically, we use $f_{\boldsymbol{\theta}} : \{0, 1\}^{n \times n} \times \mathbb{R}^{n \times d_x} \to \mathbb{R}^{n \times c}$ to denote a GNN model, where $\boldsymbol{\theta}$ collects all learnable model parameters. We use $\hat{\boldsymbol{y}} = \sigma\left(f_{\boldsymbol{\theta}}(\mathbf{A}, \mathbf{X})\right)$ to denote soft predictions where $\sigma(\cdot)$ is the softmax function. A conventional way to train a GNN model is minimizing a utility loss function $\mathcal{L}\left(f_{\boldsymbol{\theta}}, \mathbf{A}, \mathbf{X}, \mathcal{Y}, \mathcal{V}_{\text{train}}\right)$ (e.g., CE loss) over the training set. In human-centered scenarios, each individual is usually associated with sensitive attributes (e.g., gender). We use $\mathcal{S} = \{s_1, \ldots, s_n\}$ to denote a sensitive attribute set, where $s_i$ is the sensitive attribute of the node $v_i$. Based on the sensitive attributes, we can divide the nodes into different sensitive groups, denoted as $\mathcal{V}_1, \ldots, \mathcal{V}_K$, where $\mathcal{V}_k = \{v_i | s_i = k\}$ and $K$ is the number of sensitive groups. Compared with vanilla GNNs, fairness-aware GNNs should not yield discriminatory predictions against individuals from any specific sensitive subgroup (Dong et al., 2023a). Hence, the objective of training fairness-aware GNNs generally contains two aspects; one is minimizing the utility loss, i.e., $\mathcal{L}\left(f_{\boldsymbol{\theta}}, \mathbf{A}, \mathbf{X}, \mathcal{Y}, \mathcal{V}_{\text{train}}\right)$, the other is minimizing the discrimination between the predictions over different sensitive groups, denoted as $\mathcal{L}_f\left(f_{\boldsymbol{\theta}}, \mathbf{A}, \mathbf{X}, \mathcal{Y}, \mathcal{S}\right)$. It is worth noting that $\mathcal{L}$ is the CE loss in most cases, but there are various types of $\mathcal{L}_f$ for training fair GNNs.

**Problem Formulation.** In the fairness attack of GNNs, attackers aim to achieve their goals (e.g., to make the predictions of the victim GNN exhibit more bias) via slightly perturbing the input graph data in certain feasible spaces. Here, we focus on structure perturbation because perturbing the graph structure in the discrete domain presents more challenges, and the attack methods based on structure perturbation can be easily adapted to attribute perturbation. In this paper, for the simplicity of discussion, we focus on two different sensitive groups and prediction classes, but our method can be easily adapted to tasks with multiple sensitive groups and classes.

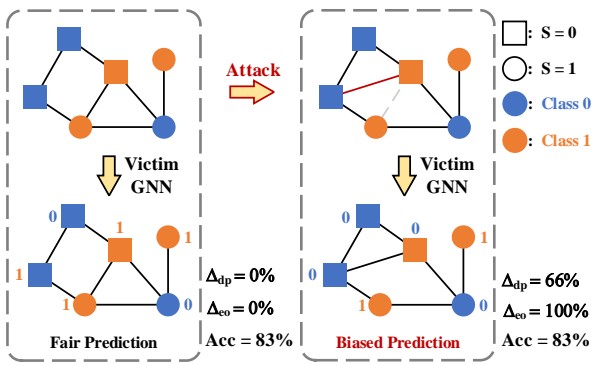

Figure 1: A toy example of fairness attacks of GNNs with unnoticeable effect on prediction utility.

More details about our attack settings are discussed in Appendix B. We formulate our problem of fairness attacks of GNNs as follows.

**Problem 1.** *Attacks on Fairness of GNNs with Unnoticeable Effect on Prediction Utility. Let $\mathcal{G} = \{\mathcal{V}, \mathcal{E}, \mathbf{X}\}$ be an attributed graph with the adjacency matrix $\mathbf{A}$. Let $\mathcal{V}_{train}$ and $\mathcal{V}_{test}$ be the labeled and unlabeled node sets, respectively, $\mathcal{Y}$ be the label set corresponding to $\mathcal{V}_{train}$, and $\mathcal{S}$ be the sensitive attribute value set of $\mathcal{V}$. Let $g_{\boldsymbol{\theta}}$ be a surrogate GNN model, $\mathcal{L}_s$, $\mathcal{L}$, and $\mathcal{L}_f$ be the surrogate loss, utility loss, and fairness loss, respectively. Given the attack budget $\Delta$ and the utility variation budget $\epsilon$, the problem of attacks on fairness of GNNs is formulated as follows.*

$$\max_{\mathbf{A}' \in \mathcal{F}} \mathcal{L}_f\left(g_{\boldsymbol{\theta}^*}, \mathbf{A}', \mathbf{X}, \mathcal{Y}, \mathcal{V}_{test}, \mathcal{S}\right)$$

$$\text{s.t. } \boldsymbol{\theta}^* = \arg\min_{\boldsymbol{\theta}} \mathcal{L}_s\left(g_{\boldsymbol{\theta}}, \tilde{\mathbf{A}}, \mathbf{X}, \mathcal{Y}, \mathcal{S}\right), \ \|\mathbf{A}' - \mathbf{A}\|_F \le 2\Delta, \quad (1)$$

$$|\mathcal{L}(g_{\boldsymbol{\theta}^*}, \mathbf{A}, \mathbf{X}, \mathcal{Y}, \mathcal{V}_{train}) - \mathcal{L}(g_{\boldsymbol{\theta}^*}, \mathbf{A}', \mathbf{X}, \mathcal{Y}, \mathcal{V}_{train})| \le \epsilon,$$

*where $\mathcal{F}$ is the feasible space of the poisoned graph structure $\mathbf{A}'$, $g_{\boldsymbol{\theta}^*}$ is a trained surrogate model.*

In this problem, the attacker's objective $\mathcal{L}_f(g_{\boldsymbol{\theta}^*}, \mathbf{A}', \mathbf{X}, \mathcal{Y}, \mathcal{V}_{\text{test}}, \mathcal{S})$ measures the bias of the predictions on test nodes. The surrogate model $g_{\boldsymbol{\theta}^*}$ is trained based on a surrogate loss $\mathcal{L}_s(g_{\boldsymbol{\theta}}, \tilde{\mathbf{A}}, \mathbf{X}, \mathcal{Y}, \mathcal{S})$. The constraints restrict the number of perturbed edges and the absolute value of the utility change over the training set. In this paper, we consider two types of attack scenarios, *fairness evasion attack* and *fairness poisoning attack* (Chakraborty et al., 2018). By perturbing the input graph data, *fairness evasion attack* jeopardizes the fairness of a well-trained fairness-aware GNN model in the test phase, while *fairness poisoning attack* makes the fairness-aware GNN model trained on such perturbed data render unfair predictions. If $\tilde{\mathbf{A}} = \mathbf{A}$, then Problem 1 boils down to the fairness evasion attack; and if $\tilde{\mathbf{A}} = \mathbf{A}'$, then Problem 1 becomes fairness poisoning attack. In this paper, we use demographic parity (Navarin et al., 2020) as the attacker's objective $\mathcal{L}_f$, the CE loss as utility loss $\mathcal{L}$, and a two-layer linearized GCN (Zügner et al., 2018) as the surrogate model $g_{\boldsymbol{\theta}}$. These choices have been verified by many previous works (Zügner et al., 2018; Zügner & Günnemann, 2019; Xu et al., 2019) to be effective in attacking the prediction utility of GNNs, while can also be changed

flexibly according to the specific task. An illustration of our problem is shown in Figure 1. We leave the detailed explanation of Figure 1 in Appendix B.

## 3 METHODOLOGY

In this section, we first introduce the design of the surrogate loss $\mathcal{L}_s$, which is the most critical part in Problem 1. Then, we propose a non-gradient optimization method to solve Problem 1 and propose a fast computation technique to reduce the time complexity of our optimization method.

### 3.1 SURROGATE LOSS DESIGN

**Existing Loss on Fairness.** The loss functions of all existing fairness-aware GNNs can be divided into two parts: utility loss term (CE loss that is shared in common) and fairness loss term (varies from different methods). Correspondingly, our surrogate loss function is designed as $\mathcal{L}_s = \mathcal{L}_{s_u} + \alpha \mathcal{L}_{s_f}$ where $\mathcal{L}_{s_u}$ is the utility loss term, i.e., the CE loss, $\mathcal{L}_{s_f}$ is the fairness loss term we aim to study, and $\alpha$ is the weight coefficient. Let $\hat{Y} \in [0,1]$ be a continuous random variable of the output soft prediction and $S$ be a discrete random variable of the sensitive attribute. Next, we categorize the fairness loss terms into three types. **(1).** $\Delta_{dp}(\hat{Y}, S) = |\Pr(\hat{Y} \geq \frac{1}{2}|S = 0) - \Pr(\hat{Y} \geq \frac{1}{2}|S = 1)|$: demographic parity (Jiang et al., 2022; Navarin et al., 2020; Zeng et al., 2021; Franco et al., 2022), that reduces the difference of positive rate among different sensitive groups during training. **(2).** $I(\hat{Y}, S) = \int_0^1 \sum_i P_{\hat{Y},S}(z,i) \log \frac{P_{\hat{Y},S}(z,i)}{P_{\hat{Y}}(z)\Pr(S=i)} dz$: mutual information of the output and the sensitive attribute (Dai & Wang, 2021; Bose & Hamilton, 2019; Kang et al., 2022), that reduces the dependence between the model output and the sensitive attribute during training. **(3).** $W(\hat{Y}, S) = \int_0^1 |F_{\hat{Y}|S=0}^{-1}(y) - F_{\hat{Y}|S=1}^{-1}(y)| dy$: Wasserstein-1 distance of the output on different sensitive groups (Fan et al., 2021; Dong et al., 2022a), that makes the conditional distributions of the model output given different sensitive groups closer during training.

**The Proposed Surrogate Loss.** Next, we introduce our proposed surrogate loss term on fairness $\mathcal{L}_{S_f}$, aiming at representing all types of aforementioned fairness loss terms, e.g., $\Delta_{dp}(\hat{Y}, S)$, $I(\hat{Y}, S)$, and $W(\hat{Y}, S)$. In particular, our idea is to find a common upper bound for these fairness loss terms as $\mathcal{L}_{S_f}$. Therefore, during the training of the surrogate model, $\mathcal{L}_{S_f}$ is minimized to a small value, and all types of fairness loss terms become even smaller consequently. In this way, the surrogate model trained by our surrogate loss will be close to that trained by any unknown victim loss, which is consistent with conventional attacks on model utility. Consequently, we argue that such surrogate loss can represent the unknown victim loss. Accordingly, we propose a novel surrogate loss function on fairness $TV(\hat{Y}, S) = \int_0^1 |P_{\hat{Y}|S=0}(z) - P_{\hat{Y}|S=1}(z)| dz$, i.e., the total variation of $\hat{Y}$ on different sensitive groups. We can prove that $TV(\hat{Y}, S)$ is a common upper bound of $\Delta_{dp}(\hat{Y}, S)$, $I(\hat{Y}, S)$ (at certain condition), and $W(\hat{Y}, S)$ so it can represent all types of fairness loss functions.

**Theorem 1.** *We have $\Delta_{dp}(\hat{Y}, S)$ and $W(\hat{Y}, S)$ upper bounded by $TV(\hat{Y}, S)$. Moreover, assuming $P_{\hat{Y}}(z) \geq \Pi_i \Pr(S = i)$ holds for any $z \in [0,1]$, $I(\hat{Y}, S)$ is also upper bounded by $TV(\hat{Y}, S)$.*

The proof of Theorem 1 is shown in Appendix A.1. We also provide a more practical variant of Theorem 1 with a weaker assumption in Appendix A.1. Combining $\mathcal{L}_{s_f}$ (total variation loss) and $\mathcal{L}_{s_u}$ (CE loss), our surrogate loss function is $\mathcal{L}_s = CE(g_{\boldsymbol{\theta}}, \tilde{\mathbf{A}}, \mathbf{X}, \mathcal{Y}) + \alpha TV(g_{\boldsymbol{\theta}}, \tilde{\mathbf{A}}, \mathbf{X}, \mathcal{S})$. Since the probability density function of the underlying random variable $\hat{Y}$ is difficult to calculate, we leverage the kernel density estimation (Parzen, 1962) to compute the total variation loss with the output soft predictions $g_{\boldsymbol{\theta}}(\tilde{\mathbf{A}}, \mathbf{X})$. The kernel estimation is computed as $\hat{P}_{\hat{Y}|S=i}(z) = \frac{1}{h|\mathcal{V}_i|} \sum_{j \in \mathcal{V}_i} K(\frac{z - g_{\boldsymbol{\theta}}(\tilde{\mathbf{A}},\mathbf{X})_{[j]}}{h})$ for $i = 0, 1$, where $K(\cdot)$ is a kernel function and $h$ is a positive bandwidth constant. Then we exploit the numerical integration to compute the total variation loss as

$$TV(g_{\boldsymbol{\theta}}, \tilde{\mathbf{A}}, \mathbf{X}, \mathcal{S}) = \frac{1}{m} \sum_{j=1}^{m} \left| \hat{P}_{\hat{Y}|S=0}\left(\frac{j}{m}\right) - \hat{P}_{\hat{Y}|S=1}\left(\frac{j}{m}\right) \right|, \tag{2}$$

where $m$ is the number of intervals in the integration. Consequently, we obtain a practical calculation of our proposed surrogate loss $\mathcal{L}_s$.

### 3.2 OPTIMIZATION

Considering that Problem 1 is a bilevel optimization problem in a discrete domain, it is extremely hard to find the exact solution. Hence, we resort to the greedy strategy and propose a sequential attack method. Our sequential attack flips target edges[1] sequentially to obtain the optimal poisoned structure $\mathbf{A}'$. It is worth noting that many attack methods on model utility find the target edge corresponding to the largest element of the gradient of $\mathbf{A}$, namely gradient-based methods (Zügner & Günnemann, 2019; Xu et al., 2019; Wu et al., 2019; Geisler et al., 2021). However, we find the efficacy of gradient-based methods is not guaranteed.

**Proposition 1.** *Gradient-based methods for optimizing the graph structure are not guaranteed to decrease the objective function.*

We leave a detailed analysis of the shortcomings of gradient-based methods in Appendix D. In contrast, we propose a non-gradient method based on a scoring function that can provably increase the attacker's objective. We first consider the unconstrained version of Problem 1. We let $\mathbf{A}^0 = \mathbf{A}$ and train the surrogate model to obtain $\boldsymbol{\theta}^0$, i.e., solving $\boldsymbol{\theta}^0 = \arg\min_{\boldsymbol{\theta}} \mathcal{L}_s(g_{\boldsymbol{\theta}}, \mathbf{A}, \mathbf{X}, \mathcal{Y}, \mathcal{S})$. For $t = 1, 2, \ldots, T_0$, we find the maximum score $r^t$ in the $t$-th iteration as

$$\max_{(u,v) \in \mathcal{C}^t} r^t(u,v) = \Delta \mathcal{L}_f^t(u,v) = \mathcal{L}_f(g_{\boldsymbol{\theta}^t}, flip_{(u,v)}\mathbf{A}^t) - \mathcal{L}_f(g_{\boldsymbol{\theta}^t}, \mathbf{A}^t), \tag{3}$$

where $\mathcal{C}^t$ is the candidate edge set in the $t$-th iteration; $flip_{(u,v)}\mathbf{A}$ denotes the adjacency matrix after flipping the edge $(u,v)$: $flip_{(u,v)}\mathbf{A}_{[i,j]} = 1 - \mathbf{A}_{[i,j]}$ if $(i,j) = (u,v)$ or $(i,j) = (v,u)$, and $flip_{(u,v)}\mathbf{A}_{[i,j]} = \mathbf{A}_{[i,j]}$ otherwise. After solving Equation (3), we denote the solution as $(u^t, v^t)$. Then we update $\mathbf{A}^t$ as $\mathbf{A}^{t+1} = flip_{(u^t,v^t)}\mathbf{A}^t$ in the $t$-th iteration. For the fairness poisoning attack, we retrain the surrogate model based on $\mathbf{A}^{t+1}$ as $\boldsymbol{\theta}^{t+1} = \arg\min_{\boldsymbol{\theta}} \mathcal{L}_s(g_{\boldsymbol{\theta}}, \mathbf{A}^{t+1})$ to update the surrogate model. After $T_0$ iterations, we have $\mathbf{A}^* = \mathbf{A}^{T_0}$ as the solution of Problem 1.

Next, to handle the first constraint, we let $T_0 \leq \Delta$ and have $\|\mathbf{A}^* - \mathbf{A}\|_F \leq \sum_{i=1}^{T_0} \|\mathbf{A}^t - \mathbf{A}^{t-1}\|_F \leq 2\Delta$ consequently. For the second constraint, we aim to make every flipping unnoticeable in terms of model utility, i.e., making $|\mathcal{L}(\mathbf{A}^{t+1}) - \mathcal{L}(\mathbf{A}^t)|$ as small as possible. We notice that this constrained optimization problem can be easily solved by projected gradient descent (PGD) (Nocedal & Wright, 2006) in the continuous domain by solving $\mathbf{A}^{t+1} = \arg\min_{|\mathcal{L}(\mathbf{A}^t) - \mathcal{L}(\mathbf{A}')| \leq \epsilon_t} \|\mathbf{A}' - (\mathbf{A}^t + \eta \nabla \mathcal{L}_f(\mathbf{A}^t))\|_F^2$, where $\epsilon_t$ is the budget of the $t$-th iteration that satisfies $\sum_{t=1}^{\Delta} \epsilon_t \leq \epsilon$ and $\eta$ is the learning rate. To solve this problem, we have the following theorem.

**Theorem 2.** *The optimal poisoned adjacency matrix $\mathbf{A}^{t+1}$ in the $t+1$-th iteration given by PGD, i.e., the solution of $\mathbf{A}^{t+1} = \arg\min_{|\mathcal{L}(\mathbf{A}^t) - \mathcal{L}(\mathbf{A}')| \leq \epsilon_t} \|\mathbf{A}' - (\mathbf{A}^t + \eta \nabla \mathcal{L}_f(\mathbf{A}^t))\|_F^2$ is*

$$\mathbf{A}^{t+1} = \begin{cases} \mathbf{A}^t + \eta \nabla_{\mathbf{A}} \mathcal{L}_f(\mathbf{A}^t), & \text{if } \eta |\nabla_{\mathbf{A}} \mathcal{L}(\mathbf{A}^t)^T \nabla_{\mathbf{A}} \mathcal{L}_f(\mathbf{A}^t)| \leq \epsilon_t, \\ \mathbf{A}^t + \eta \nabla_{\mathbf{A}} \mathcal{L}_f(\mathbf{A}^t) + \dfrac{e_t \epsilon_t - \eta \nabla_{\mathbf{A}} \mathcal{L}(\mathbf{A}^t)^T \nabla_{\mathbf{A}} \mathcal{L}_f(\mathbf{A}^t)}{\|\nabla_{\mathbf{A}} \mathcal{L}(\mathbf{A}^t)\|_F^2} \nabla_{\mathbf{A}} \mathcal{L}(\mathbf{A}^t), & \text{otherwise,} \end{cases} \tag{4}$$

*where $e_t = \text{sign}\left(\nabla_{\mathbf{A}} \mathcal{L}(\mathbf{A}^t)^T \nabla_{\mathbf{A}} \mathcal{L}_f(\mathbf{A}^t)\right)$.*

The proof of Theorem 2 is provided in Appendix A.2. It is worth noting that the solution in Equation (4) cannot be leveraged directly in the discrete domain. Hence, we use the differences $\Delta \mathcal{L}_f^t(u,v) = \mathcal{L}_f(flip_{(u,v)}\mathbf{A}^t) - \mathcal{L}_f(\mathbf{A}^t)$ and $\Delta \mathcal{L}^t(u,v) = \mathcal{L}(flip_{(u,v)}\mathbf{A}^t) - \mathcal{L}(\mathbf{A}^t)$ as zeroth-order estimations (Chen et al., 2017; Kariyappa et al., 2021) of $\nabla_{\mathbf{A}_{[u,v]}} \mathcal{L}_f(\mathbf{A}^t)$ and $\nabla_{\mathbf{A}_{[u,v]}} \mathcal{L}(\mathbf{A}^t)$, and replace them in Equation (4), respectively. Moreover, we minimize the utility budget as $\epsilon_t = 0$ to constrain the model utility change after the flipping as strictly as possible. Consequently, we adjust the scoring function to find the target edge as

$$\tilde{r}^t(u,v) = \Delta \mathcal{L}_f^t(u,v) - \frac{(\boldsymbol{p}^t)^T \boldsymbol{q}^t}{\|\boldsymbol{p}^t\|_2^2} |\Delta \mathcal{L}^t(u,v)|, \tag{5}$$

for $(u,v) \in \mathcal{C}^t$, where $\boldsymbol{p}^t \in \mathbb{R}^{|\mathcal{C}^t|}$ and $\boldsymbol{q}^t \in \mathbb{R}^{|\mathcal{C}^t|}$ are denoted as $\boldsymbol{p}_{[i]}^t = \Delta \mathcal{L}^t(\mathcal{C}_{[i]}^t)$ and $\boldsymbol{q}_{[i]}^t = \Delta \mathcal{L}_f^t(\mathcal{C}_{[i]}^t)$, respectively. Here $\mathcal{C}_{[i]}^t$ denotes the $i$-th edge in $\mathcal{C}^t$. Equation (5) can also be seen as a balanced attacker's objective between maximizing $\mathcal{L}_f(\mathbf{A}^{t+1}) - \mathcal{L}_f(\mathbf{A}^t)$ and minimizing $|\mathcal{L}(\mathbf{A}^{t+1}) - \mathcal{L}(\mathbf{A}^t)|$. With $\tilde{r}^t(u,v)$, the pseudocode of our proposed attack algorithm is shown in Appendix C.1.

---

[1]Flipping the target edge increases the attacker's objective the most.

### 3.3 FAST COMPUTATION

In this subsection, we focus on the efficient implementation of our sequential attack. The most important and costly part of our algorithm is to find the maximum value of $\tilde{r}^t(u,v)$ from $\mathcal{C}^t$. This naturally requires traversing each edge in $\mathcal{C}^t$, which is costly on a large graph. Hence, we develop a fast computation method to reduce the time complexity. Reviewing our model settings, we find that the output of node $i$, $g_{\boldsymbol{\theta}}(\mathbf{A},\mathbf{X})_{[i]}$, can be computed as $g_{\boldsymbol{\theta}}(\mathbf{A},\mathbf{X})_{[i]} = \sum_{j\in\mathcal{N}_i}(\hat{d}_{[i]}\hat{d}_{[j]})^{-1}\hat{\mathbf{A}}_{[j,:]}\mathbf{X}\boldsymbol{\theta} = \mathbf{Z}_{[i,:]}\boldsymbol{\theta}$, where $\hat{d}$ is the node degree vector, $\hat{\mathbf{A}} = \mathbf{A} + \mathbf{I}$, $\mathcal{N}_i = \{j|\hat{\mathbf{A}}_{[i,j]} = 1\}$, and $\mathbf{Z}$ denotes the aggregated feature matrix which is crucial to the fast computation. We aim to reduce the time complexity from two perspectives: (1) reducing the time complexity of computing the score $\tilde{r}^t(u,v)$ for a specific edge $(u,v)$; and (2) reducing the size of the candidate edge set $\mathcal{C}^t$. From perspective (1), we compute $flip_{(u,v)}\mathbf{Z}^t$ incrementally based on $\mathbf{Z}^t$. Then we have $g_{\boldsymbol{\theta}^t}(flip_{(u,v)}\mathbf{A}^t,\mathbf{X}) = flip_{(u,v)}\mathbf{Z}^t\boldsymbol{\theta}^t$. Consequently, both $\Delta\mathcal{L}_f^t(u,v)$ and $\Delta\mathcal{L}^t(u,v)$ can be obtained based on $g_{\boldsymbol{\theta}^t}(flip_{(u,v)}\mathbf{A}^t,\mathbf{X})$. Next, we discuss the computation of $flip_{(u,v)}\mathbf{Z}_{[i,:]}^t$ into three cases and introduce them separately. We only discuss adding edge $(u,v)$, and leave removing edge $(u,v)$ in Appendix C.1. **Case 1**: If $i\in\{u,v\}$, $flip_{(u,v)}\mathbf{Z}_{[i,:]}^t = \frac{\hat{d}_{[i]}^t}{\hat{d}_{[i]}^t+1}(\mathbf{Z}_{[i,:]}^t - \frac{\hat{\mathbf{A}}_{[i,:]}^t\mathbf{X}}{(\hat{d}_{[i]}^t)^2}) + \frac{\hat{\mathbf{A}}_{[i,:]}^t\mathbf{X}+\mathbf{X}_{[j,:]}}{(\hat{d}_{[i]}^t+1)^2} + \frac{\hat{\mathbf{A}}_{[j,:]}^t\mathbf{X}+\mathbf{X}_{[i,:]}}{(\hat{d}_{[i]}^t+1)(\hat{d}_{[j]}^t+1)}$, where $j\in\{u,v\}\backslash\{i\}$; **Case 2**: If $i\in\mathcal{N}_u^t\cup\mathcal{N}_v^t\backslash\{u,v\}$, $flip_{(u,v)}\mathbf{Z}_{[i,:]}^t = \mathbf{Z}_{[i,:]}^t - \mathbb{I}_{i\in\mathcal{N}_u^t}\cdot(\frac{\hat{\mathbf{A}}_{[u,:]}^t\mathbf{X}}{\hat{d}_{[i]}^t\hat{d}_{[u]}^t} - \frac{\hat{\mathbf{A}}_{[u,:]}^t\mathbf{X}+\mathbf{X}_{[v,:]}}{\hat{d}_{[i]}^t(\hat{d}_{[u]}^t+1)}) - \mathbb{I}_{i\in\mathcal{N}_v^t}\cdot(\frac{\hat{\mathbf{A}}_{[v,:]}^t\mathbf{X}}{\hat{d}_{[i]}^t\hat{d}_{[v]}^t} - \frac{\hat{\mathbf{A}}_{[v,:]}^t\mathbf{X}+\mathbf{X}_{[u,:]}}{\hat{d}_{[i]}^t(\hat{d}_{[v]}^t+1)})$, where $\mathbb{I}_{i\in\mathcal{N}} = 1$ if $i\in\mathcal{N}$, and $\mathbb{I}_{i\in\mathcal{N}} = 0$ otherwise; **Case 3**: If $i\notin\mathcal{N}_u^t\cup\mathcal{N}_v^t$, $flip_{(u,v)}\mathbf{Z}_{[i,:]}^t = \mathbf{Z}_{[i,:]}^t$. From perspective (2), we only choose the influential edges[2] as $\mathcal{C}^t$. Specifically, flipping $(u,v)$ is highly likely to increase $\mathcal{L}_f$ only if it results in significant changes to the predictions of a large number of nodes. Based on the aforementioned discussions, flipping $(u,v)$ can only affect the soft prediction of nodes in $\mathcal{N}_u^t\cup\mathcal{N}_v^t$ (case 1 and 2). Hence, we consider $(u,v)$ as an influential edge if the predictions of nodes in $\mathcal{N}_u^t\cup\mathcal{N}_v^t$ are easy to be changed. To this end, we propose an importance score $\rho^t(u,v)$ and collect the edges corresponding to the top $a$ importance scores into $\mathcal{C}^t$. We have $\rho^t(u,v) = \sum_{i\in\mathcal{N}_u^t\cup\mathcal{N}_v^t}M_t - |\mathbf{Z}_{[i,:]}^t\boldsymbol{\theta}^t|$, where $M_t = \max_i|\mathbf{Z}_{[i,:]}^t\boldsymbol{\theta}^t|$. When the value of $M_t - |\mathbf{Z}_{[i,:]}^t\boldsymbol{\theta}^t|$ is large, the prediction of node $i$ is easy to be changed. We analyze the time complexity of our fast computation method as follows.

**Proposition 2.** *The overall time complexity of G-FairAttack with the fast computation is $O(\bar{d}n^2 + d_xan)$, where $\bar{d}$ denotes the average degree.*

Compared with computing $\max_{(u,v)\in\mathcal{C}^t}\tilde{r}^t(u,v)$ directly, which has a complexity of $O(n^4)$, our fast computation approach distinctly improves the efficiency of the attack. Note that the time complexity of G-FairAttack can be further improved in practice by parallel computation and simpler ranking strategies. The proof of Proposition 2, detailed complexity analysis, and ways of further improving the complexity are provided in Appendix C.2.

## 4 EXPERIMENTS

In this section, we evaluate our proposed G-FairAttack. Specifically, we aim to answer the following questions through our experiments: **RQ1:** Can our proposed surrogate loss function represent the victim loss functions of different kinds of victim models? **RQ2:** Can G-FairAttack achieve unnoticeable utility change? **RQ3:** To what extent does our fast computation approach improve the efficiency of G-FairAttack? Due to the space limitation, we provide more experimental results and discussions on the results in Appendix F.

### 4.1 EXPERIMENTAL SETTINGS

The evaluation of attack methods includes two stages. In the first stage, we use an attack method to obtain the perturbed graph structure. In the second stage, for the fairness evasion attack, we train a test GNN (i.e., victim model) on the clean graph and compare the output of the test GNN

---

[2]Flipping the influential edges is highly likely to increase the attacker's objective.

Table 1: Experiment results of fairness evasion attack, where '-' means the out-of-memory case. To test the effectiveness of attack methods, we compare the fairness metric ($\Delta_{dp}$) of the results of attack methods and the clean input data (larger value means less fair). We highlight the two most effective attack methods (with the largest fairness drop) with bold and underline for each victim model.

| Victim | Attack | Facebook | | Pokec_z | | Credit | |
|---|---|---|---|---|---|---|---|
| | | ACC(%) | $\Delta_{dp}$(%) | ACC(%) | $\Delta_{dp}$(%) | ACC(%) | $\Delta_{dp}$(%) |
| GCN | Clean | $80.89 \pm 0.00$ | $5.30 \pm 0.47$ | $67.09 \pm 0.13$ | $7.13 \pm 1.17$ | $70.75 \pm 0.45$ | $13.30 \pm 1.89$ |
| | Random | $80.68 \pm 0.18$ | $5.04 \pm 0.27$ | $67.09 \pm 0.20$ | $\mathbf{7.11 \pm 1.48}$ | $71.35 \pm 0.36$ | $13.53 \pm 2.12$ |
| | FA-GNN | $80.46 \pm 0.18$ | $\mathbf{8.73 \pm 0.29}$ | $66.16 \pm 0.14$ | $1.57 \pm 0.60$ | $71.02 \pm 0.54$ | $\mathbf{15.02 \pm 2.25}$ |
| | Gradient Ascent | $80.68 \pm 0.49$ | $6.46 \pm 0.79$ | $67.17 \pm 0.17$ | $3.59 \pm 0.66$ | - | - |
| | G-FairAttack | $81.00 \pm 0.37$ | $\underline{8.13 \pm 0.74}$ | $67.24 \pm 0.29$ | $\underline{5.08 \pm 1.38}$ | $71.00 \pm 0.52$ | $\underline{13.64 \pm 1.86}$ |
| Reg | Clean | $80.36 \pm 0.18$ | $3.06 \pm 0.58$ | $66.75 \pm 0.11$ | $2.41 \pm 0.29$ | $71.60 \pm 2.69$ | $0.63 \pm 0.52$ |
| | Random | $80.15 \pm 0.18$ | $0.22 \pm 0.11$ | $66.36 \pm 0.11$ | $2.00 \pm 0.60$ | $71.66 \pm 2.66$ | $0.65 \pm 0.65$ |
| | FA-GNN | $79.73 \pm 0.18$ | $\underline{4.67 \pm 0.58}$ | $66.16 \pm 0.09$ | $\underline{5.83 \pm 0.33}$ | $71.81 \pm 2.75$ | $\underline{1.25 \pm 0.33}$ |
| | Gradient Ascent | $80.36 \pm 0.18$ | $3.99 \pm 0.58$ | $66.89 \pm 0.28$ | $3.66 \pm 0.46$ | - | - |
| | G-FairAttack | $81.00 \pm 0.37$ | $\mathbf{6.84 \pm 0.31}$ | $65.97 \pm 0.18$ | $\mathbf{6.03 \pm 0.54}$ | $71.32 \pm 2.83$ | $\mathbf{3.01 \pm 1.23}$ |
| FairGNN | Clean | $79.94 \pm 0.64$ | $2.00 \pm 1.77$ | $65.42 \pm 0.52$ | $1.37 \pm 0.83$ | $71.98 \pm 1.81$ | $2.65 \pm 0.98$ |
| | Random | $79.14 \pm 0.32$ | $0.37 \pm 0.38$ | $64.96 \pm 0.54$ | $1.35 \pm 0.71$ | $72.00 \pm 1.80$ | $\underline{2.72 \pm 0.95}$ |
| | FA-GNN | $79.14 \pm 0.32$ | $\underline{3.00 \pm 2.31}$ | $64.86 \pm 0.64$ | $\mathbf{3.27 \pm 0.75}$ | $71.98 \pm 1.79$ | $1.98 \pm 1.15$ |
| | Gradient Ascent | $79.86 \pm 0.54$ | $2.32 \pm 2.41$ | $65.01 \pm 0.53$ | $1.40 \pm 0.77$ | - | - |
| | G-FairAttack | $79.46 \pm 0.61$ | $\mathbf{3.23 \pm 2.03}$ | $65.33 \pm 0.55$ | $\underline{2.63 \pm 0.75}$ | $72.01 \pm 1.82$ | $\mathbf{2.79 \pm 1.52}$ |
| EDITS | Clean | $79.30 \pm 0.00$ | $0.27 \pm 0.00$ | $64.68 \pm 0.35$ | $1.89 \pm 0.83$ | $70.99 \pm 1.63$ | $7.49 \pm 1.64$ |
| | Random | $77.39 \pm 0.64$ | $\underline{4.65 \pm 0.94}$ | $64.68 \pm 0.35$ | $1.89 \pm 0.83$ | $69.88 \pm 1.40$ | $\mathbf{7.56 \pm 2.30}$ |
| | FA-GNN | $78.77 \pm 0.18$ | $0.60 \pm 0.35$ | $64.68 \pm 0.35$ | $1.89 \pm 0.83$ | $70.99 \pm 1.63$ | $\underline{7.50 \pm 1.63}$ |
| | Gradient Ascent | $77.07 \pm 0.00$ | $3.25 \pm 0.94$ | $64.68 \pm 0.35$ | $1.89 \pm 0.83$ | - | - |
| | G-FairAttack | $78.24 \pm 0.36$ | $\mathbf{5.55 \pm 0.47}$ | $64.91 \pm 0.63$ | $\mathbf{4.25 \pm 0.77}$ | $69.88 \pm 1.40$ | $\mathbf{7.56 \pm 2.30}$ |

on the clean graph and on the perturbed graph; for the fairness poisoning attack, we train the test GNN on the clean graph to obtain the normal model and train the test GNN on the perturbed graph to obtain the victim model; then, we compare the output of the normal model on the clean graph and the output of the victim model on the perturbed graph. We adopt three prevalent real-world datasets, i.e., Facebook (Leskovec & Mcauley, 2012), Credit (Agarwal et al., 2021), and Pokec (Dai & Wang, 2021; Dong et al., 2022a) to test the effectiveness of G-FairAttack. For attack baselines, we choose a random attack method (Hussain et al., 2022; Zügner et al., 2018), a state-of-the-art fairness attack method FA-GNN (Hussain et al., 2022), and two gradient-based methods Gradient Ascent and Metattack (Zügner & Günnemann, 2019) (adapted from utility attacks). For test GNNs, we adopt a vanilla graph convolutional network (Kipf & Welling, 2017) and three different types of fairness-aware GNNs, Regularization (Zeng et al., 2021) (Reg) for $\Delta_{dp}(\hat{Y}, S)$, FairGNN (Dai & Wang, 2021) for $I(\hat{Y}, S)$, and EDITS (Dong et al., 2022a) for $W(\hat{Y}, S)$. More details about baselines and datasets are provided in Appendices E.1 and E.2. We choose two mostly adopted fairness metrics, demographic parity $\Delta_{dp}$ (Dwork et al., 2012) and equal opportunity $\Delta_{eo}$ (Hardt et al., 2016), to measure the fairness of test GNNs. Larger values of fairness metrics denote more bias. In addition, we also report the accuracy and AUC score to show the utility of test GNNs.

## 4.2 EFFECTIVENESS OF ATTACK

We compare G-FairAttack with three attack baselines in the fairness evasion attack and fairness poisoning attack (in Appendix F.1) settings on three prevalent real-world datasets. Specifically, to test the effectiveness of attack methods, we choose four different kinds of victim GNN models to test the degradation of fairness metric values after being attacked. The attack budget $\Delta$ for Facebook and Pokec is 5%, i.e., we can flip $5\% \cdot |\mathcal{E}|$ edges at most; for Credit, the budget is 1%. We ran each experiment three times with different random seeds and reported the mean value and the standard deviation. The experimental results are shown in Table 1 and Appendix F. We can observe that: (1). G-FairAttack is the most effective attack method that jeopardizes the fairness of all types of victim GNN models, especially for fairness-aware GNNs. (2). Compared with Gradient Ascent, G-FairAttack adds a fairness loss term (total variation loss) in the surrogate loss. Consequently, G-FairAttack outperforms them in attacking different types of fairness-aware GNNs, which demonstrates that our proposed surrogate loss helps our surrogate model learn from different types of fairness-aware victim models (**RQ1**). (3). Some baselines outperform G-FairAttack in attacking

vanilla GCN because the surrogate model of G-FairAttack is trained with the surrogate loss, including a fairness loss term, while the victim GCN is trained without fairness consideration. This problem can be addressed by choosing a smaller value of $\alpha$ (weight of $\mathcal{L}_f$). Despite that, G-FairAttack successfully reduces the fairness of GCN on most benchmarks. (4). Compared with gradient-based attacks (Gradient Ascent), G-FairAttack has less space complexity as it does not need to store a dense adjacency matrix.

## 4.3 ABLATION STUDY

**Effectiveness of Surrogate Loss.** To further answer **RQ1**, we compare G-FairAttack with two variants with different surrogate losses to demonstrate the effectiveness of our proposed surrogate loss. Our proposed surrogate loss function $\mathcal{L}_s$ consists of two parts: utility loss term $\mathcal{L}_{s_u}$ (CE loss) and

Table 2: Results of G-FairAttack on Credit while replacing the total variation loss with other loss terms.

| Attack | FairGNN | | EDITS | |
|---|---|---|---|---|
| | $\Delta_{dp}$ (%) | $\Delta_{eo}$ (%) | $\Delta_{dp}$ (%) | $\Delta_{eo}$ (%) |
| Clean | $6.70 \pm 1.60$ | $4.52 \pm 1.79$ | $6.43 \pm 0.95$ | $5.77 \pm 1.06$ |
| G-FA-None | $7.87 \pm 1.31$ | $5.68 \pm 1.42$ | $10.47 \pm 1.91$ | $9.95 \pm 1.71$ |
| G-FA-$\Delta_{dp}$ | $7.82 \pm 1.17$ | $5.82 \pm 1.58$ | $10.77 \pm 1.19$ | $10.39 \pm 1.17$ |
| G-FA | $8.06 \pm 1.90$ | $6.10 \pm 2.03$ | $11.26 \pm 1.50$ | $10.87 \pm 1.56$ |

fairness loss term $\mathcal{L}_{s_f}$ (total variation loss). For the first baseline, we remove the fairness loss term from the surrogate loss and name it G-FairAttack-None. For the second baseline, we substitute the total variation loss with $\Delta_{dp}$ loss, named as G-FairAttack-$\Delta_{dp}$. In our problem, if the surrogate loss is the same as the victim loss (like white-box attacks), the attack method would definitely perform well. However, we show that G-FairAttack (with our proposed total variation surrogate loss) has the most desirable performance *even with a different victim loss from the surrogate loss*, which verifies the effectiveness of our surrogate loss. Hence, we choose FairGNN and EDITS as the victim models because their victim loss functions differ from the surrogate loss functions of G-FairAttack-None and G-FairAttack-$\Delta_{dp}$. We conduct the experiment on Credit dataset in the fairness poisoning attack setting. The results are shown in Table 2. It is demonstrated that all three attack methods successfully increase the value of fairness metrics. Among all three attack methods, G-FairAttack achieves the best performance, which demonstrates that our proposed surrogate loss helps the surrogate model learn knowledge from various types of fairness-aware GNNs.

**Effectiveness of Unnoticeable Constraints.** To answer **RQ2**, we investigate the impact of utility constraints on G-FairAttack. We remove the discrete projected update strategy and the utility constraint in G-FairAttack as G-FairAttack-C. We compare the variation of $\Delta_{dp}$ and the utility loss on the training set after the attack by G-FairAttack, G-FairAttack-C, and FA-GNN. We chose two mostly adopted victim models, i.e., GCN and regularization, to test the attack methods on Facebook. We set the utility budget at 5% of the utility loss on clean data. The results are shown in Figure 2. It demonstrates that: (1). Our G-FairAttack distinctly deteriorates the fairness of two victim models while keeping the variation of utility loss unnoticeable ($< 0.5\%$). Therefore, G-FairAttack can attack the victim model with unnoticeable utility

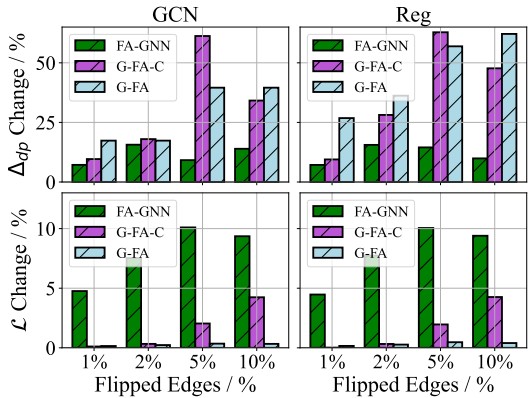

Figure 2: The changes of $\Delta_{dp}$ and the utility loss function $\mathcal{L}$ under different attack budgets.

variation. (2). Removing the utility constraint and the discrete projected update strategy from G-FairAttack, the variation of utility loss becomes much larger as the attack budget increases. The results of G-FairAttack-C verify the effectiveness of the fairness constraint and the discrete projected update strategy. (3). G-FairAttack and G-FairAttack-C deteriorate the fairness of victim models to a larger extent than FA-GNN. Because G-FairAttack is a sequential greedy method, it becomes more effective as the attack budget increases. Hence, G-FairAttack is more flexible than FA-GNN.

## 4.4 PARAMETER STUDY

In this subsection, we aim to answer **RQ3** and show the impact of the parameter $a$ (the threshold for fast computation) on the time cost for the attack and the test results for victim models. We conduct

experiments under different choices of $a$. For the simplicity of the following discussion, we take $a$ as the proportion of edges considered in $\mathcal{C}^t$ (e.g., $a = 1e^{-3}$ means we only consider the top $0.1\%$ edges with the highest important score). We record the attacker's objective $\mathcal{L}_f$ under different thresholds during the optimization process as Figure 3(a) to show the impact of $a$ of the surrogate model.

We also record the fairness metrics of regularization-based victim models attacked by G-FairAttack and the average time of G-FairAttack in each iteration while choosing different values of $a$ (in Figure 3(b)) to show the impact of $a$ on the attack performance and efficiency. Figure 3(b) demonstrates the trade-off between effectiveness and efficiency exists but is very tolerant on the effectiveness side. In the range $[5e^{-4}, 1e^{-2}]$, the performances of G-FairAttack on the victim model are similar, while the time cost decreases distinctly when $a$ gets smaller (the time cost of $a = 5e^{-4}$ is 95% lower than $a = 1e^{-2}$). Therefore, in practice,

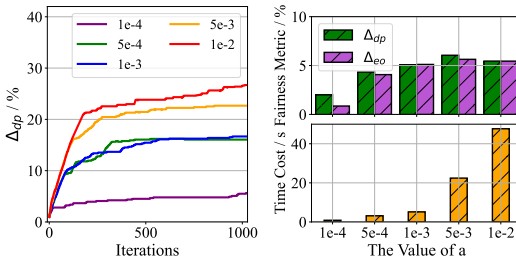

(a) Curves of attacker's objective.

(b) Test fairness metrics and training time cost.

Figure 3: The optimization curves, training time cost, and test fairness on regularization-based victim model corresponding to different values of $a$.

we can flexibly choose a proper threshold to fit the efficiency requirement. In conclusion, our fast computation distinctly reduces G-FairAttack's time cost without compromising the performance.

## 5 RELATED WORKS

**Adversarial Attack of GNNs.** To improve the robustness of GNNs, we should first fully understand how to attack GNNs. Consequently, various attacking strategies of GNNs have been proposed in recent years. Zügner et al. (2018) first formulated the poisoning attack problem on graph data as a bilevel optimization in the discrete domain and proposed the first poisoning attack method of GNNs with respect to a single target node in a gray-box setting based on the greedy strategy. Following the problem formulation in (Zügner et al., 2018), researchers proposed many effective adversarial attack methods in different settings. Zügner & Günnemann (2019) proposed the first untargeted poisoning attack method in a gray-box setting based on MAML (Finn et al., 2017). Chang et al. (2020a) proposed an untargeted evasion attack method in a black-box setting by attacking the low-rank approximation of the spectral graph filter. Wu et al. (2019) proposed an untargeted evasion attack method in a white-box setting and a corresponding defense method based on integrated gradients.

**Adversarial Attacks on Fairness.** Many recent studies investigated the fairness attack problem on tabular data. Van et al. (2022) proposed an online poisoning attack framework on fairness based on gradient ascent. They adopted the convex relaxation (Zafar et al., 2017; Donini et al., 2018) to make the loss function differentiable. Mehrabi et al. (2021) proposed two types of poisoning attacks on fairness. One incorporates demographic information into the influence attack (Koh et al., 2022), and the other generates poisoned samples within the vicinity of a chosen anchor to skew the decision boundary. Solans et al. (2021) studied the fairness poisoning attack as a bilevel optimization and solved it by KKT relaxation with a novel initialization strategy. Chhabra et al. (2023) proposed a black-box attack and defense method for fair clustering. For graph data, Hussain et al. (2022) is the first to investigate the fairness attack problem, proposing FA-GNN that randomly injects links among different sensitive groups to promote demographic parity. In a concurrent study, Kang et al. (2023) investigated the fairness attack problem as a bilevel optimization and proposed FATE, a meta-learning-based fairness attack framework that targets both group fairness and individual fairness.

## 6 CONCLUSION

Fairness-aware GNNs play a significant role in various human-centered applications. To protect GNNs from fairness attacks, we should fully understand potential ways to attack the fairness of GNNs. In this paper, we propose the first unnoticeable requirement for fairness attacks. We design a novel surrogate loss function to attack various types of fairness-aware GNNs. We also propose a sequential attack algorithm to solve the problem, and a fast computation approach to reduce the time complexity. Extensive experiments on three real-world datasets verify the efficacy of our method.

ACKNOWLEDGEMENTS

Binchi Zhang and Jundong Li are supported by the Commonwealth Cyber Initiative award HV-2Q23-003, JP Morgan Chase Faculty Research Award, and Cisco Faculty Research Award. Yada Zhu is supported in part by MIT-IBM Watson AI Lab – a research collaboration as part of the IBM AI Horizons Network.

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

# Appendix

## Table of Contents

## A PROOF

### A.1 PROOF OF THEOREM 1

**Theorem 1.** *We have $\Delta_{dp}(\hat{Y}, S)$ and $W(\hat{Y}, S)$ upper bounded by $TV(\hat{Y}, S)$. Moreover, $I(\hat{Y}, S)$ is also upper bounded by $TV(\hat{Y}, S)$ if $\forall z \in [0, 1]$, $P_{\hat{Y}}(z) \geq \Pi_i \Pr(S = i)$ holds.*

*Proof.* $\Delta_{dp}(\hat{Y}, S)$, $I(\hat{Y}, S)$, $W(\hat{Y}, S)$, and $TV(\hat{Y}, S)$ are all non-negative. We first prove that $\Delta_{dp}(\hat{Y}, S)$ is upper bounded by $TV(\hat{Y}, S)$. Recall that $TV(\hat{Y}, S) = \int_0^1 |P_{\hat{Y}|S=0}(z) - P_{\hat{Y}|S=1}(z)| dz$, we have

$$
\begin{aligned}
\Delta_{dp}(\hat{Y}, S) &= \left| \Pr\left( \hat{Y} \geq \frac{1}{2} \mid S = 0 \right) - \Pr\left( \hat{Y} \geq \frac{1}{2} \mid S = 1 \right) \right| \\
&= \left| \int_{\frac{1}{2}}^1 P_{\hat{Y}|S=0}(z) - P_{\hat{Y}|S=1}(z) dz \right| \\
&\leq \int_{\frac{1}{2}}^1 \left| P_{\hat{Y}|S=0}(z) - P_{\hat{Y}|S=1}(z) \right| dz \\
&\leq \int_0^1 \left| P_{\hat{Y}|S=0}(z) - P_{\hat{Y}|S=1}(z) \right| dz \\
&= TV(\hat{Y}, S).
\end{aligned}
$$

Next, we prove that $W(\hat{Y}, S)$ is also upper bounded by $TV(\hat{Y}, S)$.

$$
\begin{aligned}
W(\hat{Y}, S) &= \int_0^1 \left| F^{-1}_{\hat{Y}|S=0}(y) - F^{-1}_{\hat{Y}|S=1}(y) \right| dy \\
&= \int_0^1 \left| F_{\hat{Y}|S=0}(z) - F_{\hat{Y}|S=1}(z) \right| dz.
\end{aligned}
\tag{6}
$$

This equation holds because we know that $F_{\hat{Y}|S=0}(0) = F_{\hat{Y}|S=1}(0) = 0$ and $F_{\hat{Y}|S=0}(1) = F_{\hat{Y}|S=1}(1) = 1$ according to the property of cumulative distribution function and the fact that $\hat{Y} \in [0, 1]$. Hence $y = F_{\hat{Y}|S=0}(z)$ and $y = F_{\hat{Y}|S=1}(z)$ form a closed curve in $[0, 1] \times [0, 1]$ in z-y plane. Consequently, Equation (6) could be seen as computing the area of the closed curve from the y-axis and z-axis separately. Consequently, we have

$$
\begin{aligned}
W(\hat{Y}, S) &= \int_0^1 \left| F_{\hat{Y}|S=0}(z) - F_{\hat{Y}|S=1}(z) \right| dz \\
&= \int_0^1 \left| \int_0^x P_{\hat{Y}|S=0}(z) dz - \int_0^x P_{\hat{Y}|S=1}(z) dz \right| dx \\
&\leq \int_0^1 \int_0^x \left| P_{\hat{Y}|S=0}(z) dz - P_{\hat{Y}|S=1}(z) \right| dz dx \\
&= \int_0^{x'} \left| P_{\hat{Y}|S=0}(z) dz - P_{\hat{Y}|S=1}(z) \right| dz, \; x' \in [0, 1] \\
&\leq \int_0^1 \left| P_{\hat{Y}|S=0}(z) dz - P_{\hat{Y}|S=1}(z) \right| dz \\
&= TV(\hat{Y}, S).
\end{aligned}
$$

Finally, we prove that $I(\hat{Y}, S)$ is upper bounded by $TV(\hat{Y}, S)$ if $\forall z \in [0, 1]$, $P_{\hat{Y}}(z) \geq \Pi_i \Pr(S = i)$. First, we have

$$
I(\hat{Y}, S) = \int_0^1 \sum_i P_{\hat{Y}, S}(z, i) \log \frac{P_{\hat{Y}, S}(z, i)}{P_{\hat{Y}}(z) \Pr(S = i)} dz
$$

$$= \int_0^1 \sum_i \Pr(S=i) P_{\hat{Y}|S=i}(z) \log \frac{P_{\hat{Y}|S=i}(z)}{P_{\hat{Y}}(z)} dz.$$

Let $P_i = \Pr(S=i)$ for $i = 0, 1$, then we have $P_0 + P_1 = 1$ and $P_{\hat{Y}}(z) = P_0 P_{\hat{Y}|S=0}(z) + P_1 P_{\hat{Y}|S=1}(z)$. According to the fact that $\log x \leq x - 1$ for $x \in (0, 1]$, we let $x = \frac{P_{\hat{Y}|S=i}(z)}{P_{\hat{Y}}(z)}$ and have

$$
\begin{aligned}
I(\hat{Y}, S) &= \int_0^1 \sum_i P_i P_{\hat{Y}|S=i}(z) \log \frac{P_{\hat{Y}|S=i}(z)}{P_{\hat{Y}}(z)} dz \\
&\leq \int_0^1 \sum_i P_i \frac{\left(P_{\hat{Y}|S=i}(z)\right)^2}{P_{\hat{Y}}(z)} - P_i P_{\hat{Y}|S=i}(z) dz \\
&= \int_0^1 \sum_i P_i \frac{P_{\hat{Y}|S=i}(z) \left(P_{\hat{Y}|S=i}(z) - P_{\hat{Y}}(z)\right)}{P_{\hat{Y}}(z)} dz \\
&= \int_0^1 \frac{P_0 P_1 P_{\hat{Y}|S=0}(z) \left(P_{\hat{Y}|S=0}(z) - P_{\hat{Y}|S=1}(z)\right)}{P_0 P_{\hat{Y}|S=0}(z) + P_1 P_{\hat{Y}|S=1}(z)} \\
&\quad + \frac{P_0 P_1 P_{\hat{Y}|S=1}(z) \left(P_{\hat{Y}|S=1}(z) - P_{\hat{Y}|S=0}(z)\right)}{P_0 P_{\hat{Y}|S=0}(z) + P_1 P_{\hat{Y}|S=1}(z)} dz \\
&= \int_0^1 \frac{P_0 P_1 \left(P_{\hat{Y}|S=0}(z) - P_{\hat{Y}|S=1}(z)\right)^2}{P_0 P_{\hat{Y}|S=0}(z) + P_1 P_{\hat{Y}|S=1}(z)} dz.
\end{aligned}
\tag{7}
$$

Given that $P_{\hat{Y}}(z) \geq \Pi_i \Pr(S=i) \, \forall z \in [0, 1]$, we have $P_0 P_{\hat{Y}|S=0}(z) + P_1 P_{\hat{Y}|S=1}(z) \geq P_0 P_1 (P_0 + P_1) = P_0 P_1$. Consequently, we have

$$I(\hat{Y}, S) \leq \int_0^1 \left(P_{\hat{Y}|S=0}(z) - P_{\hat{Y}|S=1}(z)\right)^2 dz.$$

Considering that the training of fairness-aware GNNs makes the distributions $P_{\hat{Y}|S=0}(z)$ and $P_{\hat{Y}|S=1}(z)$ closer, we assume that $|P_{\hat{Y}|S=0}(z) - P_{\hat{Y}|S=1}(z)| \leq 1$ for $z \in [0, 1]$. To verify this assumption, we conduct numerical experiments on all three adopted datasets. Following our methodology, we use the kernel density estimation to estimate the distribution functions $P_{\hat{Y}|S=0}(z)$ and $P_{\hat{Y}|S=1}(z)$ and compute the value of $|P_{\hat{Y}|S=0}(z) - P_{\hat{Y}|S=1}(z)|$ consequently. We record the largest value of $|P_{\hat{Y}|S=0}(z) - P_{\hat{Y}|S=1}(z)|$ for $z \in [0, 1]$ and obtain the results as $0.1372 \pm 0.0425$ for Facebook, $0.0999 \pm 0.0310$ for Pokec_z, and $0.0356 \pm 0.0074$ for Credit (mean value and standard deviation under 5 random seeds), which are all far less than 1. Then, we come to

$$I(\hat{Y}, S) \leq \int_0^1 \left|P_{\hat{Y}|S=0}(z) - P_{\hat{Y}|S=1}(z)\right| dz = TV(\hat{Y}, S). \tag{8}$$

In conclusion, we have proved that $\Delta_{dp}(\hat{Y}, S)$ and $W(\hat{Y}, S)$ are upper bounded by $TV(\hat{Y}, S)$. Moreover, $I(\hat{Y}, S)$ is also upper bounded by $TV(\hat{Y}, S)$ if $\forall z \in [0, 1]$, $P_{\hat{Y}}(z) \geq \Pi_i \Pr(S = i)$ holds.

$\square$

**Remarks on Theorem 1.** It is worth noting that $I(\hat{Y}, S) \leq TV(\hat{Y}, S)$ stems from the condition of $P_{\hat{Y}}(z) \geq \Pi_i \Pr(S = i), \forall z \in [0, 1]$. Although we are not able to always ensure the correctness of the condition in practice, we can still obtain from Theorem 1 that (1) the probability of the condition holds grows larger when the number of sensitive groups increases; (2) even in the binary case, the condition is highly likely to hold in practice, considering that $\Pr(S = 0) \cdot \Pr(S = 1) \leq \frac{1}{4}$ (In a binary case, we have $\Pr(s = 0) + \Pr(s = 1) = 1$; hence, $\Pr(s = 0)\Pr(s = 1) = \Pr(s = 0)(1 - \Pr(s = 0)) \leq 1/4$).

To further improve the soundness of our theoretical analysis, we can slightly loosen the condition $P_{\hat{Y}}(z) \geq \Pi_i \Pr(S = i) \; \forall z \in [0,1]$ and obtain a new condition $\int_0^1 \left( \frac{P_0 P_1}{P_0 P_{\hat{Y}|S=0}(z) + P_1 P_{\hat{Y}|S=1}(z)} \right)^2 dz \leq 1$. Consider the last step of Equation (7). According to the Cauchy-Schwartz inequality, we have

$$I(\hat{Y}, S) \leq \int_0^1 \frac{P_0 P_1 \left( P_{\hat{Y}|S=0}(z) - P_{\hat{Y}|S=1}(z) \right)^2}{P_0 P_{\hat{Y}|S=0}(z) + P_1 P_{\hat{Y}|S=1}(z)} dz$$

$$\leq \left( \int_0^1 \left( \frac{P_0 P_1}{P_0 P_{\hat{Y}|S=0}(z) + P_1 P_{\hat{Y}|S=1}(z)} \right)^2 dz \cdot \int_0^1 \left( P_{\hat{Y}|S=0}(z) - P_{\hat{Y}|S=1}(z) \right)^4 dz \right)^{\frac{1}{2}}$$

$$\leq \left( \int_0^1 \left( P_{\hat{Y}|S=0}(z) - P_{\hat{Y}|S=1}(z) \right)^4 dz \right)^{\frac{1}{2}}$$

$$\leq \sqrt{TV(\hat{Y}, S)}.$$

Consequently, we obtain a variant of Theorem 1 as follows.

**Theorem A 1.** $I(\hat{Y}, S)$ *is upper bounded by* $\sqrt{TV(\hat{Y}, S)}$, *if* $\int_0^1 \left( \frac{P_0 P_1}{P_0 P_{\hat{Y}|S=0}(z) + P_1 P_{\hat{Y}|S=1}(z)} \right)^2 dz \leq 1$ *holds.*

According to Theorem A 1, we find a looser upper bound for $I(\hat{Y}, S)$ (still dependent on $TV(\hat{Y}, S)$) based on a weaker condition. In addition, Theorem A 1 is able to support our total variation loss as well, since we still have $I(\hat{Y}, S)$ approaches 0 when $TV(\hat{Y}, S)$ approaches 0 after training. Similar as the assumption $|P_{\hat{Y}|S=0}(z) - P_{\hat{Y}|S=1}(z)| \leq 1$, we conduct numerical experiments with kernel density estimation for estimating $P_{\hat{Y}|S=0}(z)$ and $P_{\hat{Y}|S=1}(z)$ and numerical integral for computing the value of $\int_0^1 \left( \frac{P_0 P_1}{P_0 P_{\hat{Y}|S=0}(z) + P_1 P_{\hat{Y}|S=1}(z)} \right)^2 dz$. We obtain the results of the integral as $0.8621 \pm 0.1110$ for Facebook, $0.4568 \pm 0.0666$ for Pokec_z, and $0.5934 \pm 0.0763$ for Credit (mean value and standard deviation under 5 random seeds). Experimental results verify the feasibility of the condition in Theorem A 1.

### A.2 PROOF OF THEOREM 2

**Theorem 2.** *The optimal poisoned adjacency matrix* $\mathbf{A}^{t+1}$ *in the* $t+1$-*th iteration given by PGD, i.e., the solution of* $\mathbf{A}^{t+1} = \operatorname{argmin}_{|\mathcal{L}(\mathbf{A}^t) - \mathcal{L}(\mathbf{A}')| \leq \epsilon_t} \|\mathbf{A}' - (\mathbf{A}^t + \eta \nabla \mathcal{L}_f(\mathbf{A}^t))\|_F^2$ *is*

$$\mathbf{A}^{t+1} = \begin{cases} \mathbf{A}^t + \eta \nabla_{\mathbf{A}} \mathcal{L}_f(\mathbf{A}^t), \text{ if } \eta |\nabla_{\mathbf{A}} \mathcal{L}(\mathbf{A}^t)^T \nabla_{\mathbf{A}} \mathcal{L}_f(\mathbf{A}^t)| \leq \epsilon_t, \\ \mathbf{A}^t + \eta \nabla_{\mathbf{A}} \mathcal{L}_f(\mathbf{A}^t) + \dfrac{e_t \epsilon_t - \eta \nabla_{\mathbf{A}} \mathcal{L}(\mathbf{A}^t)^T \nabla_{\mathbf{A}} \mathcal{L}_f(\mathbf{A}^t)}{\|\nabla_{\mathbf{A}} \mathcal{L}(\mathbf{A}^t)\|_F^2} \nabla_{\mathbf{A}} \mathcal{L}(\mathbf{A}^t), \text{ otherwise,} \end{cases} \quad (9)$$

*where* $e_t = \operatorname{sign}\left( \nabla_{\mathbf{A}} \mathcal{L}(\mathbf{A}^t)^T \nabla_{\mathbf{A}} \mathcal{L}_f(\mathbf{A}^t) \right)$.

*Proof.* First, we know that $\mathbf{A}' = \mathbf{A}^t$ is a feasible solution because $\mathcal{L}(\mathbf{A}^t) - \mathcal{L}(\mathbf{A}^t) = 0 \leq \epsilon_t$. Hence we assume that $\mathbf{A}'$ is close to $\mathbf{A}^t$. Consequently, we use the first-order Taylor expansion to substitute the constraint $|\mathcal{L}(\mathbf{A}') - \mathcal{L}(\mathbf{A}^t)| \leq \epsilon_t$ as $\left| \nabla \mathcal{L}(\mathbf{A}^t)^T (\mathbf{A}' - \mathbf{A}^t) \right| \leq \epsilon_t$. For simplicity, we vectorize the adjacency matrices $\mathbf{A}^t$ and $\mathbf{A}'$ here such that $\mathbf{A}^t, \mathbf{A}' \in \mathbb{R}^{n^2}$.

Next, we let $\mathbf{A}' = \mathbf{A}^t + \eta \nabla \mathcal{L}_f(\mathbf{A}^t) + \boldsymbol{\xi}$, and convert the optimization problem as follows.

$$\mathbf{A}^{t+1} = \operatorname*{argmin}_{|\nabla \mathcal{L}(\mathbf{A}^t)^T (\eta \nabla \mathcal{L}_f(\mathbf{A}^t) + \boldsymbol{\xi})| \leq \epsilon_t} \|\boldsymbol{\xi}\|_2^2. \quad (10)$$

Then we discuss the new constraint in Equation (10) $|\eta \nabla \mathcal{L}(\mathbf{A}^t)^T \nabla \mathcal{L}_f(\mathbf{A}^t) + \nabla \mathcal{L}(\mathbf{A}^t)^T \boldsymbol{\xi}| \leq \epsilon_t$ in different conditions.

(1). When $|\eta \nabla \mathcal{L}(\mathbf{A}^t)^T \nabla \mathcal{L}_f(\mathbf{A}^t)| \leq \epsilon_t$, we can easily obtain the optimal solution as $\boldsymbol{\xi} = \mathbf{0}$.

(2). When $\eta\nabla\mathcal{L}(\mathbf{A}^t)^T\nabla\mathcal{L}_f(\mathbf{A}^t) \geq \epsilon_t$, then we have

$$-\epsilon_t - \eta\nabla\mathcal{L}(\mathbf{A}^t)^T\nabla\mathcal{L}_f(\mathbf{A}^t) \leq \nabla\mathcal{L}(\mathbf{A}^t)^T\boldsymbol{\xi} \leq \epsilon_t - \eta\nabla\mathcal{L}(\mathbf{A}^t)^T\nabla\mathcal{L}_f(\mathbf{A}^t).$$

Because $\nabla\mathcal{L}(\mathbf{A}^t)^T\boldsymbol{\xi} = \|\nabla\mathcal{L}(\mathbf{A}^t)\|_2 \cdot \|\boldsymbol{\xi}\|_2 \cdot \cos\theta$, where $\theta$ is the angle of $\nabla\mathcal{L}(\mathbf{A}^t)$ and $\boldsymbol{\xi}$. To minimize $\|\boldsymbol{\xi}\|_2$, we minimize $\cos\theta$ as $\cos\theta = -1$, i.e., $\boldsymbol{\xi} = -\|\boldsymbol{\xi}\|_2 \cdot \nabla\mathcal{L}(\mathbf{A}^t)/\|\nabla\mathcal{L}(\mathbf{A}^t)\|_2$, and then have

$$\frac{-\epsilon_t + \eta\nabla\mathcal{L}(\mathbf{A}^t)^T\nabla\mathcal{L}_f(\mathbf{A}^t)}{\|\nabla\mathcal{L}(\mathbf{A}^t)\|_2} \leq \|\boldsymbol{\xi}\|_2 \leq \frac{\epsilon_t + \eta\nabla\mathcal{L}(\mathbf{A}^t)^T\nabla\mathcal{L}_f(\mathbf{A}^t)}{\|\nabla\mathcal{L}(\mathbf{A}^t)\|_2}.$$

Therefore, the solution of Equation (10) is

$$\boldsymbol{\xi} = \frac{\epsilon_t - \eta\nabla\mathcal{L}(\mathbf{A}^t)^T\nabla\mathcal{L}_f(\mathbf{A}^t)}{\|\nabla\mathcal{L}(\mathbf{A}^t)\|_2^2}\nabla\mathcal{L}(\mathbf{A}^t).$$

(3). When $\eta\nabla\mathcal{L}(\mathbf{A}^t)^T\nabla\mathcal{L}_f(\mathbf{A}^t) \leq -\epsilon_t$, we also have

$$-\epsilon_t - \eta\nabla\mathcal{L}(\mathbf{A}^t)^T\nabla\mathcal{L}_f(\mathbf{A}^t) \leq \nabla\mathcal{L}(\mathbf{A}^t)^T\boldsymbol{\xi} \leq \epsilon_t - \eta\nabla\mathcal{L}(\mathbf{A}^t)^T\nabla\mathcal{L}_f(\mathbf{A}^t).$$

Different from condition (2), the left-hand side and right-hand side here are both positive. Similarly, we let $\cos\theta = 1$ and obtain the solution of Equation (10) as

$$\boldsymbol{\xi} = \frac{-\epsilon_t - \eta\nabla\mathcal{L}(\mathbf{A}^t)^T\nabla\mathcal{L}_f(\mathbf{A}^t)}{\|\nabla\mathcal{L}(\mathbf{A}^t)\|_2^2}\nabla\mathcal{L}(\mathbf{A}^t).$$

Combine the aforementioned three conditions into $\mathbf{A}' = \mathbf{A}^t + \eta\nabla\mathcal{L}_f(\mathbf{A}^t) + \boldsymbol{\xi}$, then we have the solution as follows

$$\mathbf{A}^{t+1} = \begin{cases} \mathbf{A}^t + \eta\nabla_{\mathbf{A}}\mathcal{L}_f(\mathbf{A}^t), \textbf{ if } \eta|\nabla_{\mathbf{A}}\mathcal{L}(\mathbf{A}^t)^T\nabla_{\mathbf{A}}\mathcal{L}_f(\mathbf{A}^t)| \leq \epsilon_t, \\ \mathbf{A}^t + \eta\nabla_{\mathbf{A}}\mathcal{L}_f(\mathbf{A}^t) + \dfrac{e_t\epsilon_t - \eta\nabla_{\mathbf{A}}\mathcal{L}(\mathbf{A}^t)^T\nabla_{\mathbf{A}}\mathcal{L}_f(\mathbf{A}^t)}{\|\nabla_{\mathbf{A}}\mathcal{L}(\mathbf{A}^t)\|_F^2}\nabla_{\mathbf{A}}\mathcal{L}(\mathbf{A}^t), \text{ otherwise,} \end{cases}$$

where $e_t = \text{sign}\left(\nabla_{\mathbf{A}}\mathcal{L}(\mathbf{A}^t)^T\nabla_{\mathbf{A}}\mathcal{L}_f(\mathbf{A}^t)\right)$. $\square$

# B   ATTACK SETTINGS

In this section, we introduce our attack settings in detail from three perspectives, the attacker's goal, the attacker's knowledge, and the attacker's capability.

**Attacker's Goal.**   There are two different settings of our problem, the fairness evasion attack, and the fairness poisoning attack. In the fairness evasion attack, the attacker's goal is to let the victim model make unfair predictions on test nodes, where the victim model is trained with fairness consideration on the clean graph. Note that it is possible for real-world attackers to attack the access control of the databases to modify the input graph data, especially for edge computing systems with a coarse-grained access control (Ali et al., 2016; Xiao et al., 2019). In addition, once the model is deployed, the attacker can launch evasion attacks *at any time*, which increases the difficulty and cost of defending against evasion attacks (Zhang et al., 2022). Considering the severe impact of evasion attacks, many prevalent existing works Dai et al. (2018); Zügner et al. (2018); Zügner & Günnemann (2019); Zhang et al. (2022) make great efforts to study evasion attacks. In the fairness poisoning attack, the attacker's goal is to let the victim model make unfair predictions on test nodes, where the victim model is trained with fairness consideration on the poisoned graph. For both settings, we use commonly used fairness metrics, e.g., demographic parity (Dwork et al., 2012) and equal opportunity (Hardt et al., 2016) to measure the fairness of predictions.

**Attacker's Knowledge.**   To make our attack practical in the real world, we set several limitations on the attacker's knowledge and formulate the attack within a gray-box setting. It is worth noting that our attacker's knowledge basically follows previous attacks on prediction utility of GNNs (Wu et al., 2019; Xu et al., 2019; Zügner & Günnemann, 2019; Chang et al., 2020a; Ma et al., 2020; Li et al., 2022a; Ma et al., 2022; Lin et al., 2022). Specifically, the attacker is able to observe the node attributes $\mathbf{X}$, graph structure $\mathbf{A}$, ground truth labels $\mathcal{Y}$, and sensitive attribute value set $\mathcal{S}$, but cannot

observe the victim GNN model $f_{\boldsymbol{\theta}}$. Therefore, the attackers need to exploit a surrogate model $g_{\boldsymbol{\theta}}$ to achieve their goal.

It is worth noting that our method can be *directly* adapted to a white-box setting by replacing the trained surrogate model in the attacker's objective with the true victim model in Problem 1. In contrast, designing fairness attacks in a black-box attack setting can be extremely challenging. The difference between gray-box attacks and black-box attacks is black-box attackers are not allowed to access the ground truth labels. Different from node embeddings which can be obtained in an unsupervised way, group fairness metrics have to rely on the ground truth labels, which makes existing black-box attacks on graphs difficult to adapt to fairness attacks. Despite the difficulty of black-box fairness attacks, we provide an initial step toward a potential way to extend our framework to a black-box setting. First, the attacker can collect some data following a similar distribution, i.e., if the original graph is a Citeseer citation network, the attacker can collect data from Arxiv; if the original graph is a Facebook social network, the attacker can collect data from Twitter (X). During the data collection (preprocessing), the dimension of collected node features should be aligned with the original graph. Then, the attacker can train a state-of-the-art inductive GNN model on the collected graph data and obtain the predicted labels on the original graph. Finally, the attacker can use the predicted labels as a pseudo label to implement our G-FairAttack on the original graph.

**Attacker's Capability.** Between attacking the graph structure and the node attributes, we only consider the structure attack as the structure perturbation in the discrete domain is more challenging to solve and the structure attack can be easily adapted to obtain the attribute attack. Hence, we consider that attackers can only modify the graph structure $\mathbf{A}$, i.e., adding new edges or cutting existing edges, consistent with many previous attacks on prediction utility of GNNs (Dai et al., 2018; Xu et al., 2019; Zügner & Günnemann, 2019; Wang & Gong, 2019; Bojchevski & Günnemann, 2019; Chang et al., 2020a). In addition, the structure perturbation should be unnoticeable. Existing attacks of GNNs (Zügner et al., 2018; Zügner & Günnemann, 2019; Dai et al., 2018; Bojchevski & Günnemann, 2019) proposed the following unnoticeable constraints: $\Delta$ edges are changed at most; there are no singleton nodes, i.e., nodes without neighbors after the attack; the degree distributions before and after the attack should be the same with high confidence. We follow these works to ensure the unnoticeability of our attack. More importantly, we propose an extra unnoticeable constraint to ensure the difference of the utility losses before and after the attack is less than $\epsilon$. This unnoticeable utility constraint makes attacks on fairness difficult to recognize.

After clarifying detailed attack settings, we use Figure 1 to illustrate a toy example of our proposed attack problem. In Figure 1, We use squares to denote the sensitive group 0 and triangles to denote the sensitive group 1. We use blue to label class-0 nodes and orange to label class-1 nodes. We compute the demographic parity and the equal opportunity metrics (larger value means less fair) to evaluate the fairness of the model prediction. By modifying two edges (from left to right), the attacker can let the fairness-aware GNN make unfair predictions while preserving the accuracy.

## C  FAST COMPUTATION

### C.1  FAST COMPUTATION SKETCH

The goal of fast computation is to solve the problem $(u^t, v^t) \leftarrow \arg\max_{(u,v) \in \mathcal{C}^t} \tilde{r}^t(u,v)$ efficiently. According to the definition of $\tilde{r}^t(u,v)$, the computation of $\tilde{r}^t(u,v)$ depends on two loss functions: the attacker's objective $\mathcal{L}_f$ and the utility loss $\mathcal{L}$. Between them, $\mathcal{L}_f$ can be formulated as

$$\mathcal{L}_f\left(g_{\boldsymbol{\theta}^*}, \mathbf{A}, \mathbf{X}, \mathcal{Y}, \mathcal{V}_{\text{test}}, \mathcal{S}\right) = \left| \sum_{i \in \mathcal{V}_{\text{test}}} k_i \mathbb{I}_{\geq 0}\left(g_{\boldsymbol{\theta}^*}\left(\mathbf{A}, \mathbf{X}\right)_{[i]}\right) \right|, \quad (11)$$

where $k_i = 1/\left|\mathcal{V}_0 \cap \mathcal{V}_{\text{test}}\right|$ if $i \in \mathcal{V}_0$ and $k_i = -1/\left|\mathcal{V}_1 \cap \mathcal{V}_{\text{test}}\right|$ if $i \in \mathcal{V}_1$. $\mathbb{I}_{\geq 0}(\cdot)$ denotes an indicator function where $\mathbb{I}_{\geq 0}(x) = 1$ if $x \geq 0$ and $\mathbb{I}_{\geq 0}(x) = 0$ otherwise. The other $\mathcal{L}$ can be formulated as

$$\mathcal{L}(g_{\boldsymbol{\theta}^*}, \mathbf{A}, \mathbf{X}, \mathcal{Y}, \mathcal{V}_{\text{train}}) = -\frac{1}{|\mathcal{V}_{\text{train}}|} \sum_{i \in \mathcal{V}_{\text{train}}} y_i \log\left(\sigma(g_{\boldsymbol{\theta}^*}(\mathbf{A}, \mathbf{X})_{[i]})\right)$$
$$+ (1 - y_i) \log\left(1 - \sigma(g_{\boldsymbol{\theta}^*}(\mathbf{A}, \mathbf{X})_{[i]})\right), \quad (12)$$

---

**Algorithm 1** G-FairAttack: A Sequential Attack on Fairness of GNNs.

---
**Input:** Clean adjacency matrix $\mathbf{A}$, attribute matrix $\mathbf{X}$, attack budget $\Delta$, utility budget $\epsilon$.
**Output:** The solution of Problem 1: $\mathbf{A}^*$.
1: $t \leftarrow 0, \mathcal{C}^0 \leftarrow \mathcal{E}, \mathbf{A}^0 \leftarrow \mathbf{A}$
2: $\boldsymbol{\theta}^0 \leftarrow \arg\min_{\boldsymbol{\theta}} \mathcal{L}_s\left(g_{\boldsymbol{\theta}}, \mathbf{A}, \mathbf{X}, \mathcal{Y}, \mathcal{S}\right)$
3: **while** $t \leq \Delta$ and $\left|\mathcal{L}(g_{\boldsymbol{\theta}^t}, \mathbf{A}^t) - \mathcal{L}(g_{\boldsymbol{\theta}^0}, \mathbf{A}^0)\right| \leq \epsilon$ **do**
4: $\quad (u^t, v^t) \leftarrow \arg\max_{(u,v) \in \mathcal{C}^t} \tilde{r}^t(u, v)$, according to Algorithm 2.
5: $\quad \mathbf{A}^{t+1} \leftarrow flip_{(u^t, v^t)} \mathbf{A}^t$
6: $\quad \boldsymbol{\theta}^{t+1} \leftarrow \begin{cases} \boldsymbol{\theta}^t, & \textbf{Evasion} \\ \arg\min_{\boldsymbol{\theta}} \mathcal{L}_s\left(g_{\boldsymbol{\theta}}, \mathbf{A}^{t+1}, \mathbf{X}, \mathcal{Y}, \mathcal{S}\right), & \textbf{Poisoning} \end{cases}$
7: $\quad \mathcal{C}^{t+1} \leftarrow \mathcal{C}^t \backslash \{(u^t, v^t), (v^t, u^t)\}$
8: $\quad t \leftarrow t + 1$
9: **end while**
10: $\mathbf{A}^* \leftarrow \mathbf{A}^t$

---

where $\sigma(\cdot)$ denotes the sigmoid function. In the $t$-th iteration of our sequential attack, to compute the score function $\tilde{r}^t(u, v)$ efficiently, we should reduce the complexity of computing both $\Delta\mathcal{L}^t(u, v)$ and $\Delta\mathcal{L}_f^t(u, v)$. Specifically, our solution is to compute $flip_{(u,v)}\mathbf{Z}^t$ incrementally and obtain $g_{\boldsymbol{\theta}^t}(flip_{(u,v)}\mathbf{A}^t, \mathbf{X}) = flip_{(u,v)}\mathbf{Z}^t\boldsymbol{\theta}^t$. Then we can compute $\Delta\mathcal{L}_f(u, v)$ and $\Delta\mathcal{L}(u, v)$ according to Equation (11) and Equation (12). We first review the computation of $flip_{(u,v)}\mathbf{Z}^t$ when *a new edge $(u, v)$ is added*.

**Case 1**: If $i \in \{u, v\}$, we can compute $flip_{(u,v)}\mathbf{Z}^t_{[i,:]}$ as

$$flip_{(u,v)}\mathbf{Z}^t_{[i,:]} = \frac{\hat{\boldsymbol{d}}^t_{[i]}}{\hat{\boldsymbol{d}}^t_{[i]} + 1}(\mathbf{Z}^t_{[i,:]} - \frac{\hat{\mathbf{A}}^t_{[i,:]}\mathbf{X}}{(\hat{\boldsymbol{d}}^t_{[i]})^2}) + \frac{\hat{\mathbf{A}}^t_{[i,:]}\mathbf{X} + \mathbf{X}_{[j,:]}}{(\hat{\boldsymbol{d}}^t_{[i]} + 1)^2} + \frac{\hat{\mathbf{A}}^t_{[j,:]}\mathbf{X} + \mathbf{X}_{[i,:]}}{(\hat{\boldsymbol{d}}^t_{[i]} + 1)(\hat{\boldsymbol{d}}^t_{[j]} + 1)}, \quad (13)$$

where $j = v$ if $i = u$ and $j = u$ otherwise;

**Case 2**: If $i \in \mathcal{N}_u^t \cup \mathcal{N}_v^t \backslash \{u, v\}$, we can compute $flip_{(u,v)}\mathbf{Z}^t_{[i,:]}$ as

$$flip_{(u,v)}\mathbf{Z}^t_{[i,:]} = \mathbf{Z}^t_{[i,:]} - \mathbb{I}_{i \in \mathcal{N}_u^t} \cdot (\frac{\hat{\mathbf{A}}^t_{[u,:]}\mathbf{X}}{\hat{\boldsymbol{d}}^t_{[i]}\hat{\boldsymbol{d}}^t_{[u]}} - \frac{\hat{\mathbf{A}}^t_{[u,:]}\mathbf{X} + \mathbf{X}_{[v,:]}}{\hat{\boldsymbol{d}}^t_{[i]}(\hat{\boldsymbol{d}}^t_{[u]} + 1)}) - \mathbb{I}_{i \in \mathcal{N}_v^t} \cdot (\frac{\hat{\mathbf{A}}^t_{[v,:]}\mathbf{X}}{\hat{\boldsymbol{d}}^t_{[i]}\hat{\boldsymbol{d}}^t_{[v]}} - \frac{\hat{\mathbf{A}}^t_{[v,:]}\mathbf{X} + \mathbf{X}_{[u,:]}}{\hat{\boldsymbol{d}}^t_{[i]}(\hat{\boldsymbol{d}}^t_{[v]} + 1)}),$$
$$(14)$$

where $\mathbb{I}_{i \in \mathcal{N}} = 1$ if $i \in \mathcal{N}$, and $\mathbb{I}_{i \in \mathcal{N}} = 0$ otherwise;

**Case 3**: If $i \notin \mathcal{N}_u^t \cup \mathcal{N}_v^t$, we have $flip_{(u,v)}\mathbf{Z}^t_{[i,:]} = \mathbf{Z}^t_{[i,:]}$.

Next, we introduce the computation of $flip_{(u,v)}\mathbf{Z}^t$ when *an existing edge $(u, v)$ is removed*. Similarly, we divide the computation into three cases as follows.

**Case 1**: If $i \in \{u, v\}$, we can compute $flip_{(u,v)}\mathbf{Z}^t_{[i,:]}$ as

$$flip_{(u,v)}\mathbf{Z}^t_{[i,:]} = \frac{\hat{\boldsymbol{d}}^t_{[i]}}{\hat{\boldsymbol{d}}^t_{[i]} - 1}(\mathbf{Z}^t_{[i,:]} - \frac{\hat{\mathbf{A}}^t_{[i,:]}\mathbf{X}}{(\hat{\boldsymbol{d}}^t_{[i]})^2} - \frac{\hat{\mathbf{A}}^t_{[j,:]}\mathbf{X}}{\hat{\boldsymbol{d}}^t_{[i]}\hat{\boldsymbol{d}}^t_{[j]}}) + \frac{\hat{\mathbf{A}}^t_{[i,:]}\mathbf{X} - \mathbf{X}_{[j,:]}}{(\hat{\boldsymbol{d}}^t_{[i]} - 1)^2}, \quad (15)$$

where $j = v$ if $i = u$ and $j = u$ otherwise;

**Case 2**: If $i \in \mathcal{N}_u^t \cup \mathcal{N}_v^t \backslash \{u, v\}$, we can compute $flip_{(u,v)}\mathbf{Z}^t_{[i,:]}$ as

$$flip_{(u,v)}\mathbf{Z}^t_{[i,:]} = \mathbf{Z}^t_{[i,:]} - \mathbb{I}_{i \in \mathcal{N}_u^t} \cdot (\frac{\hat{\mathbf{A}}^t_{[u,:]}\mathbf{X}}{\hat{\boldsymbol{d}}^t_{[i]}\hat{\boldsymbol{d}}^t_{[u]}} - \frac{\hat{\mathbf{A}}^t_{[u,:]}\mathbf{X} - \mathbf{X}_{[v,:]}}{\hat{\boldsymbol{d}}^t_{[i]}(\hat{\boldsymbol{d}}^t_{[u]} - 1)}) - \mathbb{I}_{i \in \mathcal{N}_v^t} \cdot (\frac{\hat{\mathbf{A}}^t_{[v,:]}\mathbf{X}}{\hat{\boldsymbol{d}}^t_{[i]}\hat{\boldsymbol{d}}^t_{[v]}} - \frac{\hat{\mathbf{A}}^t_{[v,:]}\mathbf{X} - \mathbf{X}_{[u,:]}}{\hat{\boldsymbol{d}}^t_{[i]}(\hat{\boldsymbol{d}}^t_{[v]} - 1)}),$$
$$(16)$$

where $\mathbb{I}_{i \in \mathcal{N}} = 1$ if $i \in \mathcal{N}$, and $\mathbb{I}_{i \in \mathcal{N}} = 0$ otherwise;

**Case 3**: If $i \notin \mathcal{N}_u^t \cup \mathcal{N}_v^t$, we have $flip_{(u,v)}\mathbf{Z}^t_{[i,:]} = \mathbf{Z}^t_{[i,:]}$.

The overall fast computation algorithm is shown in Algorithm 2, where $\arg\max_{@a} \rho^t(u, v)$ is denoted as the set of $(u, v)$ corresponding to the top-$a$ elements of $\rho^t(u, v)$.

---

**Algorithm 2** The fast computation algorithm of $(u^t, v^t)$.

---

**Input:** Adjacency matrix $\mathbf{A}^t$, attribute matrix $\mathbf{X}$, output matrix $\mathbf{Z}^t$, degree vector $\hat{\boldsymbol{d}}^t$, product matrix $\hat{\mathbf{A}}^t \mathbf{X}$, model parameter $\boldsymbol{\theta}^t$.
**Output:** Target edge $(u^t, v^t)$ in Algorithm 1.
1: $\mathcal{C}^t \leftarrow \arg\max_{\substack{@_a \\ (u,v) \in \mathcal{E}^t}} \rho^t(u,v)$.
2: $k \leftarrow 0$, $\boldsymbol{p}^t \leftarrow \mathbf{0}$, $\boldsymbol{q}^t \leftarrow \mathbf{0}$.
3: **for** $(u,v) \in \mathbf{C}^t$ **do**
4:   $flip_{(u,v)}\mathbf{Z}^t \leftarrow \mathbf{Z}^t$
5:   **for** $i \in \mathcal{N}_u^t \cup \mathcal{N}_v^t$ **do**
6:    **if** $i \in \{u, v\}$ **then**
7:     Update $flip_{(u,v)}\mathbf{Z}_{[i,:]}^t$ according to Equation (13) or Equation (15).
8:    **else**
9:     Update $flip_{(u,v)}\mathbf{Z}_{[i,:]}^t$ according to Equation (14) or Equation (16).
10:    **end if**
11:   **end for**
12:   $g_{\boldsymbol{\theta}^t}(flip_{(u,v)}\mathbf{A}^t, \mathbf{X}) \leftarrow flip_{(u,v)}\mathbf{Z}^t \boldsymbol{\theta}^t$
13:   Compute $\mathcal{L}(flip_{(u,v)}\mathbf{A}^t, \mathbf{X})$ and $\mathcal{L}_f(flip_{(u,v)}\mathbf{A}^t, \mathbf{X})$ according to Equation (11) and Equation (12).
14:   $\boldsymbol{p}_{[k]}^t \leftarrow \mathcal{L}(flip_{(u,v)}\mathbf{A}^t, \mathbf{X}) - \mathcal{L}(\mathbf{A}^t, \mathbf{X})$
15:   $\boldsymbol{q}_{[k]}^t \leftarrow \mathcal{L}_f(flip_{(u,v)}\mathbf{A}^t, \mathbf{X}) - \mathcal{L}_f(\mathbf{A}^t, \mathbf{X})$
16:   $k \leftarrow k + 1$
17: **end for**
18: $\tilde{\boldsymbol{r}}^t \leftarrow \boldsymbol{q}^t - \frac{(\boldsymbol{p}^t)^T \boldsymbol{q}^t}{\|\boldsymbol{p}^t\|_2^2} \boldsymbol{p}^t$
19: $i_{\max} \leftarrow \arg\max_{i=0,\dots,|\mathcal{C}^t|-1} \tilde{\boldsymbol{r}}_{[i]}^t$
20: $(u^t, v^t) \leftarrow \mathcal{C}_{[i_{\max}]}^t$

---

## C.2 Complexity Analysis

Based on Algorithm 2, we provide a detailed complexity analysis for our proposed G-FairAttack.

**Proposition 2.** *The overall time complexity of G-FairAttack with the fast computation is $O(\bar{d}n^2 + d_x an)$, where $\bar{d}$ denotes the average degree.*

*Proof.* First, to compute $\mathcal{C}^t$, we compute $|\mathbf{Z}^t \boldsymbol{\theta}^t|$ and find the maximum $M_t$ in $O(d_x n)$. Then we compute $\rho^t(u,v) = \sum_{i \in \mathcal{N}_u^t \cup \mathcal{N}_v^t} M_t - |\mathbf{Z}_{[i,:]}^t \boldsymbol{\theta}^t|$ for $(u,v) \in \mathcal{E}^t$ in $O(\bar{d}n^2)$, and find the top-$a$ elements as $\mathcal{C}^t$ in $O(n \log a)$. Then, the computation of $\boldsymbol{p}^t$ and $\boldsymbol{q}^t$ can be divided into the following steps for each edge $(u,v) \in \mathcal{C}^t$.

1. The computation of $flip_{(u,v)}\mathbf{Z}_{[i,:]}^t$ for $i \in \mathcal{N}_u^t \cup \mathcal{N}_v^t$. We can store and update $\hat{\mathbf{A}}^t \mathbf{X}$, $\hat{\boldsymbol{d}}^t$, and $\mathbf{Z}^t$ for each iteration. Hence, the computation of Equation (13), Equation (14), Equation (15), and Equation (16) only requires $O(1)$. Consequently, the total time complexity of this step is $O(\bar{d})$.

2. The computation of $g_{\boldsymbol{\theta}^t}(flip_{(u,v)}\mathbf{A}^t, \mathbf{X}) = flip_{(u,v)}\mathbf{Z}^t \boldsymbol{\theta}^t$ requires $O(nd_x)$.

3. According to Equation (11) and Equation (12), the computation of the loss functions $\mathcal{L}(flip_{(u,v)}\mathbf{A}^t, \mathbf{X})$ and $\mathcal{L}_f(flip_{(u,v)}\mathbf{A}^t, \mathbf{X})$ based on $g_{\boldsymbol{\theta}^t}(flip_{(u,v)}\mathbf{A}^t, \mathbf{X})$ requires $O(n)$.

Considering all edges from $\mathcal{C}^t$, the computation of $\boldsymbol{p}^t$ and $\boldsymbol{q}^t$ requires $O(d_x an)$. Finally, we can compute $\tilde{\boldsymbol{r}}^t$ and find $(u^t, v^t)$ in $O(a)$. Combining all these steps, the complexity of Algorithm 2 is $O(\bar{d}n^2 + d_x an)$ in total. $\square$

With our fast computation method, the total time complexity of G-FairAttack is $O((\bar{d}n^2 + d_x an)\Delta)$. Here, the retraining of the surrogate model $\boldsymbol{\theta}^{t+1} = \arg\min_{\boldsymbol{\theta}} \mathcal{L}_s(g_{\boldsymbol{\theta}}, \mathbf{A}^{t+1}, \mathbf{X}, \mathcal{Y}, \mathcal{S})$ in the fairness poisoning attack is neglected because the convergence is not controllable. The overall G-FairAttack

algorithm is shown in Algorithm 1. We can obtain the space complexity of G-FairAttack as $O(|\mathcal{E}| + d_x n)$.

Next, we show that our complexity can be further reduced in practice. We list two potential ways as follows.

1. The first approach is to implement our fast computation algorithm in parallel. As the proof of Proposition 2 shows, the main part of G-FairAttack's complexity is the computation of $\rho^t(u,v)$ for $(u,v) \in \mathcal{E}^t$, which has $O(\bar{d}n^2)$ complexity. It is distinct that this computation can be implemented in parallel where $\mathcal{E}^t$ is partitioned into $p$ subsets, and each subset is fed into one process. By exploiting parallel computation, the overall complexity can be reduced to $O(\bar{d}n^2/p)$.

2. Instead of ranking all of the edges in $\mathcal{E}^t$ by $\rho^t(u,v)$, we can just randomly sample $a$ edges from $\mathcal{E}^t$ as $C^t$. By using random sampling instead of ranking, the overall complexity can be reduced to $O(d_x a n)$. However, the error of the fast computation might increase without a careful choice of the sampling distribution.

Note that most existing adversarial attack approaches (Zügner et al., 2018; Zügner & Günnemann, 2019; Bojchevski & Günnemann, 2019; Lin et al., 2022) do not have a lower complexity (less than $O(n^2)$) than our proposed G-FairAttack. The adversarial attacks with an $O(n^2)$ complexity can already fit most commonly used graph datasets. In general, our method is practical for most graph datasets as the existing literature. For extremely large datasets, we can also use the aforementioned strategies to reduce further the time complexity.

Finally, we would like to make a more detailed comparison of the time complexity with adopted attacking baselines. The random sampling-based baselines, i.e., random and FA-GNN, definitely have lower time complexity ($O(\Delta)$, $\Delta$ is the attack budget) because they are based on random sampling. Although their complexity is low, the effectiveness of random sampling-based attacks is very limited. For the rest gradient-based attack baselines, i.e., Gradient Ascent and Metattack, their time complexities are $O(n^2)$ (Zügner & Günnemann, 2019), the same as G-FairAttack. However, gradient-based methods have a larger space complexity compared with G-FairAttack ($O(n^2)$ vs. $O(d_x n + |\mathcal{E}|)$).

## D  G-FAIRATTACK VS GRADIENT-BASED METHODS

The fairness attack of GNNs is a discrete optimization problem, which is highly challenging to solve. Most of the existing adversarial attacks that focus on the prediction utility of GNNs adopt gradient-based methods (Zügner & Günnemann, 2019; Wu et al., 2019; Xu et al., 2019; Geisler et al., 2021) to find the maximum of the attacker's objective. As G-FairAttack, gradient-based structure attacks flip the edges sequentially. In the $t$-th iteration ($t = 1, 2, \dots$), the gradient-based optimization algorithm finds a target edge $(u^t, v^t)$ based on the gradient of the adjacency matrix $\nabla_{\mathbf{A}^t}\mathcal{L}$ where $\mathcal{L}$ is the objective function of the attacker, and flips this target edge to obtain the update adjacency matrix $\mathbf{A}^{t+1}$, which is expected to increase the attacker's objective. In particular, to obtain $\nabla_{\mathbf{A}^t}\mathcal{L}$, gradient-based methods extend the discrete adjacency matrix $\mathbf{A}^t \in \{0,1\}^{n \times n}$ to a continuous domain $\mathbb{R}^{n \times n}$ and compute $\nabla_{\mathbf{A}^t}\mathcal{L} \in \mathbb{R}^{n \times n}$. Specifically, for poisoning attacks where the problem becomes a bilevel opti-

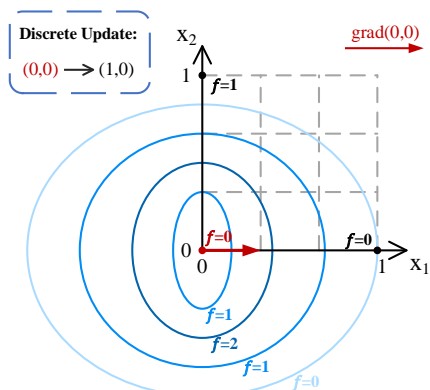

Figure 4: The limitation of the gradient-based optimization method. The blue ellipses are isolines of the loss function.

mization, existing methods exploit the meta-learning (Zügner & Günnemann, 2019) or the convex relaxation (Xu et al., 2019) techniques to remove the inner optimization. After obtaining $\nabla_{\mathbf{A}^t}\mathcal{L}$, the gradient-based methods should update $\mathbf{A}^t$ in the discrete domain instead of using gradient ascent directly. Specifically, gradient-based methods choose the target edge *corresponding to the largest element* or use random sampling to find a target element $\mathbf{A}^t_{[u,v]}$.

Although existing attacks of GNNs based on gradient-based methods successfully decrease the prediction utility of victim models, they have two main limitations.

**The first limitation** is that we cannot ensure the loss function after flipping the target edge will increase because the update in the discrete domain brings an uncontrollable error. The intuition of the gradient-based method is that we cannot update the adjacency matrix with gradient ascent because the adjacency matrix is binary and only one edge can be flipped in each time. Hence, we expect that flipping the target edge corresponding to the largest gradient component can lead to the largest increment of the attacker's objective $\mathcal{L}(\mathbf{A}, \mathbf{X})$. However, this expectation can be false since the update of the adjacency matrix (flipping one target edge) has a fixed length. Next, we prove Proposition 1.

**Proposition 1.** *Gradient-based methods for optimizing the graph structure are not guaranteed to decrease the objective function.*

*Proof.* Consider the loss function $\mathcal{L}(\mathbf{A}, \mathbf{X})$ near a specific point $\mathbf{A}_0$ where $\mathbf{A} \in \mathbb{R}^{n^2}$ is a vectorized adjacency matrix. Based on Taylor's Theorem, we have

$$\mathcal{L}(\mathbf{A}, \mathbf{X}) = \mathcal{L}(\mathbf{A}_0, \mathbf{X}) + \nabla_{\mathbf{A}} \mathcal{L}(\mathbf{A}_0, \mathbf{X})^{\top}(\mathbf{A} - \mathbf{A}_0) + R_1(\mathbf{A}),$$

where $R_1(\mathbf{A}) = h_1(\mathbf{A})\|\mathbf{A} - \mathbf{A}_0\|$ is the Peano remainder and we have $\lim_{\mathbf{A} \to \mathbf{A}_0} h_1(\mathbf{A}) = 0$. For gradient-based methods, we have $\mathbf{A}_1 = \mathbf{A}_0 + \mathbf{e}_k$ where $\mathbf{e}_k$ is the basis vector at the $k$-th dimension. Then, we have

$$\mathcal{L}(\mathbf{A}_1, \mathbf{X}) = \mathcal{L}(\mathbf{A}_0, \mathbf{X}) + \nabla_{\mathbf{A}} \mathcal{L}(\mathbf{A}_0, \mathbf{X})[k] + h_1(\mathbf{A}_1).$$

Here, we know that $\nabla_{\mathbf{A}} \mathcal{L}(\mathbf{A}_0, \mathbf{X})[k]$ is the largest positive element of $\nabla_{\mathbf{A}} \mathcal{L}(\mathbf{A}_0, \mathbf{X})$, which is a fixed value. Then, we expect that choosing $\mathbf{A}_1$ can lead to the fact that $\mathcal{L}(\mathbf{A}_1, \mathbf{X}) > \mathcal{L}(\mathbf{A}_0, \mathbf{X})$, i.e., $\nabla_{\mathbf{A}} \mathcal{L}(\mathbf{A}_0, \mathbf{X})[k] + h_1(\mathbf{A}_1) > 0$. However, this inequality is not true without further assumptions when $\|\mathbf{A}_1 - \mathbf{A}_0\|_0 = 1$. $\qquad\square$

In comparison, we also show that the error can be controlled in the continuous domain by a careful selection of the learning rate. In the continuous domain, the situation is different because we can make the value of $\|\mathbf{A}_1 - \mathbf{A}_0\|$ arbitrarily small by tuning the learning rate $\eta$ where $\mathbf{A}_1 = \mathbf{A}_0 + \eta \nabla_{\mathbf{A}} \mathcal{L}(\mathbf{A}_0, \mathbf{X})[k]\mathbf{e}_k$. Note that we have $\lim_{\mathbf{A} \to \mathbf{A}_0} h_1(\mathbf{A}) = 0$ and the value of $\nabla_{\mathbf{A}} \mathcal{L}(\mathbf{A}_0, \mathbf{X})[k]$ is fixed. Hence we can choose a proper $\eta$ which makes $\|\mathbf{A}_1 - \mathbf{A}_0\|$ small enough to ensure $|h_1(\mathbf{A}_1)| < \nabla_{\mathbf{A}} \mathcal{L}(\mathbf{A}_0, \mathbf{X})[k]$. Finally, we have $\nabla_{\mathbf{A}} \mathcal{L}(\mathbf{A}_0, \mathbf{X})[k] + h_1(\mathbf{A}_1) > 0$ and $\mathcal{L}(\mathbf{A}_1, \mathbf{X}) > \mathcal{L}(\mathbf{A}_0, \mathbf{X})$ consequently. We also provide a two-dimensional case in Figure 4. In the iteration, the optimization starts at $(0, 0)$. The gradient at $(0, 0)$ is $[1 \; 0]^{\top}$. According to the gradient-based methods, the next point should be at $(1, 0)$. However, the loss after updating does not increase $f(1, 0) = f(0, 0)$. Instead, $(0, 1)$ is a better update for $f(0, 1) > f(0, 0)$ in this iteration.

**The second limitation** is the large space complexity. The computation of $\nabla_{\mathbf{A}^t} \mathcal{L}$ requires storing a dense adjacency matrix with $O(n^2)$ space complexity, which is costly at a large scale.

In contrast, our proposed G-FairAttack successfully addresses these limitations. First, G-FairAttack can ensure the increase of the attacker's objective after flipping the target edge since G-FairAttack exploits a ranking-based method to choose the target edge in each iteration. In particular, we choose the target edge as $\max_{(u,v) \in \mathcal{C}^t} r^t(u, v) = \mathcal{L}_f(g_{\boldsymbol{\theta}^t}, flip_{(u,v)}\mathbf{A}^t) - \mathcal{L}_f(g_{\boldsymbol{\theta}^t}, \mathbf{A}^t)$, which ensures that flipping the target edge can maximize the increment of attacker's objective. Second, unlike gradient-based methods, G-FairAttack does not require storing a dense adjacency matrix as no gradient computations are involved. Referring to the discussion in Appendix C.2, the space complexity of G-FairAttack is $O(|\mathcal{E}| + d_x n)$, much lower than gradient-based methods ($O(n^2)$). Moreover, we conduct an experiment to demonstrate the superiority of G-FairAttack compared with gradient-based optimization methods. Specifically, we compare the effectiveness of G-FairAttack with two gradient-based optimization methods in both fairness evasion attack and fairness poisoning attack settings. If an optimization algorithm leads to a larger increase in the attacker's objective, it is more effective. For the fairness evasion attack, we choose Gradient Ascent as the baseline method. Given the attack budget $\Delta = 0.5\%|\mathcal{E}|$, we record the value of the attacker objective $\mathcal{L}_f(g_{\boldsymbol{\theta}^*}, \mathbf{A}^t)$, i.e., demographic parity based on a surrogate model in the $t$-th iteration, for $t = 0, \ldots, \Delta$. The result is shown in Figure 5(a). We observe that the attacker objective keeps increasing during our non-gradient optimization (adopted by G-FairAttack) process and reaches an optimum rapidly, while the gradient-based optimization method (adopted by Gradient Ascent) cannot ensure the increment of the attacker objective during the optimization.

Table 3: Dataset statistics.

| Dataset | #Nodes | #Edges | #Attributes | #Train/% | #Validation/% | #Test/% | Sensitive |
|---------|--------|--------|-------------|----------|---------------|---------|-----------|
| Facebook | 1,045 | 53,498 | 574 | 50% | 20% | 30% | Gender |
| Pokec | 7,659 | 41,100 | 277 | 13% | 25% | 25% | Region |
| Credit | 30,000 | 200,526 | 13 | 20% | 20% | 30% | Age |

For the fairness poisoning attack, we choose Metattack as the baseline method. During the optimization process, the surrogate model $g_{\theta_t}$ is retrained in each iteration. The convergence of the surrogate model during the retraining is not controllable. To make a fair comparison, we should ensure both optimization methods compute the attacker objective based on the same surrogate model. Hence we fix $\mathbf{A}^t$ and $g_{\theta^t}$ for both methods and record the increment of the attacker objective

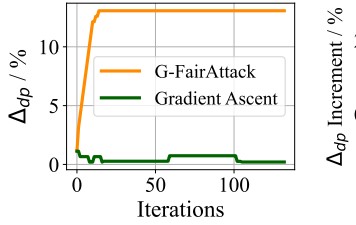 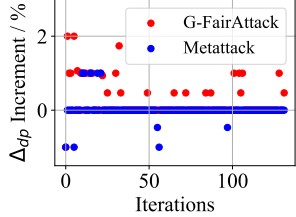

(a) Fairness evasion attack.  (b) Fairness poisoning attack.

Figure 5: The variation of attacker's objective during the optimization process on Facebook, comparing non-gradient methods (G-FairAttack) with gradient-based methods (Gradient Ascent, Metattack).

$\mathcal{L}_f(g_{\theta^t}, \mathbf{A}^{t+1}) - \mathcal{L}_f(g_{\theta^t}, \mathbf{A}^t)$ in the $t$-th iteration, for $t = 0, \ldots, \Delta$, where $\mathbf{A}^{t+1}$ is obtained based on different optimization methods. In conclusion, we compare the increment of the attacker's objective caused by a single optimization step of different optimization methods in this case. The result is shown in Figure 5(b). For G-FairAttack, the attacker's objective increases in 28 iterations, i.e., positive $\Delta_{dp}$ variation, and decreases in 1 iteration. For Metattack, the attacker objective increases in only 5 iterations and decreases in 5 iterations as well. Therefore, a single step of G-FairAttack leads to a larger increase of the attacker's objective than Metattack given the same initial point $\mathbf{A}^t$. In conclusion, our proposed G-FairAttack leads to a better solution in the optimization process compared with gradient-based methods in both fairness evasion attacks and fairness poisoning attacks.

# E  IMPLEMENTATION DETAILS

## E.1  DATASETS

The statistics of these datasets are shown in Table 3. In the Facebook graph (Leskovec & Mcauley, 2012), the nodes represent user accounts of Facebook, and the edges represent the friendship relations between users. Node attributes are collected from user profiles. The sensitive attributes of user nodes are their genders. The task of Facebook is to predict the education type of the users. In the Credit defaulter graph (Dai & Wang, 2021), the nodes represent credit card users, and the edges represent whether two users are similar or not according to their spending and payment patterns. The sensitive attributes of user nodes are their ages. The task of Credit is to predict whether a user will default on the credit card payment or not. In the Pokec graph (Agarwal et al., 2021), the nodes represent user accounts of a prevalent social network in Slovakia, and the edges represent the friendship relations between users. The sensitive attributes of user nodes are their regions. The task of Pokec is to predict the working fields of the users. It is worth noting that we use a subgraph of Pokec in (Dong et al., 2022a) instead of the original version in (Dai & Wang, 2021). In our experiment implementation, we adopt the *PyGDebias* library (Dong et al., 2023a) to load these datasets.

## E.2  BASELINES

We first introduce the detailed settings of the attack baselines in the first stage of evaluation.

- Random (Zügner et al., 2018; Hussain et al., 2022): Given the attack budget $\Delta$, we randomly flip $\Delta$ edges (removing existing edges or adding new edges) and obtain the attacked graph. It is a random method that fits both the fairness evasion attack setting and the fairness poisoning attack setting.

- FA-GNN (Hussain et al., 2022): Given the attack budget $\Delta$, we randomly link $\Delta$ pairs of nodes that belong to different classes and different sensitive groups. Specifically, we choose the most effective link strategy, "DD", in our experiments. It is also a random method that fits both attack settings.

- Gradient Ascent: Given the attack budget $\Delta$, we flip $\Delta$ edges sequentially in $\Delta$ iterations. In each iteration, we compute the gradient $\nabla_{\mathbf{A}'} \mathcal{L}_f(g_{\theta^*}, \mathbf{A}')$ and flip one edge corresponding to the largest element of the gradient, where $\theta^* = \arg\min_{\theta} \mathcal{L}_s(g_{\theta}, \mathbf{A})$. To make $\mathcal{L}_f$ differentiable, we use the soft predictions to substitute the prediction labels in $\mathcal{L}_f$ as (Zeng et al., 2021). Here, we choose a two-layer graph convolutional network (Kipf & Welling, 2017) as the surrogate model $g_{\theta}$, and CE loss as the surrogate loss function $\mathcal{L}_s$. Gradient Ascent is an optimization-based method that only fits the fairness evasion attack setting.

- Metattack (Zügner & Günnemann, 2019): Given the attack budget $\Delta$, we sequentially flip $\Delta$ edges in $\Delta$ iterations. In each iteration, we compute the meta gradient $\nabla_{\mathbf{A}'} \mathcal{L}_f(g_{\theta^*}(\mathbf{A}'), \mathbf{A}')$ by MAML (Finn et al., 2017) and flip one edge corresponding to the largest element of the meta gradient, where $\theta^* = \arg\min_{\theta} \mathcal{L}_s(g_{\theta}, \mathbf{A}')$. To make $\mathcal{L}_f$ differentiable, we implement the same adaption of $\mathcal{L}_f$ as Gradient Ascent. Here, we also choose a two-layer GCN as the surrogate model $g_{\theta}$ and CE loss as the surrogate loss function $\mathcal{L}_s$. Metattack is also an optimization-based method, while it only fits the fairness poisoning attack.

It is worth noting that for both Gradient Ascent and Metattack, we should flip the sign of the gradient components for connected node pairs as this yields the gradient for a change in the negative direction (i.e., removing the edge) (Zügner & Günnemann, 2019). Hence, we flip the edge corresponding to the largest score $\nabla_{\mathbf{A}_{[u,v]}} \mathcal{L}_f \cdot (-2 \cdot \mathbf{A}_{[u,v]} + 1)$ for $(u, v) \in \mathcal{E}$ for Gradient Ascent and Metattack.

Next, we introduce our adopted test GNNs in the second stage. We choose four types of GNNs, including a vanilla GNN and three types of fairness-aware GNNs.

- Vanilla: We choose a two-layer GCN (Kipf & Welling, 2017), which is a mostly adopted GNN model in existing works.

- $\Delta_{dp}$: We choose the output-based regularization method (Navarin et al., 2020; Zeng et al., 2021; Wang et al., 2022a; Dong et al., 2023c). Specifically, we choose a two-layer GCN as the backbone and add a regularization term $\Delta_{dp}$ to the loss function. Moreover, to make the regularization term differentiable, we use the soft prediction to substitute the prediction label in $\Delta_{dp}$ as (Zeng et al., 2021).

- $I(\hat{Y}, S)$: For mutual information loss, except for directly decreasing the mutual information, the adversarial training (Bose & Hamilton, 2019; Dai & Wang, 2021) could also be seen as a specific case of exploiting the mutual information loss according to (Kang et al., 2022). In an adversarial training framework, an adversary is trained to predict $S$ based on $\hat{Y}$. If the prediction is more accurate, it demonstrates that the output $\hat{Y}$ of the GNN contains more information about the sensitive attribute $S$, i.e., the GNN model is less fair. We choose FairGNN (Dai & Wang, 2021) as the baseline in the mutual information type. FairGNN contains a discriminator to predict the sensitive attribute based on the output of a GNN backbone. The GNN backbone is trained to fool the discriminator for predicting the sensitive attribute. Specifically, we choose a single-layer GCN as the GNN backbone.

- $W(\hat{Y}, S)$: We choose EDITS (Dong et al., 2022a), a model agnostic debiasing framework for GNNs. EDITS finds a debiased adjacency matrix and a debiased node attribute matrix by minimizing the Wasserstein distance of the distributions of node embeddings on different sensitive groups. With the debiased input graph data, the fairness of the GNN backbone is improved. Specifically, we choose a two-layer GCN as the GNN backbone.

### E.3 EXPERIMENTAL SETTINGS

The implementation of our experiments could be divided into two parts, fairness attack methods, and the test GNN models. For attack methods, we use the source code of FA-GNN (Hussain et al., 2022) and Metattack (Zügner & Günnemann, 2019). Other attack methods including G-FairAttack are implemented in PyTorch (Paszke et al., 2019). For test GNN models, we use the source code of GCN (Kipf & Welling, 2017), FairGNN (Dai & Wang, 2021), and EDITS (Dong et al., 2022a).

Table 4: The hyperparameter settings of four different types of test GNNs on three benchmarks in fairness evasion attack and fairness poisoning attack settings. The detailed definition and description of parameters $\alpha$, $\beta$, $\mu_1$, $\mu_2$, $\mu_3$, $\mu_4$, and $r$ refers to (Dai & Wang, 2021; Dong et al., 2022a).

| Test GNN | Hyperparameter | Fairness Evasion Attack | | | Fairness Poisoning Attack | | |
|---|---|---|---|---|---|---|---|
| | | Facebook | Pokec | Credit | Facebook | Pokec | Credit |
| GCN | learning rate | $1e^{-4}$ | $1e^{-4}$ | $1e^{-2}$ | $1e^{-4}$ | $1e^{-3}$ | $1e^{-3}$ |
| | weight decay | $1e^{-5}$ | $1e^{-2}$ | $1e^{-5}$ | $1e^{-5}$ | $1e^{-5}$ | $1e^{-5}$ |
| | dropout | 0.5 | 0.2 | 0.5 | 0.5 | 0.5 | 0.5 |
| | epochs | 2,000 | 1,000 | 1,000 | 1,000 | 500 | 200 |
| Reg | learning rate | $1e^{-4}$ | $1e^{-3}$ | $5e^{-2}$ | $1e^{-4}$ | $1e^{-3}$ | $5e^{-3}$ |
| | weight decay | $1e^{-5}$ | $1e^{-5}$ | $1e^{-5}$ | $1e^{-5}$ | $1e^{-5}$ | $1e^{-5}$ |
| | dropout | 0.5 | 0.2 | 0.2 | 0.5 | 0.5 | 0.5 |
| | epochs | 2,000 | 10,000 | 2,000 | 1,000 | 1,000 | 500 |
| | fairness loss weight $\alpha$ | 1 | 150 | 1 | 1 | 80 | 1 |
| FairGNN | learning rate | $1e^{-3}$ | $5e^{-3}$ | $1e^{-2}$ | $1e^{-4}$ | $5e^{-3}$ | $1e^{-3}$ |
| | weight decay | $1e^{-5}$ | $1e^{-5}$ | $1e^{-5}$ | $1e^{-5}$ | $1e^{-5}$ | $1e^{-5}$ |
| | dropout | 0.5 | 0.5 | 0.2 | 0.5 | 0.5 | 0.5 |
| | epochs | 2,000 | 5,000 | 1,000 | 1,500 | 5,000 | 1,000 |
| | covariance constraint weight $\alpha$ | 60 | 1 | 20 | 2 | 100 | 30 |
| | adversarial debiasing weight $\beta$ | 10 | 500 | 10 | 3 | 1,000 | 1 |
| EDITS | learning rate | $1e^{-4}$ | $1e^{-4}$ | $5e^{-3}$ | $1e^{-3}$ | $1e^{-4}$ | $1e^{-3}$ |
| | weight decay | $1e^{-5}$ | $1e^{-5}$ | $1e^{-5}$ | $1e^{-5}$ | $1e^{-5}$ | $1e^{-5}$ |
| | dropout | $5e^{-2}$ | $5e^{-2}$ | $5e^{-2}$ | $5e^{-2}$ | $5e^{-2}$ | $5e^{-2}$ |
| | epochs | 1,000 | 500 | 500 | 2,000 | 1,000 | 200 |
| | $\mu_1$ | $1e^{-2}$ | $3e^{-2}$ | $5e^{-2}$ | $1e^{-2}$ | $3e^{-2}$ | $5e^{-2}$ |
| | $\mu_2$ | 0.8 | 70 | 0.1 | 0.8 | 70 | 0.1 |
| | $\mu_3$ | 0.1 | 1 | 1 | 0.1 | 1 | 1 |
| | $\mu_4$ | 20 | 15 | 15 | 20 | 15 | 15 |
| | edge binarization threshold $r$ | $1e^{-4}$ | 0.3 | $5e^{-3}$ | $1e^{-4}$ | 0.2 | $2e^{-2}$ |

We exploit Adam optimizer (Kingma & Ba, 2015) to optimize the surrogate model, gradient-based attacks, and the test GNNs. All experiments are implemented on an Nvidia RTX A6000 GPU. We provide the hyperparameter settings of G-FairAttack in Table 5, and the hyperparameter settings of test GNNs in Table 4. [3]

### E.4 REQUIRED PACKAGES

We list some key packages in Python required for implementing G-FairAttack as follows.

- Python == 3.9.13
- torch == 1.11.0
- torch-geometric == 2.0.4
- numpy == 1.21.5
- numba == 0.56.3
- networkx == 2.8.4
- scikit-learn == 1.1.1
- scipy == 1.9.1
- dgl == 0.9.1
- deeprobust == 0.2.5

---

[3]The open-source code is available at `https://github.com/zhangbinchi/G-FairAttack`.

## F  SUPPLEMENTARY EXPERIMENTS

### F.1  EFFECTIVENESS OF ATTACK

In Section 4.2, we discussed the effectiveness of G-FairAttack in attacking the fairness of different types of GNNs. Here, we provide more results of the same experiment as Section 4.2 but in different attack settings and more fairness metrics. Table 6 shows the experimental results of fairness evasion attacks in $\Delta_{eo}$ fairness metric, Table 7 shows the experimental results of fairness poisoning attacks in $\Delta_{dp}$ fairness metric, and Table 8 shows the experimental results of fairness poisoning attacks in $\Delta_{eo}$ fairness metric. From the re-

Table 5: The hyperparameter settings of G-FairAttack on three benchmarks. The "(s)" denotes that the corresponding hyperparameter is related to the training of the surrogate model.

| Hyperparameter | Facebook | Pokec | Credit |
|---|---|---|---|
| attack budget $\Delta$ | 5% | 5% | 1% |
| utility budget $\epsilon$ | 5% | 5% | 5% |
| threshold $a$ | 0.1 | $5e^{-3}$ | $1e^{-4}$ |
| learning rate (s) | $1e^{-3}$ | $1e^{-3}$ | $5e^{-2}$ |
| weight decay (s) | $1e^{-5}$ | $1e^{-5}$ | $1e^{-5}$ |
| epochs (s) | 2,000 | 2,000 | 1,000 |
| dropout (s) | 0.5 | 0.5 | 0.5 |
| fairness loss weight $\alpha$ (s) | 1 | 1 | 1 |
| bandwidth $h$ (s) | 0.1 | 0.1 | 0.1 |
| number of intervals $m$ (s) | 10,000 | 10,000 | 1,000 |

sults, we can find that the overall performance of G-FairAttack is better than any other baselines, which highlights the clear superiority of G-FairAttack. In addition, we can find that G-FairAttack seems less desirable in the cases where vanilla GCN serves as the victim model. The reason for this phenomenon is our surrogate loss contains two parts, utility loss term $L$ and fairness loss term $L_f$, while the real victim loss of vanilla GCN only contains the utility loss term $L$. However, in the general case (without knowing the type of the victim model), G-FairAttack can outperform all other baselines when the surrogate loss differs from the victim loss (when attacking fairness-aware GNNs). Despite that, the shortage in attacking vanilla GNNs can be solved easily by choosing a smaller hyperparameter $\alpha$ (the weight of $L_f$). As Table 5 shows, we fix the hyperparameter setting (also fix $\alpha$) for all attack methods when attacking different types of victim models because the attacker cannot choose different hyperparameters based on the specific type of the victim model in the gray-box attack setting. By fixing the hyperparameter settings for different victim models, the experimental results demonstrate that our G-FairAttack is **victim model-agnostic** and **easy to use** (in terms of hyperparameter tuning). In addition, we can also find that the fairness poisoning attack is a more challenging task (easier to fail), while effective fairness poisoning attack methods induce a more serious deterioration in the fairness of the victim model. Furthermore, we obtain that edge rewiring based methods (EDITS) can effectively defend against some fairness attacks (more details about this conclusion are discussed in Appendix G).

### F.2  EFFECTIVENESS OF SURROGATE LOSS

In Section 4.3, we discussed the effectiveness of our proposed surrogate loss in representing different types of fairness loss in different victim models. Here, we provide more results of the same experiment as Section 4.3 but in different attack settings and datasets. Table 9 shows the experimental results of fairness evasion attacks on the Facebook dataset. As mentioned in Section 4.3, the surrogate loss function in the attack method and the victim loss function in the victim model are different in our attack setting. The attacker does not know the form of the victim loss function due to the gray-box setting. If the surrogate loss is the same as the victim loss, the attack method naturally can have a desirable performance (as white-box attacks). However, in this experiment, we demonstrate that G-FairAttack (with our proposed surrogate loss) has the most desirable performance when attacking different victim models **with a different victim loss** from the surrogate loss. Consequently, our experiment verifies that our surrogate loss function is adaptable to attacking different types of victim models.

Table 9: Results of G-FairAttack on Facebook while replacing the total variation loss with other loss terms.

| Attack | FairGNN | | EDITS | |
|---|---|---|---|---|
| | $\Delta_{dp}$ (%) | $\Delta_{eo}$ (%) | $\Delta_{dp}$ (%) | $\Delta_{eo}$ (%) |
| Clean | $6.23 \pm 0.69$ | $4.78 \pm 0.98$ | $1.15 \pm 0.25$ | $2.31 \pm 0.58$ |
| G-FA-None | $6.35 \pm 0.83$ | $4.78 \pm 0.98$ | $4.19 \pm 0.29$ | $2.88 \pm 0.33$ |
| G-FA-$\Delta_{dp}$ | $6.35 \pm 0.83$ | $4.78 \pm 0.98$ | $3.86 \pm 0.29$ | $2.88 \pm 0.33$ |
| G-FA | $8.62 \pm 0.91$ | $5.70 \pm 0.64$ | $4.19 \pm 0.29$ | $2.88 \pm 0.33$ |

Table 6: Experiment results of fairness evasion attack, where AUC and $\Delta_{eo}$ are adopted as the prediction utility metric and the fairness metric, correspondingly.

| Victim | Attack | Facebook | | Pokec_z | | Credit | |
|---|---|---|---|---|---|---|---|
| | | AUC(%) | $\Delta_{eo}$(%) | AUC(%) | $\Delta_{eo}$(%) | AUC(%) | $\Delta_{eo}$(%) |
| GCN | Clean | 64.53 ± 0.05 | 0.52 ± 0.41 | 72.87 ± 0.20 | 8.75 ± 0.94 | 70.48 ± 0.14 | 12.29 ± 2.14 |
| | Random | 65.05 ± 0.10 | 1.55 ± 0.00 | 72.56 ± 0.20 | **8.71 ± 2.50** | 70.53 ± 0.13 | 12.57 ± 2.29 |
| | FA-GNN | 63.00 ± 0.06 | 2.12 ± 0.58 | 72.12 ± 0.28 | 1.31 ± 1.05 | 70.44 ± 0.12 | **14.39 ± 2.65** |
| | Gradient Ascent | 64.77 ± 0.11 | 1.93 ± 0.33 | 72.88 ± 0.18 | 5.93 ± 1.74 | - | - |
| | G-FairAttack | 64.20 ± 0.03 | **2.38 ± 0.68** | 72.91 ± 0.20 | 6.69 ± 1.63 | 70.44 ± 0.11 | 12.75 ± 2.12 |
| Reg | Clean | 64.09 ± 0.16 | 0.38 ± 0.00 | 71.08 ± 0.35 | 1.73 ± 1.15 | 70.26 ± 0.34 | 0.66 ± 0.84 |
| | Random | 64.49 ± 0.26 | 0.77 ± 0.33 | 70.86 ± 0.30 | 1.34 ± 1.12 | 70.31 ± 0.34 | 0.82 ± 0.98 |
| | FA-GNN | 62.28 ± 0.23 | 0.90 ± 0.78 | 70.12 ± 0.25 | 5.23 ± 0.38 | 70.34 ± 0.36 | 0.32 ± 0.27 |
| | Gradient Ascent | 64.40 ± 0.12 | 0.20 ± 0.00 | 71.21 ± 0.49 | 2.25 ± 1.19 | - | - |
| | G-FairAttack | 63.30 ± 0.14 | **1.74 ± 0.33** | 71.11 ± 0.26 | **7.14 ± 1.26** | 70.21 ± 0.37 | **2.08 ± 1.20** |
| FairGNN | Clean | 65.24 ± 0.49 | 0.97 ± 0.77 | 69.71 ± 0.09 | 0.81 ± 0.69 | 67.32 ± 0.46 | 0.94 ± 0.67 |
| | Random | 64.88 ± 0.52 | 0.44 ± 0.29 | 69.36 ± 0.20 | 0.48 ± 0.40 | 67.32 ± 0.44 | 0.92 ± 0.69 |
| | FA-GNN | 59.78 ± 0.96 | 1.01 ± 1.29 | 69.57 ± 0.28 | **2.58 ± 0.58** | 67.26 ± 0.49 | 0.95 ± 0.42 |
| | Gradient Ascent | 65.19 ± 0.47 | 1.01 ± 1.29 | 69.21 ± 0.16 | 1.25 ± 0.43 | - | - |
| | G-FairAttack | 64.34 ± 0.45 | **1.79 ± 1.30** | 69.83 ± 0.14 | 2.30 ± 0.49 | 67.30 ± 0.47 | **1.12 ± 1.04** |
| EDITS | Clean | 70.95 ± 0.28 | 0.38 ± 0.00 | 70.89 ± 0.44 | 5.15 ± 0.51 | 69.11 ± 0.50 | 6.15 ± 1.67 |
| | Random | 68.13 ± 0.20 | 0.97 ± 0.68 | 70.89 ± 0.44 | 5.15 ± 0.51 | 69.04 ± 0.06 | **6.93 ± 2.13** |
| | FA-GNN | 70.72 ± 0.27 | 0.38 ± 0.00 | 70.89 ± 0.44 | 5.15 ± 0.51 | 69.11 ± 0.50 | 6.16 ± 1.66 |
| | Gradient Ascent | 68.36 ± 0.24 | 0.57 ± 0.56 | 70.89 ± 0.44 | 5.15 ± 0.51 | - | - |
| | G-FairAttack | 68.12 ± 0.22 | **1.23 ± 0.73** | 70.99 ± 0.44 | **6.18 ± 1.23** | 69.04 ± 0.06 | **6.93 ± 2.13** |

Table 7: Experiment results of fairness poisoning attack, where ACC and $\Delta_{dp}$ are adopted as the prediction utility metric and the fairness metric, correspondingly.

| Victim | Attack | Facebook | | Pokec_z | | Credit | |
|---|---|---|---|---|---|---|---|
| | | ACC(%) | $\Delta_{dp}$(%) | ACC(%) | $\Delta_{dp}$(%) | ACC(%) | $\Delta_{dp}$(%) |
| GCN | Clean | 80.25 ± 0.64 | 6.33 ± 0.54 | 61.51 ± 0.54 | 8.39 ± 1.25 | 69.77 ± 0.36 | 10.61 ± 0.73 |
| | Random | 80.15 ± 1.02 | 4.17 ± 1.58 | 60.98 ± 0.29 | 7.76 ± 0.93 | 69.71 ± 0.55 | 10.96 ± 1.35 |
| | FA-GNN | 79.41 ± 0.67 | 5.84 ± 0.32 | 59.97 ± 0.90 | 2.46 ± 1.18 | 69.56 ± 0.81 | **18.58 ± 1.78** |
| | Metattack | 79.30 ± 1.15 | **33.33 ± 5.97** | 60.14 ± 0.29 | **44.90 ± 0.10** | - | - |
| | G-FairAttack | 78.55 ± 0.18 | 14.99 ± 2.08 | 61.81 ± 0.21 | 6.15 ± 1.52 | 69.57 ± 1.13 | 10.87 ± 0.96 |
| Reg | Clean | 80.41 ± 0.18 | 4.63 ± 1.18 | 65.20 ± 1.26 | 1.55 ± 0.82 | 68.22 ± 0.73 | 1.48 ± 0.28 |
| | Random | 80.18 ± 0.30 | 1.41 ± 1.23 | 63.00 ± 1.50 | 2.01 ± 1.98 | 68.62 ± 0.48 | 1.58 ± 0.64 |
| | FA-GNN | 79.38 ± 0.31 | 1.42 ± 0.89 | 64.02 ± 0.47 | 0.76 ± 0.81 | 69.29 ± 0.63 | 0.79 ± 0.41 |
| | Metattack | 80.81 ± 0.40 | 9.45 ± 0.96 | 62.89 ± 1.90 | 4.88 ± 1.62 | - | - |
| | G-FairAttack | 77.95 ± 0.16 | **15.65 ± 0.69** | 65.45 ± 0.55 | **7.76 ± 0.08** | 67.80 ± 1.34 | **2.32 ± 0.51** |
| FairGNN | Clean | 80.36 ± 0.18 | 2.71 ± 0.50 | 60.87 ± 3.00 | 0.80 ± 0.62 | 75.57 ± 0.65 | 4.78 ± 1.58 |
| | Random | 78.66 ± 0.00 | 0.18 ± 0.31 | 54.21 ± 5.10 | 1.79 ± 0.93 | 75.42 ± 0.57 | 5.08 ± 1.71 |
| | FA-GNN | 78.77 ± 0.18 | 0.33 ± 0.58 | 60.54 ± 2.48 | 1.24 ± 0.77 | 75.38 ± 0.88 | 4.89 ± 0.56 |
| | Metattack | 78.66 ± 0.84 | 3.86 ± 3.51 | 55.16 ± 6.70 | **6.08 ± 3.64** | - | - |
| | G-FairAttack | 77.92 ± 0.18 | **10.64 ± 0.97** | 59.36 ± 1.19 | 3.91 ± 3.07 | 75.47 ± 0.56 | **5.85 ± 1.66** |
| EDITS | Clean | 79.62 ± 1.10 | 1.36 ± 1.54 | 62.33 ± 0.84 | 3.32 ± 0.64 | 67.73 ± 0.46 | 7.23 ± 0.33 |
| | Random | 81.21 ± 0.32 | 3.86 ± 2.27 | 62.49 ± 0.69 | 4.62 ± 0.97 | 69.19 ± 0.86 | 7.91 ± 0.91 |
| | FA-GNN | 79.83 ± 0.18 | 3.04 ± 0.50 | 63.38 ± 0.18 | 2.47 ± 1.38 | 67.55 ± 0.89 | 7.30 ± 0.19 |
| | Metattack | 81.95 ± 1.21 | 4.50 ± 0.67 | 62.72 ± 0.19 | 2.83 ± 0.80 | - | - |
| | G-FairAttack | 81.95 ± 0.48 | **6.15 ± 0.77** | 62.65 ± 0.81 | **4.85 ± 0.44** | 69.01 ± 0.79 | **8.14 ± 0.41** |

## F.3 ATTACK PATTERNS

In this experiment, we compare the patterns of G-FairAttack with FA-GNN (Hussain et al., 2022). FA-GNN divides edges into four different groups, 'EE', 'ED', 'DE', and 'DD', where edges in EE link two nodes with the same label and the same sensitive attribute, edges in ED link two nodes with the same label and different sensitive attributes, edges in DE link two nodes with different labels and the same sensitive attribute, edges in DD link two nodes with different labels and different sensitive attributes In particular, we record the proportion of poisoned edges in these four groups yielded by

Table 8: Experiment results of fairness poisoning attack, where AUC and $\Delta_{eo}$ are adopted as the prediction utility metric and the fairness metric, correspondingly.

| Victim | Attack | Facebook | | Pokec_z | | Credit | |
|---|---|---|---|---|---|---|---|
| | | AUC(%) | $\Delta_{eo}$(%) | AUC(%) | $\Delta_{eo}$(%) | AUC(%) | $\Delta_{eo}$(%) |
| GCN | Clean | 64.53 ± 0.05 | 1.16 ± 0.78 | 67.53 ± 0.25 | 10.16 ± 1.20 | 69.36 ± 0.08 | 9.83 ± 0.76 |
| | Random | 65.49 ± 0.37 | 1.29 ± 0.87 | 66.97 ± 0.11 | 8.81 ± 1.07 | 69.45 ± 0.06 | 10.18 ± 1.62 |
| | FA-GNN | 64.45 ± 0.68 | 2.51 ± 0.33 | 65.77 ± 0.43 | 1.78 ± 0.66 | 69.16 ± 0.11 | **18.55 ± 2.02** |
| | Metattack | 65.02 ± 0.17 | **24.84 ± 6.37** | 64.33 ± 0.33 | **43.71 ± 1.18** | - | - |
| | G-FairAttack | 62.41 ± 0.07 | 9.53 ± 1.56 | 67.26 ± 0.26 | 9.71 ± 1.84 | 69.37 ± 0.14 | 10.23 ± 1.29 |
| Reg | Clean | 64.57 ± 0.21 | 1.02 ± 0.91 | 68.93 ± 1.25 | 1.39 ± 0.85 | 69.70 ± 0.49 | 1.01 ± 1.40 |
| | Random | 64.45 ± 0.36 | 1.40 ± 0.62 | 66.41 ± 2.23 | 1.22 ± 0.41 | 69.79 ± 0.46 | 0.86 ± 1.00 |
| | FA-GNN | 61.67 ± 0.53 | 0.29 ± 0.58 | 66.65 ± 0.66 | 3.05 ± 2.29 | 69.67 ± 0.76 | 1.42 ± 0.79 |
| | Metattack | 64.31 ± 0.17 | 4.88 ± 1.43 | 65.94 ± 2.99 | 6.23 ± 0.38 | - | - |
| | G-FairAttack | 61.69 ± 0.14 | **9.80 ± 0.91** | 68.06 ± 1.58 | **6.88 ± 2.35** | 69.76 ± 0.49 | **2.36 ± 1.50** |
| FairGNN | Clean | 65.57 ± 0.40 | 0.65 ± 0.78 | 65.22 ± 2.04 | 0.94 ± 0.83 | 65.95 ± 0.46 | 3.04 ± 1.35 |
| | Random | 64.66 ± 0.75 | 0.19 ± 0.33 | 59.56 ± 1.92 | 2.46 ± 2.43 | 65.48 ± 0.55 | 3.35 ± 1.41 |
| | FA-GNN | 63.54 ± 0.14 | 0.00 ± 0.00 | 64.73 ± 2.79 | 2.13 ± 1.35 | 65.13 ± 0.60 | 3.19 ± 0.66 |
| | Metattack | 65.10 ± 0.42 | 3.60 ± 4.13 | 57.99 ± 8.86 | **7.13 ± 3.07** | - | - |
| | G-FairAttack | 62.99 ± 0.29 | **4.70 ± 1.44** | 65.79 ± 2.64 | 2.78 ± 1.51 | 65.89 ± 0.53 | **3.83 ± 1.64** |
| EDITS | Clean | 79.00 ± 2.49 | 0.19 ± 0.33 | 67.00 ± 0.59 | 3.19 ± 0.84 | 69.83 ± 0.13 | 6.81 ± 0.49 |
| | Random | 71.58 ± 0.36 | 1.03 ± 0.95 | 68.23 ± 0.62 | 6.79 ± 1.00 | 69.55 ± 0.27 | 6.91 ± 1.05 |
| | FA-GNN | 77.06 ± 2.13 | 0.58 ± 0.58 | 67.55 ± 0.32 | 2.86 ± 2.67 | 69.82 ± 0.17 | 6.83 ± 0.52 |
| | Metattack | 71.60 ± 0.40 | 0.52 ± 0.44 | 67.51 ± 0.56 | 3.15 ± 1.23 | - | - |
| | G-FairAttack | 70.33 ± 1.76 | **1.61 ± 0.59** | 68.26 ± 0.57 | **6.98 ± 0.29** | 69.62 ± 0.20 | **7.29 ± 0.53** |

Table 10: Attack patterns (statistics of poisoned edges in different groups) of G-FairAttack.

| Dataset | Fairness Evasion Attack | | | | Fairness Poisoning Attack | | | |
|---|---|---|---|---|---|---|---|---|
| | EE/% | ED/% | DE/% | DD/% | EE/% | ED/% | DE/% | DD/% |
| Facebook | 43.46 | 28.42 | 14.44 | 13.96 | 31.86 | 28.72 | 19.07 | 20.34 |
| Pokec | 25.51 | 23.47 | 26.39 | 24.63 | 26.58 | 24.73 | 22.01 | 26.68 |
| Credit | 74.89 | 7.37 | 16.06 | 1.68 | 38.63 | 22.59 | 24.78 | 13.99 |

**G-FairAttack and FA-GNN.** The results of attack patterns of G-FairAttack are shown in Table 10. According to the study in (Hussain et al., 2022), injecting edges in DD and EE can increase the statistical parity difference. Based on this guidance, FA-GNN randomly injects edges that belong to group DD to attack the fairness of GNNs. Hence, the attack patterns of FA-GNN for all datasets and attack settings are all the same, i.e., 100% for the DD group and 0% for the other groups. However, our proposed G-FairAttack has different patterns of poisoned edges. For the Facebook and the Credit dataset, G-FairAttack poisons more EE edges than other groups. For the Pokec dataset, the proportion of poisoned edges in four groups is balanced. Although we cannot analyze the reason for the apparent difference in this paper, we can argue that G-FairAttack is much harder to defend because it does not have a fixed pattern for all cases, unlike FA-GNN.

## F.4 ATTACK GENERALIZATION

Although the generalization capability of adversarial attacks on GNNs with a linearized surrogate model has been verified by previous works (Zügner et al., 2018; Zügner & Günnemann, 2019), we conduct numerical experiments to verify the generalization capability of G-FairAttack to other GNN architectures. In the experiments, G-FairAttack is still trained with a two-layer linearized GCN as the surrogate model, while we choose two different GNN architectures, GraphSAGE (Hamilton et al., 2017) and GAT (Veličković et al., 2018), as the backbone of adopted victim models. Experimental results are shown in Tables 11 to 14. From the results, we can observe that G-FairAttack still successfully reduces the fairness of victim models with different GNN backbones, and G-FairAttack still has the most desirable performance on attacking different types of fairness-aware GNNs. In addition, to verify the generality of G-FairAttack, we conduct experiments on the German Credit dataset (Agarwal et al., 2021). In the German dataset, nodes represent customers of a German bank, and edges

Table 11: Experiment results of fairness evasion attack on the Facebook dataset. All victim models adopt GraphSAGE as the GNN backbone.

| | Attack | ACC(%) | AUC(%) | $\Delta_{dp}$(%) | $\Delta_{eo}$(%) |
|---|---|---|---|---|---|
| SAGE | Clean | $93.20 \pm 0.15$ | $93.78 \pm 0.08$ | $14.00 \pm 0.72$ | $7.08 \pm 0.64$ |
| | Random | $93.95 \pm 0.26$ | $94.32 \pm 0.11$ | $15.14 \pm 0.63$ | $7.21 \pm 0.64$ |
| | FA-GNN | $93.31 \pm 0.26$ | $93.83 \pm 0.07$ | $13.14 \pm 0.63$ | $5.28 \pm 0.64$ |
| | Gradient Ascent | $92.99 \pm 0.26$ | $94.00 \pm 0.02$ | $14.67 \pm 0.63$ | $\mathbf{7.98 \pm 0.64}$ |
| | G-FairAttack | $93.42 \pm 0.30$ | $93.63 \pm 0.09$ | $\mathbf{15.33 \pm 0.25}$ | $7.53 \pm 0.00$ |
| Reg | Clean | $91.93 \pm 0.15$ | $93.82 \pm 0.12$ | $4.13 \pm 0.72$ | $1.93 \pm 0.27$ |
| | Random | $92.78 \pm 0.15$ | $94.41 \pm 0.13$ | $\mathbf{6.44 \pm 0.47}$ | $2.32 \pm 0.00$ |
| | FA-GNN | $92.04 \pm 0.26$ | $93.91 \pm 0.05$ | $2.95 \pm 1.32$ | $0.58 \pm 0.27$ |
| | Gradient Ascent | $91.83 \pm 0.15$ | $93.89 \pm 0.10$ | $3.67 \pm 0.44$ | $1.55 \pm 0.55$ |
| | G-FairAttack | $92.15 \pm 0.30$ | $93.72 \pm 0.13$ | $5.46 \pm 0.25$ | $\mathbf{2.38 \pm 0.91}$ |
| FairGNN | Clean | $87.26 \pm 0.94$ | $93.74 \pm 0.50$ | $0.91 \pm 0.56$ | $0.90 \pm 0.64$ |
| | Random | $87.16 \pm 0.91$ | $93.83 \pm 0.40$ | $1.83 \pm 1.13$ | $0.45 \pm 0.64$ |
| | FA-GNN | $87.37 \pm 1.17$ | $93.07 \pm 0.48$ | $1.91 \pm 1.06$ | $0.45 \pm 0.64$ |
| | Gradient Ascent | $87.48 \pm 0.84$ | $93.58 \pm 0.47$ | $1.14 \pm 0.63$ | $\mathbf{0.90 \pm 0.64}$ |
| | G-FairAttack | $87.26 \pm 1.30$ | $93.66 \pm 0.48$ | $\mathbf{2.16 \pm 0.96}$ | $\mathbf{0.90 \pm 0.64}$ |
| EDITS | Clean | $93.42 \pm 0.15$ | $93.62 \pm 0.03$ | $8.94 \pm 0.41$ | $1.22 \pm 0.64$ |
| | Random | $92.60 \pm 0.14$ | $93.82 \pm 0.16$ | $11.57 \pm 0.20$ | $\mathbf{6.76 \pm 0.00}$ |
| | FA-GNN | $93.39 \pm 0.14$ | $93.67 \pm 0.07$ | $9.07 \pm 0.42$ | $1.45 \pm 0.68$ |
| | Gradient Ascent | $92.83 \pm 0.27$ | $93.80 \pm 0.09$ | $\mathbf{12.29 \pm 0.63}$ | $5.75 \pm 0.58$ |
| | G-FairAttack | $92.57 \pm 0.15$ | $93.74 \pm 0.12$ | $11.61 \pm 0.22$ | $\mathbf{6.76 \pm 0.00}$ |

Table 12: Experiment results of fairness poisoning attack on the Facebook dataset. All victim models adopt GraphSAGE as the GNN backbone.

| | Attack | ACC(%) | AUC(%) | $\Delta_{dp}$(%) | $\Delta_{eo}$(%) |
|---|---|---|---|---|---|
| SAGE | Clean | $91.83 \pm 0.18$ | $94.05 \pm 0.11$ | $16.78 \pm 0.58$ | $9.40 \pm 0.78$ |
| | Random | $92.99 \pm 0.32$ | $94.09 \pm 0.03$ | $15.91 \pm 0.52$ | $8.56 \pm 0.45$ |
| | FA-GNN | $93.10 \pm 0.48$ | $94.83 \pm 0.10$ | $11.09 \pm 0.71$ | $5.22 \pm 0.00$ |
| | Metattack | $92.15 \pm 0.18$ | $92.91 \pm 0.15$ | $\mathbf{19.16 \pm 0.75}$ | $11.84 \pm 0.99$ |
| | G-FairAttack | $90.66 \pm 0.18$ | $93.83 \pm 0.04$ | $18.89 \pm 0.27$ | $\mathbf{12.94 \pm 0.33}$ |
| Reg | Clean | $83.86 \pm 0.48$ | $93.08 \pm 0.22$ | $4.03 \pm 0.53$ | $0.45 \pm 0.78$ |
| | Random | $82.69 \pm 0.37$ | $93.24 \pm 0.15$ | $3.96 \pm 0.31$ | $0.00 \pm 0.00$ |
| | FA-GNN | $83.33 \pm 0.37$ | $93.06 \pm 0.31$ | $5.38 \pm 0.54$ | $0.00 \pm 0.00$ |
| | Metattack | $84.39 \pm 0.00$ | $92.58 \pm 0.17$ | $2.54 \pm 0.00$ | $0.00 \pm 0.00$ |
| | G-FairAttack | $84.50 \pm 0.48$ | $90.73 \pm 0.37$ | $\mathbf{6.19 \pm 0.29}$ | $\mathbf{2.70 \pm 0.00}$ |
| FairGNN | Clean | $88.96 \pm 2.05$ | $94.17 \pm 0.19$ | $2.54 \pm 1.71$ | $1.35 \pm 0.00$ |
| | Random | $86.73 \pm 3.19$ | $94.06 \pm 0.33$ | $2.89 \pm 2.32$ | $0.00 \pm 0.00$ |
| | FA-GNN | $86.09 \pm 4.42$ | $94.48 \pm 0.23$ | $1.61 \pm 0.70$ | $0.00 \pm 0.00$ |
| | Metattack | $86.41 \pm 2.89$ | $93.75 \pm 0.15$ | $3.13 \pm 2.24$ | $0.90 \pm 0.78$ |
| | G-FairAttack | $86.41 \pm 0.80$ | $92.90 \pm 0.30$ | $\mathbf{5.03 \pm 0.48}$ | $\mathbf{3.15 \pm 0.78}$ |
| EDITS | Clean | $91.08 \pm 0.00$ | $92.08 \pm 0.28$ | $4.57 \pm 0.74$ | $1.36 \pm 0.33$ |
| | Random | $92.25 \pm 0.18$ | $93.83 \pm 0.09$ | $16.74 \pm 1.14$ | $9.72 \pm 1.26$ |
| | FA-GNN | $89.17 \pm 0.32$ | $91.23 \pm 0.17$ | $5.46 \pm 0.75$ | $7.41 \pm 1.45$ |
| | Metattack | $91.83 \pm 0.49$ | $93.65 \pm 0.11$ | $17.27 \pm 0.72$ | $10.04 \pm 0.89$ |
| | G-FairAttack | $92.04 \pm 0.32$ | $93.55 \pm 0.13$ | $\mathbf{17.58 \pm 0.47}$ | $\mathbf{10.43 \pm 0.58}$ |

are generated based on the similarity between credit accounts. The task is to predict whether a customer has a high credit risk or low, with gender as the sensitive attribute. In this experiment, we choose GraphSAGE (Hamilton et al., 2017) as the backbone of the victim models and adopt both fairness evasion and poisoning settings. Experimental results are shown in Tables 16 and 17. We can observe that the superiority of G-FairAttack is preserved on the German dataset. In conclusion, supplementary experiments verify the generalization capability of G-FairAttack in attacking victim models with various GNN architectures and consolidate the universality of G-FairAttack.

Table 13: Experiment results of fairness evasion attack on the Facebook dataset. All victim models adopt GAT as the GNN backbone.

| | Attack | ACC(%) | AUC(%) | $\Delta_{dp}$(%) | $\Delta_{eo}$(%) |
|---|---|---|---|---|---|
| GAT | Clean | $79.93 \pm 1.39$ | $66.97 \pm 2.36$ | $2.84 \pm 1.34$ | $0.71 \pm 0.44$ |
| | Random | $79.30 \pm 0.55$ | $67.33 \pm 0.50$ | $1.11 \pm 0.47$ | $0.51 \pm 0.48$ |
| | FA-GNN | $80.15 \pm 0.48$ | $67.45 \pm 2.45$ | $\mathbf{4.62 \pm 1.60}$ | $0.26 \pm 0.44$ |
| | Gradient Ascent | $80.25 \pm 1.40$ | $67.30 \pm 2.16$ | $4.24 \pm 1.16$ | $\underline{1.35 \pm 1.00}$ |
| | G-FairAttack | $79.09 \pm 1.21$ | $66.76 \pm 2.50$ | $\underline{4.35 \pm 1.66}$ | $\mathbf{1.73 \pm 1.67}$ |
| Reg | Clean | $80.15 \pm 0.36$ | $66.50 \pm 1.08$ | $4.88 \pm 0.60$ | $0.52 \pm 0.48$ |
| | Random | $79.83 \pm 0.48$ | $67.68 \pm 1.29$ | $2.82 \pm 2.10$ | $0.45 \pm 0.49$ |
| | FA-GNN | $80.15 \pm 0.36$ | $69.79 \pm 1.89$ | $\mathbf{5.64 \pm 0.96}$ | $\mathbf{1.10 \pm 0.95}$ |
| | Gradient Ascent | $80.36 \pm 0.18$ | $65.89 \pm 1.27$ | $5.19 \pm 1.22$ | $0.52 \pm 0.59$ |
| | G-FairAttack | $78.77 \pm 0.37$ | $66.74 \pm 1.09$ | $\underline{5.48 \pm 1.66}$ | $\underline{1.09 \pm 0.29}$ |
| FairGNN | Clean | $79.19 \pm 0.67$ | $66.17 \pm 1.75$ | $2.47 \pm 2.94$ | $0.64 \pm 0.29$ |
| | Random | $79.62 \pm 0.55$ | $62.54 \pm 2.47$ | $1.49 \pm 0.84$ | $0.26 \pm 0.44$ |
| | FA-GNN | $79.19 \pm 0.74$ | $64.29 \pm 3.51$ | $1.84 \pm 2.15$ | $\underline{0.90 \pm 0.62}$ |
| | Gradient Ascent | $78.98 \pm 0.32$ | $64.03 \pm 1.73$ | $\underline{2.73 \pm 2.82}$ | $0.39 \pm 0.33$ |
| | G-FairAttack | $78.45 \pm 0.80$ | $63.94 \pm 2.30$ | $\mathbf{3.69 \pm 2.41}$ | $\mathbf{2.13 \pm 0.78}$ |
| EDITS | Clean | $79.30 \pm 0.96$ | $68.59 \pm 3.81$ | $0.66 \pm 0.60$ | $3.27 \pm 2.19$ |
| | Random | $76.01 \pm 1.76$ | $63.08 \pm 1.90$ | $\underline{0.81 \pm 0.65}$ | $\mathbf{4.74 \pm 2.04}$ |
| | FA-GNN | $79.30 \pm 0.96$ | $68.58 \pm 3.87$ | $0.66 \pm 0.60$ | $\underline{3.27 \pm 2.19}$ |
| | Gradient Ascent | $77.50 \pm 1.50$ | $61.28 \pm 2.44$ | $0.81 \pm 0.20$ | $2.95 \pm 3.62$ |
| | G-FairAttack | $76.01 \pm 1.94$ | $63.08 \pm 1.92$ | $\mathbf{1.04 \pm 0.59}$ | $\mathbf{4.74 \pm 2.61}$ |

Table 14: Experiment results of fairness poisoning attack on the Facebook dataset. All victim models adopt GAT as the GNN backbone.

| | Attack | ACC(%) | AUC(%) | $\Delta_{dp}$(%) | $\Delta_{eo}$(%) |
|---|---|---|---|---|---|
| GAT | Clean | $69.21 \pm 0.80$ | $60.03 \pm 0.52$ | $8.09 \pm 3.17$ | $4.34 \pm 4.09$ |
| | Random | $71.76 \pm 0.48$ | $57.17 \pm 1.87$ | $3.81 \pm 3.36$ | $4.23 \pm 4.39$ |
| | FA-GNN | $80.68 \pm 0.49$ | $73.64 \pm 1.66$ | $2.97 \pm 1.22$ | $3.27 \pm 2.33$ |
| | Metattack | $69.11 \pm 0.85$ | $63.29 \pm 1.80$ | $\mathbf{74.62 \pm 3.79}$ | $\mathbf{70.30 \pm 2.58}$ |
| | G-FairAttack | $73.35 \pm 1.33$ | $57.36 \pm 2.10$ | $\underline{9.85 \pm 1.62}$ | $\underline{5.30 \pm 3.20}$ |
| Reg | Clean | $79.09 \pm 0.81$ | $64.86 \pm 0.81$ | $2.22 \pm 2.57$ | $0.71 \pm 0.11$ |
| | Random | $78.98 \pm 0.32$ | $67.50 \pm 0.53$ | $1.00 \pm 1.00$ | $0.00 \pm 0.00$ |
| | FA-GNN | $78.66 \pm 0.00$ | $69.19 \pm 2.44$ | $0.00 \pm 0.00$ | $0.00 \pm 0.00$ |
| | Metattack | $76.85 \pm 1.29$ | $63.39 \pm 1.12$ | $8.60 \pm 7.05$ | $\mathbf{7.40 \pm 5.04}$ |
| | G-FairAttack | $78.77 \pm 0.37$ | $59.78 \pm 1.00$ | $\mathbf{10.99 \pm 1.47}$ | $\underline{7.28 \pm 0.99}$ |
| FairGNN | Clean | $78.56 \pm 0.48$ | $56.57 \pm 7.18$ | $1.09 \pm 1.50$ | $0.77 \pm 1.33$ |
| | Random | $78.66 \pm 1.15$ | $52.88 \pm 1.68$ | $1.20 \pm 0.69$ | $1.15 \pm 0.70$ |
| | FA-GNN | $77.17 \pm 2.57$ | $49.37 \pm 4.58$ | $\underline{1.29 \pm 2.23}$ | $\underline{1.23 \pm 2.12}$ |
| | Metattack | $79.30 \pm 0.55$ | $52.19 \pm 4.20$ | $0.62 \pm 0.68$ | $0.84 \pm 1.13$ |
| | G-FairAttack | $77.60 \pm 0.97$ | $57.90 \pm 5.65$ | $\mathbf{3.20 \pm 2.88}$ | $\mathbf{4.12 \pm 3.65}$ |
| EDITS | Clean | $73.89 \pm 1.99$ | $61.19 \pm 3.47$ | $4.36 \pm 2.94$ | $3.59 \pm 4.22$ |
| | Random | $78.88 \pm 0.97$ | $62.58 \pm 3.75$ | $1.83 \pm 0.88$ | $\underline{3.02 \pm 1.12}$ |
| | FA-GNN | $78.66 \pm 0.00$ | $58.58 \pm 3.62$ | $0.00 \pm 0.00$ | $0.00 \pm 0.00$ |
| | Metattack | $78.77 \pm 2.35$ | $69.58 \pm 5.19$ | $\underline{2.77 \pm 0.94}$ | $0.84 \pm 0.77$ |
| | G-FairAttack | $75.58 \pm 5.61$ | $64.60 \pm 8.87$ | $\mathbf{6.96 \pm 4.05}$ | $\mathbf{5.19 \pm 7.07}$ |

## F.5 ADVANCED ATTACK BASELINES

We supplement experiments on more previous attacks with fairness-targeted adaptations. We choose two more recent adversarial attacks for GNNs in terms of prediction utility, namely MinMax (Wu et al., 2019) and PRBCD (Geisler et al., 2021). To satisfy our fairness attack settings, we modify the attacker's objective with the demographic parity loss term (in the same way as adapting gradient ascent attack and Metattack). We implement all attack baselines in a fairness evasion attack setting with the same budget (5%). We compare the performance of these attacks with G-FairAttack based on four different victim models with GraphSAGE as the GNN backbone on the Facebook dataset. Results are shown in Table 15. From the experimental results, we can observe that (1) G-FairAttack

Table 15: Experiment results of fairness evasion attack on the Facebook dataset compared with more advanced attack baselines. All victim models adopt GraphSAGE as the GNN backbone.

| | Attack | ACC(%) | AUC(%) | $\Delta_{dp}(\%)$ | $\Delta_{eo}(\%)$ | Train ACC(%) |
|---|---|---|---|---|---|---|
| SAGE | Clean | $92.36 \pm 0.32$ | $94.06 \pm 0.14$ | $16.71 \pm 0.79$ | $10.10 \pm 0.41$ | $100.00 \pm 0.00$ |
| | PRBCD | $92.68 \pm 0.00$ | $94.26 \pm 0.10$ | $15.23 \pm 0.31$ | $8.11 \pm 0.33$ | $99.62 \pm 0.00$ |
| | MinMax | $92.99 \pm 0.00$ | $94.32 \pm 0.15$ | $\underline{15.69 \pm 0.31}$ | $\underline{8.69 \pm 0.33}$ | $98.66 \pm 0.19$ |
| | G-FairAttack | $92.15 \pm 0.37$ | $93.89 \pm 0.15$ | $\mathbf{18.71 \pm 0.74}$ | $\mathbf{11.91 \pm 0.97}$ | $100.00 \pm 0.00$ |
| Reg | Clean | $91.29 \pm 0.49$ | $93.69 \pm 0.13$ | $0.86 \pm 0.83$ | $1.79 \pm 1.87$ | $98.66 \pm 0.33$ |
| | PRBCD | $91.72 \pm 0.85$ | $94.00 \pm 0.07$ | $\underline{1.33 \pm 2.00}$ | $\underline{1.73 \pm 0.33}$ | $98.08 \pm 0.51$ |
| | MinMax | $91.61 \pm 0.49$ | $94.01 \pm 0.08$ | $0.77 \pm 0.41$ | $\underline{1.73 \pm 0.33}$ | $97.19 \pm 0.11$ |
| | G-FairAttack | $91.19 \pm 0.37$ | $93.66 \pm 0.12$ | $\mathbf{1.80 \pm 0.60}$ | $\mathbf{2.12 \pm 1.20}$ | $98.53 \pm 0.29$ |
| FairGNN | Clean | $92.57 \pm 0.48$ | $93.97 \pm 0.20$ | $3.38 \pm 1.44$ | $2.02 \pm 0.58$ | $98.98 \pm 0.55$ |
| | PRBCD | $92.46 \pm 0.26$ | $94.03 \pm 0.26$ | $3.36 \pm 1.05$ | $\underline{1.93 \pm 0.33}$ | $98.53 \pm 0.29$ |
| | MinMax | $92.67 \pm 0.55$ | $94.12 \pm 0.27$ | $\underline{3.53 \pm 1.60}$ | $\underline{1.93 \pm 0.33}$ | $97.57 \pm 0.29$ |
| | G-FairAttack | $92.57 \pm 0.18$ | $93.80 \pm 0.23$ | $\mathbf{4.40 \pm 1.14}$ | $\mathbf{2.05 \pm 1.25}$ | $98.98 \pm 0.55$ |
| EDITS | Clean | $93.42 \pm 0.18$ | $93.66 \pm 0.07$ | $9.12 \pm 0.58$ | $1.67 \pm 0.78$ | $98.34 \pm 0.11$ |
| | PRBCD | $93.10 \pm 0.73$ | $93.88 \pm 0.14$ | $\underline{10.78 \pm 1.15}$ | $\mathbf{5.41 \pm 2.34}$ | $97.89 \pm 0.19$ |
| | MinMax | $93.20 \pm 0.37$ | $93.88 \pm 0.13$ | $\mathbf{10.94 \pm 1.70}$ | $\underline{4.31 \pm 1.90}$ | $97.45 \pm 0.29$ |
| | G-FairAttack | $93.10 \pm 0.15$ | $93.88 \pm 0.15$ | $\underline{10.78 \pm 1.15}$ | $\mathbf{5.41 \pm 2.34}$ | $97.95 \pm 0.22$ |

Table 16: Experiment results of fairness evasion attack on the German dataset. All victim models adopt GraphSAGE as the GNN backbone.

| | Attack | ACC(%) | AUC(%) | $\Delta_{dp}(\%)$ | $\Delta_{eo}(\%)$ |
|---|---|---|---|---|---|
| SAGE | Clean | $58.80 \pm 1.06$ | $66.56 \pm 0.98$ | $58.02 \pm 5.05$ | $56.20 \pm 5.76$ |
| | Random | $58.80 \pm 1.06$ | $66.44 \pm 1.02$ | $56.32 \pm 2.70$ | $53.57 \pm 3.01$ |
| | FA-GNN | $59.60 \pm 0.80$ | $68.44 \pm 0.96$ | $55.18 \pm 3.90$ | $52.66 \pm 4.88$ |
| | Gradient Ascent | $58.13 \pm 1.51$ | $66.14 \pm 1.00$ | $\mathbf{59.52 \pm 4.58}$ | $\mathbf{57.99 \pm 4.93}$ |
| | G-FairAttack | $58.00 \pm 1.74$ | $65.75 \pm 1.21$ | $\underline{58.73 \pm 4.81}$ | $\underline{56.83 \pm 4.95}$ |
| Reg | Clean | $61.87 \pm 2.41$ | $61.40 \pm 0.78$ | $2.91 \pm 2.28$ | $4.34 \pm 1.76$ |
| | Random | $61.33 \pm 2.01$ | $61.44 \pm 0.96$ | $3.90 \pm 1.35$ | $5.18 \pm 1.08$ |
| | FA-GNN | $62.00 \pm 3.02$ | $61.79 \pm 0.65$ | $3.69 \pm 2.11$ | $\underline{5.22 \pm 2.05}$ |
| | Gradient Ascent | $61.20 \pm 2.40$ | $60.53 \pm 1.02$ | $\underline{3.97 \pm 4.17}$ | $4.62 \pm 2.99$ |
| | G-FairAttack | $61.33 \pm 2.27$ | $61.49 \pm 1.44$ | $\mathbf{4.61 \pm 3.49}$ | $\mathbf{6.65 \pm 2.53}$ |
| FairGNN | Clean | $64.40 \pm 3.12$ | $59.73 \pm 3.99$ | $4.26 \pm 3.39$ | $5.36 \pm 2.32$ |
| | Random | $64.53 \pm 3.11$ | $59.71 \pm 3.83$ | $4.61 \pm 2.80$ | $5.67 \pm 3.63$ |
| | FA-GNN | $64.67 \pm 2.89$ | $60.03 \pm 4.06$ | $4.40 \pm 2.85$ | $\mathbf{5.95 \pm 3.22}$ |
| | Gradient Ascent | $64.80 \pm 3.17$ | $59.40 \pm 4.62$ | $\underline{4.90 \pm 2.86}$ | $5.64 \pm 2.35$ |
| | G-FairAttack | $64.27 \pm 2.89$ | $59.66 \pm 3.57$ | $\mathbf{5.46 \pm 3.14}$ | $\underline{5.92 \pm 2.33}$ |
| EDITS | Clean | $68.27 \pm 1.97$ | $57.48 \pm 2.13$ | $2.17 \pm 1.22$ | $3.12 \pm 1.77$ |
| | Random | $68.13 \pm 1.22$ | $58.90 \pm 1.62$ | $\mathbf{2.81 \pm 2.14}$ | $\mathbf{3.41 \pm 2.13}$ |
| | FA-GNN | $68.27 \pm 1.97$ | $57.48 \pm 2.14$ | $2.17 \pm 1.22$ | $3.12 \pm 1.77$ |
| | Gradient Ascent | $68.13 \pm 1.22$ | $58.90 \pm 1.62$ | $\mathbf{2.81 \pm 2.14}$ | $\mathbf{3.41 \pm 2.13}$ |
| | G-FairAttack | $68.13 \pm 1.22$ | $58.90 \pm 1.62$ | $\mathbf{2.81 \pm 2.14}$ | $\mathbf{3.41 \pm 2.13}$ |

has the most desirable performance in attacking different types of (fairness-aware) victim models and (2) G-FairAttack best preserves the prediction utility of victim models over the training set. In conclusion, we obtain that our proposed surrogate loss and constrained optimization technique help G-FairAttack address the two proposed challenges of fairness attacks while a simple adaptation of previous attacks is not effective in solving these challenges.

## G DEFENSE AGAINST FAIRNESS ATTACKS OF GNNS

In this paper, the purpose of investigating the fairness attack problem on GNNs is to highlight the vulnerability of GNNs on fairness and to inspire further research on the fairness defense of GNNs. Hence, we would like to discuss the defense against fairness attacks of GNNs. Considering the difficulty in fairness defense, this topic deserves a careful further study, and we only provide some simple insights on defending against fairness attacks of GNNs in this section.

Table 17: Experiment results of fairness poisoning attack on the German dataset. All victim models adopt GraphSAGE as the GNN backbone.

| | Attack | ACC(%) | AUC(%) | $\Delta_{dp}$(%) | $\Delta_{eo}$(%) |
|---|---|---|---|---|---|
| SAGE | Clean | $59.60 \pm 1.06$ | $63.87 \pm 1.30$ | $41.29 \pm 7.36$ | $36.94 \pm 7.94$ |
| | Random | $62.13 \pm 1.97$ | $65.48 \pm 0.57$ | $41.52 \pm 8.40$ | $38.34 \pm 5.29$ |
| | FA-GNN | $64.67 \pm 1.01$ | $71.22 \pm 2.08$ | $43.99 \pm 0.63$ | $42.79 \pm 2.16$ |
| | Metattack | $58.13 \pm 3.63$ | $64.03 \pm 0.71$ | $\underline{46.31 \pm 3.22}$ | $\mathbf{43.87 \pm 3.49}$ |
| | G-FairAttack | $63.87 \pm 2.34$ | $66.10 \pm 1.68$ | $\mathbf{47.98 \pm 4.99}$ | $\underline{43.56 \pm 5.48}$ |
| Reg | Clean | $58.00 \pm 2.50$ | $60.70 \pm 2.35$ | $0.80 \pm 0.28$ | $1.37 \pm 1.55$ |
| | Random | $60.13 \pm 2.84$ | $61.83 \pm 1.25$ | $\underline{0.80 \pm 0.13}$ | $3.15 \pm 2.83$ |
| | FA-GNN | $60.80 \pm 1.74$ | $64.42 \pm 0.64$ | $0.72 \pm 0.88$ | $4.83 \pm 1.37$ |
| | Metattack | $58.67 \pm 1.01$ | $60.21 \pm 0.25$ | $0.42 \pm 0.33$ | $2.59 \pm 0.52$ |
| | G-FairAttack | $57.87 \pm 3.72$ | $60.42 \pm 1.48$ | $\mathbf{3.38 \pm 1.86}$ | $\mathbf{7.77 \pm 3.48}$ |
| FairGNN | Clean | $60.91 \pm 2.58$ | $62.15 \pm 3.70$ | $1.87 \pm 1.93$ | $1.48 \pm 1.50$ |
| | Random | $66.33 \pm 2.98$ | $63.76 \pm 5.71$ | $\underline{4.50 \pm 3.57}$ | $\underline{6.33 \pm 4.44}$ |
| | FA-GNN | $68.22 \pm 3.36$ | $63.07 \pm 5.19$ | $2.59 \pm 2.73$ | $1.86 \pm 1.10$ |
| | Metattack | $67.78 \pm 4.92$ | $63.04 \pm 4.35$ | $2.06 \pm 2.09$ | $1.98 \pm 1.07$ |
| | G-FairAttack | $65.34 \pm 4.70$ | $64.32 \pm 6.83$ | $\mathbf{7.86 \pm 1.68}$ | $\mathbf{10.64 \pm 2.54}$ |
| EDITS | Clean | $64.01 \pm 1.83$ | $65.59 \pm 0.92$ | $12.74 \pm 1.44$ | $6.66 \pm 3.28$ |
| | Random | $61.90 \pm 1.16$ | $66.18 \pm 1.56$ | $10.02 \pm 3.78$ | $7.84 \pm 5.78$ |
| | FA-GNN | $63.45 \pm 2.51$ | $64.66 \pm 1.55$ | $\underline{19.45 \pm 2.53}$ | $\underline{13.62 \pm 3.63}$ |
| | Metattack | $62.24 \pm 0.51$ | $63.46 \pm 1.97$ | $15.92 \pm 8.11$ | $10.77 \pm 3.41$ |
| | G-FairAttack | $65.34 \pm 1.02$ | $65.31 \pm 1.49$ | $\mathbf{20.43 \pm 5.37}$ | $\mathbf{15.30 \pm 5.53}$ |

**(1).** According to the study in (Hussain et al., 2022), injecting edges that belong to DD and EE groups can increase the statistical parity difference. Hence, a simple fairness defense strategy is to delete edges in DD and EE groups randomly. This strategy makes it possible to remove some poisoned edges in the input graph. However, this method cannot defend against G-FairAttack because G-FairAttack can poison the edges in all groups (EE, ED, DE, and DD).

**(2).** In our opinion, preprocessing debiasing frameworks such as EDITS can be a promising paradigm for fairness defense. Next, we explain the reasons in detail. We first review the processes of EDITS framework. EDITS is a preprocessing framework for GNNs (Dong et al., 2022a). First, we feed the clean graph into EDITS framework and obtain a debiased graph by reconnecting some edges and changing the node attributes where we can modify the debiasing extent with a threshold. Then, EDITS runs a vanilla GNN model (without fairness consideration), such as GCN, on the debiased graph. Finally, we find that the output of GCN on the debiased graph is less biased than the output of GCN on the clean graph. As a preprocessing framework, EDITS would flip the edges *again* to obtain a debiased graph for training *after* we poison the graph structure by attacking methods. Consequently, we can obtain that *EDITS can obtain very similar debiased graphs for any two different poisoned graphs with a strict debiasing threshold,* while the accuracy will also decrease as the debiasing threshold becomes stricter because the graph structure has been changed too much. In conclusion, EDITS has a tradeoff between the debiasing effect and the prediction utility. EDITS can be a strong fairness defense method for GNNs by sacrificing the prediction utility.

**(3).** A possible defense strategy against fairness attacks of GNNs is to solve a similar optimization problem as Problem 1 while minimizing the attacker's objective as

$$\min_{\mathbf{A}' \in \mathcal{F}} \mathcal{L}_f \left( g_{\boldsymbol{\theta}^*}, \mathbf{A}', \mathbf{X}, \mathcal{Y}, \mathcal{V}_{\text{test}}, \mathcal{S} \right)$$

$$s.t. \ \boldsymbol{\theta}^* = \arg\min_{\boldsymbol{\theta}} \mathcal{L}_s \left( g_{\boldsymbol{\theta}}, \mathbf{A}', \mathbf{X}, \mathcal{Y}, \mathcal{S} \right), \ \|\mathbf{A}' - \mathbf{A}\|_F \leq 2\Delta, \tag{17}$$

$$\mathcal{L}(g_{\boldsymbol{\theta}^*}, \mathbf{A}, \mathbf{X}, \mathcal{Y}, \mathcal{V}_{\text{train}}) - \mathcal{L}(g_{\boldsymbol{\theta}^*}, \mathbf{A}', \mathbf{X}, \mathcal{Y}, \mathcal{V}_{\text{train}}) \leq \epsilon.$$

The meaning of this optimization problem is to find the rewired graph structure that minimizes the prediction bias. The prediction bias is computed based on the model trained on the rewired graph. It can be seen as an inverse process of G-FairAttack. As a result, we can rewire the problematic edges that hurt the fairness of the model trained on the rewired graph.

## H    BROADER IMPACT

Adversarial attacks on fairness can make a significant impact in real-world scenarios (Solans et al., 2021; Mehrabi et al., 2021; Hussain et al., 2022). In particular, fairness attacks can exist in many different real-world scenarios.

- For personal benefits, malicious attackers can exploit the fairness attack to affect a GNN model (for determining the salary of an employer or the credit/loan of a user account) into favoring specific demographic groups by predicting higher values of money while disadvantaging other groups.
- For commercial competitions, a malicious competitor can attack the fairness of a GNN-based recommender system deployed by a tech company and make its users unsatisfied, especially when defending techniques of GNNs' utility have been widely studied while defending techniques of GNNs' fairness remain undeveloped.
- For governmental credibility, malicious adversaries can attack models used by a government agency with the goal of making them appear unfair in order to depreciate their value and credibility.

In addition, adversarial attack on fairness is widely studied on independent and identically distributed data. Extensive works (Chang et al., 2020b; Solans et al., 2021; Mehrabi et al., 2021; Van et al., 2022; Chhabra et al., 2023) have verified the vulnerability of algorithmic fairness of machine learning models. In this paper, we find the vulnerability of algorithmic fairness also exists in GNNs by proposing a novel adversarial attack on fairness of GNNs. It has the potential risk of being leveraged by malicious attackers with access to the input data of a deployed GNN model. Despite that, our research has a larger positive influence compared with the potential risk. Considering the lack of defense methods on fairness of GNNs, our study highlights the vulnerability of GNNs in terms of fairness and inspires further research on the fairness defense of GNNs. Moreover, we also provide discussions on attack patterns and simple ways to defend against fairness attacks.

