# OpenReview forum: "Adversarial Attacks on Fairness of Graph Neural Networks"
_ICLR.cc/2024/Conference — ICLR 2024 poster_

### Official Review · Reviewer_dSr8 · 2023-10-30

**Soundness:** 2 fair
**Presentation:** 2 fair
**Contribution:** 2 fair
**Rating:** 6
**Confidence:** 5

**Summary:**

The paper discusses the growing importance of fairness-aware Graph Neural Networks (GNNs) and their vulnerability to adversarial attacks. The authors introduce the G-FairAttack framework as a tool to compromise the fairness of GNNs without significantly affecting prediction utility. They also present a fast computation technique to enhance the efficiency of G-FairAttack.

**Strengths:**

1.Relevance of Topic: The paper addresses the crucial and timely subject of fairness in GNNs, a significant area in AI research.
2.Innovative Framework: The introduction of the G-FairAttack framework, complemented by a fast computation technique, offers a novel perspective on understanding vulnerabilities in fairness-aware GNNs.

**Weaknesses:**

1． Unpractical attack setting. The proposed evasion attack is not practical as there is no motivation for the model owner to replace data in a transductive learning setting, as shown by equation 1.

2． Overclaimed contribution. The evidence is needed when presenting “In this way, the surrogate model trained by our surrogate loss will be close to that trained by any unknown victim loss, which is consistent with conventional attacks on model utility.”

3． Untenable theoretical analysis. The theorem 1 is proved by unconvincing assumptions, e.g., $P_{\hat{Y}}(z) \geq \Pi_i \operatorname{Pr}(S=i)$ and other assumptions. Please be aware there is a difference between assumption and proof. The remarks for theorem 1 indicate its unconvincing nature. A lot of logical error exists in the proof part. For example, it is hard to say Pr(s=0)Pr(s=1)<=1/4 without evidence. In the paragraph above eq (8), |P_{s=0}(z)- P_{s=0}(z)|<=1 always hold according to the definition of fairness, what is the meaning to prove it, as it cannot support the whole analysis pipeline. Authors are strongly suggested to revise it to avoid analysis mistakes in the proof.

4． Unclear experimental setting. What is the target GNN architecture when discussing the effectiveness? This point should be clarified to avoid attackers having knowledge of the target GNN architecture. E.g., using surrogate GCN to attack target GCN.

5． Missed baselines. According to an existing study, “Adversarial Inter-Group Link Injection Degrades the Fairness of Graph Neural Networks”, a baseline method in this paper is injecting inter-group links. This is practical in the proposed setting where attackers have knowledge of the training graph.

6. Metrics.  In table 7, It looks like the metattack achieves better attack performance than the proposed method when considering the the metric $\Delta dp / \Delta Acc$, which should be an effective metric to evaluate the efficiency of the proposed method.

**Questions:**

Refer to the weakness part.

---

> ### Author Response · Authors · 2023-11-17
> **Author Response (1/2)**
>
> Thank you very much for your thoughtful feedback. We are honored to have this valuable chance to address your raised concerns and questions. We believe that our paper will be much stronger thanks to your efforts.
>
> > Unpractical attack setting: The proposed evasion attack is not practical as there is no motivation for the model owner to replace data in a transductive learning setting, as shown by equation 1.
>
> Response: We thank the reviewer for pointing this out and would like to clarify the misunderstanding of the motivation for studying fairness evasion attacks. In the real-world deployment of graph learning models, the data can be modified by attackers rather than the model owner. It is possible for attackers to attack the access control of the databases, especially for edge computing systems with a coarse-grained access control [1,2]. In addition, once the model is deployed, the attacker can launch evasion attacks *at any time*, which increases the difficulty and cost of defending against evasion attacks [3]. Considering the severe impact of evasion attacks, many prevalent existing works [3,4,5,6] make great efforts to study evasion attacks in a transductive setting which is adopted by a substantial part of GNNs. In addition, our fairness evasion attack can be easily adapted to an inductive setting by modifying the training graph to the test graph. In our experiment, we use a transductive learning setting since all adopted victim models are implemented in a transductive setting as well.
>
> [1] Ali, I., et al. "Internet of Things Security, Device Authentication and Access Control: A Review." IJCSIS 2016.
>
> [2] Xiao, Y., et al. "Edge computing security: State of the art and challenges." Proceedings of the IEEE 2019.
>
> [3] Zhang, H, et al. "Projective ranking-based gnn evasion attacks." TKDE 2022.
>
> [4] Dai, H., et al. "Adversarial attack on graph structured data." ICML 2018.
>
> [5] Zügner, D., et al. "Adversarial attacks on neural networks for graph data." KDD 2018.
>
> [6] Zügner, D., et al. "Adversarial Attacks on Graph Neural Networks via Meta Learning." ICLR 2019.
>
> > Overclaimed contribution: The evidence is needed when presenting “In this way, the surrogate model trained by our surrogate loss will be close to that trained by any unknown victim loss, which is consistent with conventional attacks on model utility.”
>
> Response: Thanks for pointing this out. However, we believe there exists a misunderstanding of this sentence. We should first clarify that the mentioned sentence only *interpreted the intuition of the design* of our surrogate loss function rather than describing our contributions. In addition, we agree with the reviewer that evidence is needed for intuition, and Theorem 1 is the evidence for this intuition. In Theorem 1, we proved that $TV(\hat{Y},S)$ is an upper bound of all types of fairness loss terms. Consequently, by training with $TV(\hat{Y},S)$, our surrogate model has a small value over all different types of fairness loss function $\Delta_{dp}(\hat{Y},S)$, $I(\hat{Y},S)$, and $W(\hat{Y},S)$, just as all types of fairness-aware GNNs. Hence, our surrogate model can be seen as a better surrogation of the unknown victim model. In conclusion, we argue that we have provided valid evidence for the mentioned intuition.
>
> > Unclear experimental setting: What is the target GNN architecture when discussing the effectiveness? This point should be clarified to avoid attackers having knowledge of the target GNN architecture. E.g., using surrogate GCN to attack target GCN.
>
> Response: We thank the reviewer for pointing out this crucial point. We provided the details of the target GNN architecture in Appendix E.2. The architecture of each target model was set following the default settings in [1, 2] (single/two-layer GCN with the ReLU activation). In addition, there is no need to worry about the knowledge leakage. We changed the type of victim model to GraphSAGE and GAT and preserved the surrogate model as two-layer linearized GCN and G-FairAttack still achieved a desirable performance (Table 11-14).
>
> [1] Dai, E., & Wang, S. "Say no to the discrimination: Learning fair graph neural networks with limited sensitive attribute information." WSDM 2021.
>
> [2] Dong, Y., et al. "Edits: Modeling and mitigating data bias for graph neural networks." TheWebConf 2022.

---

> ### Author Response · Authors · 2023-11-17
> **Author Response (2/2)**
>
> > Untenable theoretical analysis: The theorem 1 is proved by unconvincing assumptions, e.g., $P_{\hat{Y}}(z)\geq\Pi_i\mathrm{Pr}(S=i)$ and other assumptions. Please be aware there is a difference between assumption and proof. The remarks for theorem 1 indicate its unconvincing nature.
>
> Response: Thanks for mentioning this point. However, we argue that our assumptions regarding Theorem 1 are reasonable as (1) the probability of the condition holds grows larger when the number of sensitive groups increases; (2) even in the binary case, the condition is highly likely to hold in practice (explained in Appendix A.1). Moreover, we added a remark on Theorem 1 to show the best we could achieve theoretically to make the assumption practical, and **the assumption in the remark had been verified to be feasible on all adopted datasets**. Other than the theoretical aspect, the empirical results on attacking FairGNN also verified the efficacy of our surrogate model in representing $I(\hat{Y}, S)$-based fairness-aware GNNs. To avoid misunderstanding, we will add a more clear statement of the assumption to Theorem 1.
>
> > A lot of logical error exists in the proof part. For example, it is hard to say $Pr(s=0)Pr(s=1)<=1/4$ without evidence. In the paragraph above eq (8), $|P_{s=0}(z)- P_{s=0}(z)|<=1$ always hold according to the definition of fairness, what is the meaning to prove it, as it cannot support the whole analysis pipeline. Authors are strongly suggested to revise it to avoid analysis mistakes in the proof.
>
> Response: We thank the reviewer for raising these questions. However, we believe there exists some misunderstanding. First, we clarify the correctness of $\mathrm{Pr}(s=0)\mathrm{Pr}(s=1)<=1/4$ under the binary sensitive attribute. In a binary case, we have $\mathrm{Pr}(s=0)+\mathrm{Pr}(s=1)=1$. Hence, $\mathrm{Pr}(s=0)\mathrm{Pr}(s=1)=\mathrm{Pr}(s=0)(1-\mathrm{Pr}(s=0))\leq1/4$. Second, we will interpret the meaning of the assumption $|P_{\hat{Y}|S=0}(z)-P_{\hat{Y}|S=1}(z)|\leq 1$. In the total variation loss, we considered $P_{\hat{Y}|S=1}(z)$, the continuous pdf of the random variable $\hat{Y}\in[0,1]$ conditional on the random variable $s\in\{0,1\}$, rather than a discrete probability $\mathrm{Pr}(\hat{Y}=1|S=0)$. Hence, the value of $P_{\hat{Y}|S=0}(z)$ can be larger than 1 for some $z\in[0,1]$ and we made and verified the assumption $|P_{\hat{Y}|S=0}(z)-P_{\hat{Y}|S=1}(z)|\leq 1$ consequently.
>
> > Missed baselines: According to an existing study, “Adversarial Inter-Group Link Injection Degrades the Fairness of Graph Neural Networks”, a baseline method in this paper is injecting inter-group links. This is practical in the proposed setting where attackers have knowledge of the training graph.
>
> Response: Actually, we had adopted the mentioned baseline in our experiments as "FA-GNN". In this work [1], the author proposed four different types of link injection strategies and we chose the most powerful one, "DD" link injection, as our FA-GNN baseline.
>
> [1] Hussain, H., et al. "Adversarial Inter-Group Link Injection Degrades the Fairness of Graph Neural Networks." ICDM 2022.
>
> > Metrics. In table 7, It looks like the metattack achieves better attack performance than the proposed method when considering the the metric $\Delta_{dp}/\Delta_{Acc}$, which should be an effective metric to evaluate the efficiency of the proposed method.
>
> Response: We thank the reviewer for pointing this out. In Table 7, Metattack **only** performs better when attacking the vanilla GNN and G-FairAttack has more desirable results when attacking **all other types of** fairness-aware GNNs. In addition, we explained in Appendix F.1 that G-FairAttack's seemingly suboptimal performance on vanilla GNN can be solved by choosing a smaller $\alpha$. To demonstrate that G-FairAttack is victim model-agnostic and easy to use, we fixed rather than adjusted the value of $\alpha$ when attacking different victim models. Hence, we argue that the superiority of G-FairAttack is reflected by the fact that only G-FairAttack has a desirable attacking performance on all types of (fairness-aware) GNNs. In addition, we would also like to emphasize that except for having better empirical performance than all previous methods, we provided **sufficient insights** for our desirable performance in different perspectives, e.g., the reason why G-FairAttack performs better than gradient-based methods, the reason why only G-FairAttack has steadily satisfying performance in attacking different types of (fairness-aware) GNNs.
>
> We extend our best gratitude for your efforts in the rebuttal phase. We greatly value this opportunity to improve our paper. We have revised our paper based on all reviewer's comments with red text. If you have any further questions or concerns on our response, please feel free to engage in discussion with us. We sincerely appreciate your time and consideration.

---

> > ### Comment · Reviewer_dSr8 · 2023-11-21
> >
> > Dear authors,
> >
> > Thanks for your efforts in clarifying your paper. Hope my comments help to improve the manuscript. Your answers make sense to me and I have raised the score.

---

> ### Author Response · Authors · 2023-11-22
>
> Dear reviewer,
>
> We sincerely appreciate your thoughtful feedback and thank you again for your time and efforts in the discussion phase.
>
> Best,
>
> Authors

---

### Official Review · Reviewer_qWJG · 2023-10-31

**Soundness:** 2 fair
**Presentation:** 3 good
**Contribution:** 2 fair
**Rating:** 6
**Confidence:** 3

**Summary:**

The paper investigates adversarial attacks on the fairness of Graph Neural Networks (GNNs). The authors introduce an attack framework, G-FairAttack, designed to corrupt the fairness of various types of fairness-aware GNNs subtly, without noticeably affecting prediction utility.  G-FairAttack is formulated as an optimization problem, considering a gray-box attack setting where the attacker has limited knowledge of the model. The authors propose a surrogate loss function and a non-gradient attack algorithm to solve the optimization problem, ensuring that the attacks are unnoticeable and effectively compromise the fairness of the GNNs.

**Strengths:**

The introduction of G-FairAttack brings a new perspective to the understanding of adversarial attacks in the context of fairness-aware models.

By uncovering vulnerabilities related to fairness, the paper contributes valuable insights that can guide the development of more robust and ethical AI systems.

The paper includes extensive experiments that validate the effectiveness of the proposed attacks. This empirical evaluation strengthens the credibility of the findings and their relevance to practical scenarios involving fairness-aware GNNs.

**Weaknesses:**

The assumptions about the attacker's knowledge might not cover all possible real-world scenarios. The gray-box setting is a middle ground, but exploring both black-box and white-box attacks could provide a fuller picture of the vulnerabilities.

The performance of  G-FairAttack is worse than random attack under some scenarios in Table 1, 6 and 7.

**Questions:**

Would it be possible to apply the G-FairAttack framework to a broader range of datasets, such as those utilized in EDITS paper you referenced, rather than limiting the evaluation to only three datasets?

How much computational time is required to execute G-FairAttack?

Why do all the attack methods seem to have minimal influence on the utility score? Is there a trade-off between utility and fairness scores? How is the attack budget determined for fairness attacks, and under what circumstances would the utility score significantly decrease?

---

> ### Author Response · Authors · 2023-11-17
> **Author Response for Weaknesses**
>
> Thank you very much for your thoughtful feedback. We are honored to have this valuable chance to address your raised concerns and questions. We believe that our paper will be much stronger thanks to your efforts.
>
> > W1: The assumptions about the attacker's knowledge might not cover all possible real-world scenarios. The gray-box setting is a middle ground, but exploring both black-box and white-box attacks could provide a fuller picture of the vulnerabilities.
>
> Response: We agree with the reviewer that exploring different settings can provide a fuller picture, and we would like to add a discussion on extending G-FairAttack to white-box and black-box settings. First, G-FairAttack can be *directly* adapted to a white-box setting by replacing the trained surrogate model in the attacker's objective with the true victim model in Problem 1. However, designing fairness attacks in a black-box attack setting can be very challenging. The difference between grey-box attacks and black-box attacks is that black-box attackers are not allowed to access the ground truth labels. Different from node embeddings which can be obtained in an unsupervised way, group fairness metrics have to rely on the ground truth labels, which makes existing black-box attacks on graphs difficult to adapt to fairness attacks. Despite the difficulty of black-box fairness attacks, we then try our best to introduce a potential way to extend our framework to a black-box setting. First, the attacker can collect some data following a similar distribution, i.e., if the target graph is a Citeseer citation network, the attacker can collect data from Arxiv; if the target graph is a Facebook social network, the attacker can collect data from Twitter (X). It is worth noting that the dimension of collected node features should be aligned with the target graph. Then, the attacker can train a state-of-the-art inductive GNN model on the collected graph data and obtain the predicted labels on the target graph. Finally, the attacker can use the predicted labels as a pseudo label to implement our G-FairAttack on the target graph.
>
> > W2: The performance of G-FairAttack is worse than random attack under some scenarios in Table 1, 6 and 7.
>
> Response: We thank the reviewer for pointing this out. We argue that the baselines such as the random method only outperform G-FairAttack in very few cases due to the randomness in the challenging fairness-attacking task. In these cases, the superiority of the random attack is marginal and not statistically significant. It is obvious that the overall performance of G-FairAttack is the most desirable compared with all baselines and the superiority of G-FairAttack is solid. We emphasize that the most important advantage of G-FairAttack is that it can achieve generally desirable attack performance under **different types of** fairness-aware GNNs. In addition, we would also like to emphasize that except for having better empirical performance than all previous methods, we provided **sufficient insights** for our desirable performance in different perspectives, e.g., the reason why G-FairAttack performs better than gradient-based methods, the reason why only G-FairAttack has steadily satisfying performance in attacking different types of (fairness-aware) GNNs.

---

> ### Author Response · Authors · 2023-11-17
> **Author Response for Questions**
>
> > Q1: Would it be possible to apply the G-FairAttack framework to a broader range of datasets, such as those utilized in EDITS paper you referenced, rather than limiting the evaluation to only three datasets?
>
> Response: We add new experimental results of fairness evasion attacks and fairness poisoning attacks on the German Credit dataset in Tables 16 and 17.
>
> > Q2: How much computational time is required to execute G-FairAttack?
>
> Response: The computational time for G-FairAttack on Facebook is 1.80 s/edge under the evasion attack and 5.62 s/edge under the poisoning attack; on Pokec\_z is 22.41 s/edge under the evasion attack and 48.24 s/edge under the poisoning attack; on Credit is 35.44 s/edge under the evasion attack and 40.25 s/edge under the poisoning attack. These results follow the settings in Table 4. It is worth noting that the computational time can be further reduced by decreasing the value of $a$ according to the results in Figure 3. In addition, we provided a detailed time complexity analysis in Appendix C.
>
> > Q3: Why do all the attack methods seem to have minimal influence on the utility score?
>
> Response: In our opinion, all baselines have a relatively small influence on the utility score because we set a small attack budget (5\% or 1\% of the total edges). Compared with evasion attacks (only input data is modified, the victim model remains unchanged), poisoning attacks result in a larger utility change since other than the input data, the victim model is also changed (being trained on the perturbed graph) compared with the original one.
>
> > Q3: Is there a trade-off between utility and fairness scores?
>
> Response: Many existing works have studied the fairness-utility tradeoff in fair machine learning [1-3]. However, attackers do not need to consider the tradeoff as utility is not considered in the objectives in our setting. According to the specific needs, attackers can also add a utility term to the objective to jointly attack the fairness and the utility of the victim model.
>
> [1] Dutta, S., et al. "Is there a trade-off between fairness and accuracy? a perspective using mismatched hypothesis testing." ICML 2020.
>
> [2] Ge, Y., et al. "Toward Pareto efficient fairness-utility trade-off in recommendation through reinforcement learning." WSDM 2022.
>
> [3] Li, P., et al. "Achieving fairness at no utility cost via data reweighing with influence." ICML 2022.
>
> > Q3: How is the attack budget determined for fairness attacks?
>
> Response: To set the attack budget, we followed the existing works on utility attacks to determine the number of edges that can be flipped and constrained the change of utility loss function over the training set to be no more than 5\% for our proposed unnoticeable effect.
>
> > Q3: And under what circumstances would the utility score significantly decrease?
>
> Response: Existing works on utility attacks were proven to be effective in corrupting the model utility [1]. When attackers add a utility term to the objective, the utility score might significantly decrease.
>
> [1] Jin, Wei, et al. "Adversarial attacks and defenses on graphs." SIGKDD Explorations 2021.
>
> We extend our best gratitude for your efforts in the rebuttal phase. We greatly value this opportunity to improve our paper. We have revised our paper based on all reviewer's comments with red text. If you have any further questions or concerns on our response, please feel free to engage in discussion with us. We sincerely appreciate your time and consideration.

---

> > ### Comment · Reviewer_qWJG · 2023-11-23
> >
> > Thank you for the authors' response. Based on the answer to Q2, I note that G-FairAttack has a limitation in computational speed, especially with large datasets. However, since my other concerns have been addressed, I am willing to increase my score to 6.

---

> > > ### Author Response · Authors · 2023-11-23
> > >
> > > We deeply appreciate the reviewer for providing further feedback. Hereby, we would like to take this valuable opportunity to address the last concern as much as possible. We would like to point out that (1) in the experiments, we chose the hyperparameters (especially $a$) to make the running time of G-FairAttack fall in a feasible range, and *the computational speed could be further improved* by decreasing the value of $a$ without distinctly compromising the attack efficacy (see Section 4.4 Parameter Study for more detailed results); (2) according to our complexity analysis, G-FairAttack has a feasible time complexity ($O(d_xan)$) which is *smaller than the prevalent attack methods*. Hence, we argue that the computational speed is *no limitation* for G-FairAttack, and we would like to take more scalable attacks as a future direction considering that faster attack algorithms can always be better. We thank the reviewer again for the constructive suggestions and dedicated time in the discussion phase.

---

### Official Review · Reviewer_B8qV · 2023-10-31

**Soundness:** 3 good
**Presentation:** 2 fair
**Contribution:** 3 good
**Rating:** 6
**Confidence:** 4

**Summary:**

- The paper studies the problem of adversarial attacks on fairness of GNNs. The authors propose a general framework called G-FairAttack to attack various types of fairness-aware GNNs from the perspective of fairness, with an unnoticeable impact on prediction utility. The authors employ a greedy strategy and propose a non-gradient sequential attack method. In addition, the authors introduce a fast computation technique to reduce the time complexity of G-FairAttack.

**Strengths:**

- The paper is well written and easy to read.
- The proposed unnoticeable fairness attacks of GNNs are novel and interesting.
- The theoretical analysis demonstrates that the designed surrogate loss function serves as a common upper bound for three fairness loss functions.

**Weaknesses:**

- Grey-box attack scenarios are relatively uncommon in real-world applications. I believe it would be more interesting if it could be extended to black-box attack settings.
- In terms of the utility metrics, G-FairAttack and the baseline seem to have a relatively small difference. I believe this does not fully reflect the authors' claim of making attacks unnoticeable. In other words, the issue mentioned by the authors in the introduction, "no existing work considers unnoticeable utility change in fairness attacks," does not appear to be very pressing.

**Questions:**

- In Table 1, when the victim is EDITS, and the dataset is Pokec_z, did any issues arise with the baseline, or were the results of all four baselines identical?
- Regarding the invisibility of fairness attacks, should consideration extend to structural modifications that are not easily noticeable, apart from merely constraining them through budget limitations?

---

> ### Author Response · Authors · 2023-11-17
> **Author Response for Weaknesses**
>
> Thank you very much for your thoughtful feedback. We are honored to have this valuable chance to address your raised concerns and questions. We believe that our paper will be much stronger thanks to your efforts.
>
> > W1: Grey-box attack scenarios are relatively uncommon in real-world applications. I believe it would be more interesting if it could be extended to black-box attack settings.
>
> Response: We agree with the reviewer that the black-box setting is more interesting. However, the grey-box attack is also a realistic setting that has been widely studied [1-4]. We made some real-world use cases of grey-box attacks in the introduction. Actually, grey-box attackers only have one more knowledge, the ground truth label, compared with black-box attackers [5]. For most real-world tasks, the ground truth label is not difficult to obtain (except for predicting some sensitive information). However, designing fairness attacks without ground truth labels can be very challenging. Different from node embeddings which can be obtained in an unsupervised way, group fairness metrics have to rely on the ground truth labels, which makes existing black-box attacks on graphs difficult to adapt to fairness attacks. Despite the difficulty of black-box fairness attacks, we would like to try our best to introduce a potential way to extend our framework to a black-box setting. First, the attacker can collect some data following a similar distribution, i.e., if the target graph is a Citeseer citation network, the attacker can collect data from Arxiv; if the target graph is a Facebook social network, the attacker can collect data from Twitter (X). It is worth noting that the dimension of collected node features should be aligned with the target graph. Then, the attacker can train a state-of-the-art inductive GNN model on the collected graph data and obtain the predicted labels on the target graph. Finally, the attacker can use the predicted labels as a pseudo label to implement our G-FairAttack on the target graph.
>
> [1] Zügner, D., et al. "Adversarial attacks on neural networks for graph data." KDD 2018.
>
> [2] Zügner, D., and Stephan G. "Adversarial Attacks on Graph Neural Networks via Meta Learning." ICLR 2019.
>
> [3] Wang, B., and Neil G. "Attacking graph-based classification via manipulating the graph structure." CCS 2019.
>
> [4] Liu, X., et al. "A Unified Framework for Data Poisoning Attack to Graph-based Semi-supervised Learning." NeurIPS 2019.
>
> [5] Jin, Wei, et al. "Adversarial attacks and defenses on graphs." SIGKDD Explorations 2021.
>
> > W2: In terms of the utility metrics, G-FairAttack, and the baseline seem to have a relatively small difference. I believe this does not fully reflect the authors' claim of making attacks unnoticeable. In other words, the issue mentioned by the authors in the introduction, "no existing work considers unnoticeable utility change in fairness attacks," does not appear to be very pressing.
>
> Response: First, we should clarify that the unnoticeable effect is **only** related to the utility metrics in Table 15: the utility metrics in Table 1 to Table 14 are all on the test set; however, the test set utility cannot be monitored by the model developer due to the missing labels. Hence, we only considered the utility change on the **training set** in our unnoticeable condition, as Problem 1 shows. In Table 15 (Train ACC column), we can observe that G-FairAttack always has the smallest training accuracy difference compared with the original model. In addition, our ablation study in Figure 2 also shows that our unnoticeable constraint distinctly decreases the model utility change over the training set.
>
> We then show the necessity of solving the unnoticeable issue, a pressing issue in fairness attacks on graphs. We can see in Table 15 that the accuracy of victim models over the training set is normally very high (close to 100\%). Hence, a small deterioration in the performance over the training set can be quite distinct to be noticed. Specifically, for two victim models (SAGE and FairGNN), our attacked input graph results in exactly the same accuracy over the training set as the clean graph, and the accuracy value is 100\% for SAGE.

---

> ### Author Response · Authors · 2023-11-17
> **Author Response for Questions**
>
> > Q1: In Table 1, when the victim is EDITS, and the dataset is Pokec_z, did any issues arise with the baseline, or were the results of all four baselines identical?
>
> Response: We thank the reviewer for pointing this out. We have included a comprehensive discussion on EDITS in Appendix G. Specifically, EDITS is a preprocessing framework for GNNs: We first feed the clean graph into the EDITS framework and obtain a debiased graph by reconnecting some edges and changing the node attributes where we can modify the debiasing extent with a threshold. Then, EDITS runs a vanilla GNN model (without fairness consideration), such as GCN, on the debiased graph. Finally, we find that the output of GCN on the debiased graph is less biased than the output of GCN on the clean graph. As a preprocessing framework, EDITS will flip the edges *again* to obtain a debiased graph for training *after* we poison the graph structure by attacking methods. Consequently, the input graphs under different attack baselines after preprocessing can be very similar, resulting in similar performance.
>
> > Q2: Regarding the invisibility of fairness attacks, should consideration extend to structural modifications that are not easily noticeable, apart from merely constraining them through budget limitations?
>
> Response: We agree with the reviewer that the definition of invisibility can be further extended. Currently, this paper focuses on a widely adopted setting, which is to set a budget for the perturbation. However, the proposed concern will be an interesting future work to explore. When considering the unnoticeability of structural modification, it will be desirable to preserve some important statistical information of the input graph, e.g., centrality, density, and degree distribution. A potential way to take the statistical information into account is to set the difference of the statistical information after the attack as the relaxed constraints. It is a non-trivial question as different statistical information should be tackled in different ways and time complexity should also be considered. In conclusion, we agree with the reviewer's opinion and think it is an interesting future direction for attacks on graphs.
>
> We extend our best gratitude for your efforts in the rebuttal phase. We greatly value this opportunity to improve our paper. We have revised our paper based on all reviewer's comments with red text. If you have any further questions or concerns on our response, please feel free to engage in discussion with us. We sincerely appreciate your time and consideration.

---

### Author Response · Authors · 2023-11-22

Dear reviewers,

We hope this message finds you well. As the discussion phase is coming to a conclusion, we are still eager for your further feedback. We believe that our revised submission has been much stronger thanks to your constructive comments. If any further information or clarification is needed, please feel free to let us know. We sincerely appreciate all reviewers for your time and efforts in the reviewing process.

Best,

Authors

---

### Comment · Area_Chair_ptBZ · 2023-11-23
**Please check author response**

Dear Reviewers,

If you haven't done so, please check the author response and reply with your feedback. Thank you!

Best,
AC

---

### Meta-Review · Area_Chair_ptBZ · 2023-12-15

**Metareview:**

This paper studies an interesting problem of the adversarial robustness of fairness in GNNs. The proposed method seems technically sound and is shown effective in empirical evaluation. The grey-box setting slightly undermines the realisticness of the proposed attack. Some other concerns were well-addressed by the authors during the response period. Overall the novel problem setting and method make the paper a solid contribution to the field.

**Justification For Why Not Higher Score:**

The setting may be unrealistic in many application scenarios.

**Justification For Why Not Lower Score:**

Interesting problem and sound method.

---

### Decision · Program_Chairs · 2024-01-16

Accept (poster)